# Expressiveness and Approximation Properties of Graph Neural Networks

**Floris Geerts**
Department of Computer Science, University of Antwerp, Belgium
`floris.geerts@uantwerpen.be`

**Juan L. Reutter**
School of Engineering, Pontificia Universidad Católica de Chile, Chile & IMFD, Chile
`jreutter@ing.puc.cl`

## Abstract

Characterizing the separation power of graph neural networks (GNNs) provides an understanding of their limitations for graph learning tasks. Results regarding separation power are, however, usually geared at specific GNN architectures, and tools for understanding arbitrary GNN architectures are generally lacking. We provide an elegant way to easily obtain bounds on the separation power of GNNs in terms of the Weisfeiler-Leman (WL) tests, which have become the yardstick to measure the separation power of GNNs. The crux is to view GNNs as expressions in a procedural tensor language describing the computations in the layers of the GNNs. Then, by a simple analysis of the obtained expressions, in terms of the number of indexes and the nesting depth of summations, bounds on the separation power in terms of the WL-tests readily follow. We use tensor language to define Higher-Order Message-Passing Neural Networks (or $k$-MPNNs), a natural extension of MPNNs. Furthermore, the tensor language point of view allows for the derivation of universality results for classes of GNNs in a natural way. Our approach provides a toolbox with which GNN architecture designers can analyze the separation power of their GNNs, without needing to know the intricacies of the WL-tests. We also provide insights in what is needed to boost the separation power of GNNs.

## 1 Introduction

Graph Neural Networks (GNNs) (Merkwirth & Lengauer, 2005; Scarselli et al., 2009) cover many popular deep learning methods for graph learning tasks (see Hamilton (2020) for a recent overview). These methods typically compute vector embeddings of vertices or graphs by relying on the underlying adjacency information. Invariance (for graph embeddings) and equivariance (for vertex embeddings) of GNNs ensure that these methods are oblivious to the precise representation of the graphs.

**Separation power.** Our primary focus is on the separation power of GNN architectures, i.e., on their ability to separate vertices or graphs by means of the computed embeddings. It has become standard to characterize GNN architectures in terms of the separation power of graph algorithms such as color refinement (CR) and $k$-dimensional Weisfeiler-Leman tests ($k$-WL), as initiated in Xu et al. (2019) and Morris et al. (2019). Unfortunately, understanding the separation power of any given GNN architecture requires complex proofs, geared at the specifics of the architecture. We provide a *tensor language-based technique* to analyze the separation power of general GNNs.

**Tensor languages.** Matrix query languages (Brijder et al., 2019; Geerts et al., 2021b) are defined to assess the expressive power of linear algebra. Balcilar et al. (2021a) observe that, by casting various GNNs into the MATLANG (Brijder et al., 2019) matrix query language, one can use existing separation results (Geerts, 2021) to obtain upper bounds on the separation power of GNNs in terms of 1-WL and 2-WL. In this paper, we considerably extend this approach by defining, and studying, a new general-purpose tensor language specifically designed for modeling GNNs. As in Balcilar et al. (2021a), our focus on tensor languages allows us to obtain new insights about GNN architectures.

First, since tensor languages can only define invariant and equivariant graph functions, any GNN that can be cast in our tensor language inherits these desired properties. More importantly, the separation power of our tensor language is as closely related to CR and $k$-WL as GNNs are. Loosely speaking, *if tensor language expressions use $k + 1$ indices, then their separation power is bounded by $k$-WL. Furthermore, if the maximum nesting of summations in the expression is $t$, then $t$ rounds of $k$-WL are needed to obtain an upper bound on the separation power.* A similar connection is obtained for CR and a fragment of tensor language that we call "guarded" tensor language.

We thus reduce problem of assessing the separation power of *any* specific GNN architecture to the problem of specifying it in our tensor language, analyzing the number of indices used and counting their summation depth. This is usually much easier than dealing with intricacies of CR and $k$-WL, as casting GNNs in our tensor language is often as simple as writing down their layer-based definition. We believe that this provides a nice toolbox for GNN designers to assess the separation power of their architecture. We use this toolbox to recover known results about the separation power of specific GNN architectures such as GINs (Xu et al., 2019), GCNs (Kipf & Welling, 2017), Folklore GNNs (Maron et al., 2019b), $k$-GNNs (Morris et al., 2019), and several others. We also derive new results: we answer an open problem posed by Maron et al. (2019a) by showing that the separation power of Invariant Graph Networks ($k$-IGNs), introduced by Maron et al. (2019b), is bounded by $(k-1)$-WL. In addition, we revisit the analysis by Balcilar et al. (2021b) of ChebNet (Defferrard et al., 2016), and show that CayleyNet (Levie et al., 2019) is bounded by 2-WL.

When writing down GNNs in our tensor language, the less indices needed, the stronger the bounds in terms of $k$-WL we obtain. After all, $(k-1)$-WL is known to be strictly less separating than $k$-WL (Otto, 2017). Thus, it is important to minimize the number of indices used in tensor language expressions. We connect this number to the notion of *treewidth*: expressions of treewidth $k$ can be translated into expressions using $k+1$ indices. This corresponds to optimizing expressions, as done in many areas in machine learning, by reordering the summations (a.k.a. variable elimination).

**Approximation and universality.** We also consider the ability of GNNs to approximate general invariant or equivariant graph functions. Once more, instead of focusing on specific architectures, we use our tensor languages to obtain general approximation results, which naturally translate to universality results for GNNs. We show: $(k+1)$-index tensor language expressions suffice to approximate any (invariant/equivariant) graph function whose separating power is bounded by $k$-WL, and we can further refine this by comparing the number of rounds in $k$-WL with the summation depth of the expressions. These results provide a finer picture than the one obtained by Azizian & Lelarge (2021). Furthermore, focusing on "guarded" tensor expressions yields a similar universality result for CR, a result that, to our knowledge, was not known before. We also provide the link between approximation results for tensor expressions and GNNs, enabling us to transfer our insights into universality properties of GNNs. As an example, we show that $k$-IGNs can approximate any graph function that is less separating than $(k-1)$-WL. This case was left open in Azizian & Lelarge (2021).

In summary, we draw new and interesting connections between tensor languages, GNN architectures and classic graph algorithms. We provide a general recipe to bound the separation power of GNNs, optimize them, and understand their approximation power. We show the usefulness of our method by recovering several recent results, as well as new results, some of them left open in previous work.

**Related work.** Separation power has been studied for specific classes of GNNs (Morris et al., 2019; Xu et al., 2019; Maron et al., 2019b; Chen et al., 2019; Morris et al., 2020; Azizian & Lelarge, 2021). A first general result concerns the bounds in terms of CR and 1-WL of Message-Passing Neural Networks (Gilmer et al., 2017; Morris et al., 2019; Xu et al., 2019). Balcilar et al. (2021a) use the MATLANG matrix query language to obtain upper bounds on the separation power of various GNNs. MATLANG can only be used to obtain bounds up to 2-WL and is limited to matrices. Our tensor language is more general and flexible and allows for reasoning over the number of indices, treewidth, and summation depth of expressions. These are all crucial for our main results. The tensor language introduced resembles sum-MATLANG (Geerts et al., 2021b), but with the added ability to represent tensors. Neither separation power nor guarded fragments were considered in Geerts et al. (2021b). See Section A in the supplementary material for more details. For universality, Azizian & Lelarge (2021) is closest in spirit. Our approach provides an elegant way to recover and extend their results. Azizian & Lelarge (2021) describe how their work (and hence also ours) encompasses previous works (Keriven & Peyré, 2019; Maron et al., 2019c; Chen et al., 2019). Our results use connections between $k$-WL and logics (Immerman & Lander, 1990; Cai et al., 1992), and CR and

guarded logics (Barceló et al., 2020). The optimization of algebraic computations and the use of treewidth relates to the approaches by Aji & McEliece (2000) and Abo Khamis et al. (2016).

## 2 BACKGROUND

We denote sets by $\{\}$ and multisets by $\{\{\}\}$. For $n \in \mathbb{N}$, $n > 0$, $[n] := \{1, \ldots, n\}$. Vectors are denoted by $v, w, \ldots$, matrices by $A, B, \ldots$, and tensors by $\mathsf{S}, \mathsf{T}, \ldots$. Furthermore, $v_i$ is the $i$-th entry of vector $v$, $A_{ij}$ is the $(i, j)$-th entry of matrix $A$ and $\mathsf{S}_i$ denotes the $i = (i_1, \ldots, i_k)$-th entry of a tensor $\mathsf{S}$. If certain dimensions are unspecified, then this is denoted by a ":". For example, $A_{i:}$ and $A_{:j}$ denote the $i$-th row and $j$-th column of matrix $A$, respectively. Similarly for slices of tensors.

We consider undirected simple graphs $G = (V_G, E_G, \mathsf{col}_G)$ equipped with a vertex-labelling $\mathsf{col}_G : V_G \to \mathbb{R}^\ell$. We assume that graphs have size $n$, so $V_G$ consists of $n$ vertices and we often identify $V_G$ with $[n]$. For a vertex $v \in V_G$, $N_G(v) := \{u \in V_G \mid vu \in E_G\}$. We let $\mathcal{G}$ be the set of all graphs of size $n$ and let $\mathcal{G}_s$ be the set of pairs $(G, v)$ with $G \in \mathcal{G}$ and $v \in V_G^s$. Note that $\mathcal{G} = \mathcal{G}_0$.

The *color refinement* algorithm (CR) (Morgan, 1965) iteratively computes vertex labellings based on neighboring vertices, as follows. For a graph $G$ and vertex $v \in V_G$, $\mathsf{cr}^{(0)}(G, v) := \mathsf{col}_G(v)$. Then, for $t \geq 0$, $\mathsf{cr}^{(t+1)}(G, v) := (\mathsf{cr}^{(t)}(G, v), \{\{\mathsf{cr}^{(t)}(G, u) \mid u \in N_G(v)\}\})$. We collect all vertex labels to obtain a label for the entire graph by defining $\mathsf{gcr}^{(t)}(G) := \{\{\mathsf{cr}^{(t)}(G, v) \mid v \in V_G\}\}$. The *$k$-dimensional Weisfeiler-Leman* algorithm ($k$-WL) (Cai et al., 1992) iteratively computes labellings of *$k$-tuples of vertices*. For a $k$-tuple $v$, its *atomic type* in $G$, denoted by $\mathsf{atp}_k(G, v)$, is a vector in $\mathbb{R}^{2\binom{k}{2}+k\ell}$. The first $\binom{k}{2}$ entries are $0/1$-values encoding the equality type of $v$, i.e., whether $v_i = v_j$ for $1 \leq i < j \leq k$. The second $\binom{k}{2}$ entries are $0/1$-values encoding adjacency information, i.e., whether $v_i v_j \in E_G$ for $1 \leq i < j \leq k$. The last $k\ell$ real-valued entries correspond to $\mathsf{col}_G(v_i) \in \mathbb{R}^\ell$ for $1 \leq i \leq k$. Initially, for a graph $G$ and $v \in V_G^k$, $k$-WL assigns the label $\mathsf{wl}_k^{(0)}(G, v) := \mathsf{atp}_k(G, v)$. For $t \geq 0$, $k$-WL revises the label according to $\mathsf{wl}_k^{(t+1)}(G, v) := (\mathsf{wl}_k^{(t)}(G, v), M)$ with $M := \{\{(\mathsf{atp}_{k+1}(G, vu), \mathsf{wl}_k^{(t)}(G, v[u/1]), \ldots, \mathsf{wl}_k^{(t)}(G, v[u/k])) \mid u \in V_G\}\}$, where $v[u/i] := (v_1, \ldots, v_{i-1}, u, v_{i+1}, \ldots, v_k)$. We use $k$-WL to assign labels to vertices and graphs by defining: $\mathsf{vwl}_k^{(t)}(G, v) := \mathsf{wl}_k^{(t)}(G, (v, \ldots, v))$, for vertex-labellings, and $\mathsf{gwl}_k^{(t)} := \{\{\mathsf{wl}_k^{(t)}(G, v) \mid v \in V_G^k\}\}$, for graph-labellings. We use $\mathsf{cr}^{(\infty)}$, $\mathsf{gcr}^{(\infty)}$, $\mathsf{vwl}_k^{(\infty)}$, and $\mathsf{gwl}_k^{(\infty)}$ to denote the stable labellings produced by the corresponding algorithm over an arbitrary number of rounds. Our version of 1-WL differs from CR in that 1-WL also uses information from non-adjacent vertices; this distinction only matters for vertex embeddings (Grohe, 2021). We use the "folklore" $k$-WL of Cai et al. (1992), except Cai et al. use 1-WL to refer to CR. While equivalent to "oblivious" $(k + 1)$-WL (Grohe, 2021), used in some other works on GNNs, care is needed when comparing to our work.

Let $G$ be a graph with $V_G = [n]$ and let $\sigma$ be a permutation of $[n]$. We denote by $\sigma \star G$ the isomorphic copy of $G$ obtained by applying the permutation $\sigma$. Similarly, for $v \in V_G^k$, $\sigma \star v$ is the permuted version of $v$. Let $\mathbb{F}$ be some feature space. A function $f : \mathcal{G}_0 \to \mathbb{F}$ is called *invariant* if $f(G) = f(\sigma \star G)$ for any permutation $\pi$. More generally, $f : \mathcal{G}_s \to \mathbb{F}$ is *equivariant* if $f(\sigma \star G, \sigma \star v) = f(G, v)$ for any permutation $\sigma$. The functions $\mathsf{cr}^{(t)} : \mathcal{G}_1 \to \mathbb{F}$ and $\mathsf{vwl}_k^{(t)} : \mathcal{G}_1 \to \mathbb{F}$ are equivariant, whereas $\mathsf{gcr}^{(t)} : \mathcal{G}_0 \to \mathbb{F}$ and $\mathsf{gwl}_k^{(t)} : \mathcal{G}_0 \to \mathbb{F}$ are invariant, for any $t \geq 0$ and $k \geq 1$.

## 3 SPECIFYING GNNS

Many GNNs use linear algebra computations on vectors, matrices or tensors, interleaved with the application of activation functions or MLPs. To understand the separation power of GNNs, we introduce a specification language, TL, for tensor language, that allows us to specify any algebraic computation in a procedural way by explicitly stating how each entry is to be computed. We gauge the separation power of GNNs by specifying them as TL expressions, and syntactically analyzing the components of such TL expressions. This technique gives rise to *Higher-Order Message-Passing Neural Networks* (or $k$-MPNNs), a natural extension of MPNNs (Gilmer et al., 2017). For simplicity, we present TL using summation aggregation only but arbitrary aggregation functions on multisets of real values can be used as well (Section C.5 in the supplementary material).

To introduce TL, consider a typical layer in a GNN of the form $\boldsymbol{F}' = \sigma(\boldsymbol{A} \cdot \boldsymbol{F} \cdot \boldsymbol{W})$, where $\boldsymbol{A} \in \mathbb{R}^{n \times n}$ is an adjacency matrix, $\boldsymbol{F} \in \mathbb{R}^{n \times \ell}$ are vertex features such that $\boldsymbol{F}_{i:} \in \mathbb{R}^{\ell}$ is the feature vector of vertex $i$, $\sigma$ is a non-linear activation function, and $\boldsymbol{W} \in \mathbb{R}^{\ell \times \ell}$ is a weight matrix. By exposing the indices in the matrices and vectors we can equivalently write: for $i \in [n]$ and $s \in [\ell]$:

$$F'_{is} := \sigma\Big(\sum_{j \in [n]} A_{ij} \cdot \big(\sum_{t \in [\ell]} W_{ts} \cdot F_{jt}\big)\Big).$$

In TL, we do not work with specific matrices or indices ranging over $[n]$, but focus instead on expressions applicable to any matrix. We use index variables $x_1$ and $x_2$ instead of $i$ and $j$, replace $A_{ij}$ with a placeholder $E(x_1, x_2)$ and $F_{jt}$ with placeholders $P_t(x_2)$, for $t \in [\ell]$. We then represent the above computation in TL by $\ell$ expressions $\psi_s(x_1)$, one for each feature column, as follows:

$$\psi_s(x_1) = \sigma\Big(\sum_{x_2} E(x_1, x_2) \cdot \big(\sum_{t \in [\ell]} W_{ts} \cdot P_t(x_2)\big)\Big).$$

These are pure syntactical expressions. To give them a semantics, we assign to $E$ a matrix $\boldsymbol{A} \in \mathbb{R}^{n \times n}$, to $P_t$ column vectors $\boldsymbol{F}_{:t} \in \mathbb{R}^{n \times 1}$, for $t \in [\ell]$, and to $x_1$ an index $i \in [n]$. By letting the variable $x_2$ under the summation range over $1, 2, \ldots, n$, the TL expression $\psi_s(i)$ evaluates to $F'_{is}$. As such, $\boldsymbol{F}' = \sigma(\boldsymbol{A} \cdot \boldsymbol{F} \cdot \boldsymbol{W})$ can be represented as a specific instance of the above TL expressions. Throughout the paper we reason about expressions in TL rather than specific instances thereof. Importantly, by showing that certain properties hold for expressions in TL, these properties are inherited by all of its instances. We use TL to enable a theoretical analysis of the separating power of GNNs; It is not intended as a practical programming language for GNNs.

**Syntax.** We first give the syntax of TL expressions. We have a binary predicate $E$, to represent adjacency matrices, and unary vertex predicates $P_s$, $s \in [\ell]$, to represent column vectors encoding the $\ell$-dimensional vertex labels. In addition, we have a (possibly infinite) set $\Omega$ of functions, such as activation functions or MLPs. Then, $\mathsf{TL}(\Omega)$ expressions are defined by the following grammar:

$$\varphi := \mathbf{1}_{x\,\mathsf{op}\,y} \mid E(x, y) \mid P_s(x) \mid \varphi \cdot \varphi \mid \varphi + \varphi \mid a \cdot \varphi \mid f(\varphi, \ldots, \varphi) \mid \sum_x \varphi$$

where $\mathsf{op} \in \{=, \neq\}$, $x, y$ are index variables that specify entries in tensors, $s \in [\ell]$, $a \in \mathbb{R}$, and $f \in \Omega$. Summation aggregation is captured by $\sum_x \varphi$.[1] We sometimes make explicit which functions are used in expressions in $\mathsf{TL}(\Omega)$ by writing $\mathsf{TL}(f_1, f_2, \ldots)$ for $f_1, f_2, \ldots$ in $\Omega$. For example, the expressions $\psi_s(x_1)$ described earlier are in $\mathsf{TL}(\sigma)$.

The set of *free index variables* of an expression $\varphi$, denoted by $\mathsf{free}(\varphi)$, determines the order of the tensor represented by $\varphi$. It is defined inductively: $\mathsf{free}(\mathbf{1}_{x\,\mathsf{op}\,y}) = \mathsf{free}(E(x, y)) := \{x, y\}$, $\mathsf{free}(P_s(x)) = \{x\}$, $\mathsf{free}(\varphi_1 \cdot \varphi_2) = \mathsf{free}(\varphi_1 + \varphi_2) := \mathsf{free}(\varphi_1) \cup \mathsf{free}(\varphi_2)$, $\mathsf{free}(a \cdot \varphi_1) := \mathsf{free}(\varphi_1)$, $\mathsf{free}(f(\varphi_1, \ldots, \varphi_p)) := \cup_{i \in [p]} \mathsf{free}(\varphi_i)$, and $\mathsf{free}(\sum_x \varphi_1) := \mathsf{free}(\varphi_1) \setminus \{x\}$. We sometimes explicitly write the free indices. In our example expressions $\psi_s(x_1)$, $x_1$ is the free index variable.

An important class of expressions are those that only use index variables $\{x_1, \ldots, x_k\}$. We denote by $\mathsf{TL}_k(\Omega)$ the *$k$-index variable fragment* of $\mathsf{TL}(\Omega)$. The expressions $\psi_s(x_1)$ are in $\mathsf{TL}_2(\sigma)$.

**Semantics.** We next define the semantics of expressions in $\mathsf{TL}(\Omega)$. Let $G = (V_G, E_G, \mathsf{col}_G)$ be a vertex-labelled graph. We start by defining the *interpretation* $[\![\cdot, \nu]\!]_G$ of the predicates $E$, $P_s$ and the (dis)equality predicates, relative to $G$ and a *valuation* $\nu$ assigning a vertex to each index variable:

$$[\![E(x, y), \nu]\!]_G := \text{ if } \nu(x)\nu(y) \in E_G \text{ then } 1 \text{ else } 0 \quad [\![P_s(x), \nu]\!]_G := \mathsf{col}_G(\nu(x))_s \in \mathbb{R}$$
$$[\![\mathbf{1}_{x\,\mathsf{op}\,y}, \nu]\!]_G := \text{ if } \nu(x) \mathsf{op} \nu(y) \text{ then } 1 \text{ else } 0.$$

In other words, $E$ is interpreted as the adjacency matrix of $G$ and the $P_s$'s interpret the vertex-labelling $\mathsf{col}_G$. Furthermore, we lift interpretations to arbitrary expressions in $\mathsf{TL}(\Omega)$, as follows:

$$[\![\varphi_1 \cdot \varphi_2, \nu]\!]_G := [\![\varphi_1, \nu]\!]_G \cdot [\![\varphi_2, \nu]\!]_G \qquad [\![\varphi_1 + \varphi_2, \nu]\!]_G := [\![\varphi_1, \nu]\!]_G + [\![\varphi_2, \nu]\!]_G$$
$$[\![\textstyle\sum_x \varphi_1, \nu]\!]_G := \sum_{v \in V_G} [\![\varphi_1, \nu[x \mapsto v]]\!]_G \qquad [\![a \cdot \varphi_1, \nu]\!]_G := a \cdot [\![\varphi_1, \nu]\!]_G$$
$$[\![f(\varphi_1, \ldots, \varphi_p), \nu]\!]_G := f([\![\varphi_1, \nu]\!]_G, \ldots, [\![\varphi_p, \nu]\!]_G)$$

where, $\nu[x \mapsto v]$ is the valuation $\nu$ but which now maps the index $x$ to the vertex $v \in V_G$. For simplicity, we identify valuations with their images. For example, $[\![\varphi(x), v]\!]_G$ denotes $[\![\varphi(x), x \mapsto v]\!]_G$. To illustrate the semantics, for each $v \in V_G$, our example expressions satisfy $[\![\psi_s, v]\!]_G = F'_{vs}$ for $\boldsymbol{F}' = \sigma(\boldsymbol{A} \cdot \boldsymbol{F} \cdot \boldsymbol{W})$ when $\boldsymbol{A}$ is the adjacency matrix of $G$ and $\boldsymbol{F}$ represents the vertex labels.

---

[1]We can replace $\sum_x \varphi$ by a more general aggregation construct $\mathsf{aggr}_x^F(\varphi)$ for arbitrary functions $F$ that assign a real value to multisets of real values. We refer to the supplementary material (Section C.5) for details.

$k$-**MPNNs.** Consider a function $f : \mathcal{G}_s \rightarrow \mathbb{R}^\ell : (G, \boldsymbol{v}) \mapsto f(G, \boldsymbol{v}) \in \mathbb{R}^\ell$ for some $\ell \in \mathbb{N}$. We say that the function $f$ *can be represented in* $\mathsf{TL}(\Omega)$ if there exists $\ell$ expressions $\varphi_1(x_1, \ldots, x_s), \ldots, \varphi_\ell(x_1, \ldots, x_s)$ in $\mathsf{TL}(\Omega)$ such that for each graph $G$ and each $s$-tuple $\boldsymbol{v} \in V_G^s$:

$$f(G, \boldsymbol{v}) = \big( [\![\varphi_1, \boldsymbol{v}]\!]_G, \ldots, [\![\varphi_\ell, \boldsymbol{v}]\!]_G \big).$$

Of particular interest are *kth-order* MPNN*s* (or $k$-MPNNs) which refers to the class of functions that can be represented in $\mathsf{TL}_{k+1}(\Omega)$. We can regard GNNs as functions $f : \mathcal{G}_s \rightarrow \mathbb{R}^\ell$. Hence, a GNN *is a* $k$-MPNN if its corresponding functions are $k$-MPNNs. For example, we can interpret $\boldsymbol{F}' = \sigma(\boldsymbol{A} \cdot \boldsymbol{F} \cdot \boldsymbol{W})$ as a function $f : \mathcal{G}_1 \rightarrow \mathbb{R}^\ell$ such that $f(G, v) := \boldsymbol{F}'_{v:}$. We have seen that for each $s \in [\ell]$, $[\![\psi_s, v]\!]_G = F'_{vs}$ with $\psi_s \in \mathsf{TL}_2(\sigma)$. Hence, $f(G, v) = ([\![\psi_1, v]\!]_G, \ldots, [\![\psi_\ell, v]\!]_G)$ and thus $f$ belongs to 1-MPNNs and our example GNN is a 1-MPNN.

**TL represents equivariant or invariant functions.** We make a simple observation which follows from the type of operators allowed in expressions in $\mathsf{TL}(\Omega)$.

**Proposition 3.1.** Any function $f : \mathcal{G}_s \rightarrow \mathbb{R}^\ell$ represented in $\mathsf{TL}(\Omega)$ is equivariant (invariant if $s = 0$).

An immediate consequence is that when a GNN is a $k$-MPNN, it is automatically invariant or equivariant, depending on whether graph or vertex tuple embeddings are considered.

## 4 SEPARATION POWER OF TENSOR LANGUAGES

Our first main results concern the characterization of the separation power of tensor languages in terms of the color refinement and $k$-dimensional Weisfeiler-Leman algorithms. We provide a fine-grained characterization by taking the number of rounds of these algorithms into account. This will allow for measuring the separation power of classes of GNNs in terms of their number of layers.

### 4.1 SEPARATION POWER

We define the separation power of graph functions in terms of an equivalence relation, based on the definition from Azizian & Lelarge (2021), hereby first focusing on their ability to separate vertices.[2]

**Definition 1.** Let $\mathcal{F}$ be a set of functions $f : \mathcal{G}_1 \rightarrow \mathbb{R}^{\ell_f}$. The equivalence relation $\rho_1(\mathcal{F})$ is defined by $\mathcal{F}$ on $\mathcal{G}_1$ as follows: $\big( (G, v), (H, w) \big) \in \rho_1(\mathcal{F}) \iff \forall f \in \mathcal{F}, f(G, v) = f(H, w)$. $\qquad \square$

In other words, when $\big( (G, v), (H, w) \big) \in \rho_1(\mathcal{F})$, no function in $\mathcal{F}$ can separate $v$ in $G$ from $w$ in $H$. For example, we can view $\mathsf{cr}^{(t)}$ and $\mathsf{vwl}_k^{(t)}$ as functions from $\mathcal{G}_1$ to some $\mathbb{R}^\ell$. As such $\rho_1(\mathsf{cr}^{(t)})$ and $\rho_1(\mathsf{vwl}_k^{(t)})$ measure the separation power of these algorithms. The following strict inclusions are known: for all $k \geq 1$, $\rho_1(\mathsf{vwl}_{k+1}^{(t)}) \subset \rho_1(\mathsf{vwl}_k^{(t)})$ and $\rho_1(\mathsf{vwl}_1^{(t)}) \subset \rho_1(\mathsf{cr}^{(t)})$ (Otto, 2017; Grohe, 2021). It is also known that more rounds ($t$) increase the separation power of these algorithms (Fürer, 2001).

For a fragment $\mathcal{L}$ of $\mathsf{TL}(\Omega)$ expressions, we define $\rho_1(\mathcal{L})$ as the equivalence relation associated with all functions $f : \mathcal{G}_1 \rightarrow \mathbb{R}^{\ell_f}$ that can be represented in $\mathcal{L}$. By definition, we here thus consider expressions in $\mathsf{TL}(\Omega)$ with one free index variable resulting in vertex embeddings.

### 4.2 MAIN RESULTS

We first provide a link between $k$-WL and tensor language expressions using $k+1$ index variables:

**Theorem 4.1.** *For each $k \geq 1$ and any collection $\Omega$ of functions, $\rho_1\big( \mathsf{vwl}_k^{(\infty)} \big) = \rho_1\big( \mathsf{TL}_{k+1}(\Omega) \big)$.*

This theorem gives us new insights: if we wish to understand how a new GNN architecture compares against the $k$-WL algorithms, all we need to do is to show that such an architecture can be represented in $\mathsf{TL}_{k+1}(\Omega)$, i.e., is a $k$-MPNN, an arguably much easier endeavor. As an example of how to use this result, it is well known that triangles can be detected by 2-WL but not by 1-WL. Thus, in order to design GNNs that can detect triangles, layer definitions in $\mathsf{TL}_3$ rather than $\mathsf{TL}_2$ should be used.

We can do much more, relating the rounds of $k$-WL to the notion of summation depth of $\mathsf{TL}(\Omega)$ expressions. We also present present similar results for functions computing graph embeddings.

---

[2]We differ slightly from Azizian & Lelarge (2021) in that they only define equivalence relations on graphs.

The *summation depth* $\mathsf{sd}(\varphi)$ of a $\mathsf{TL}(\Omega)$ expression $\varphi$ measures the nesting depth of the summations $\sum_x$ in the expression. It is defined inductively: $\mathsf{sd}(\mathbf{1}_{x\,\mathsf{op}\,y}) = \mathsf{sd}(E(x,y)) = \mathsf{sd}(P_s(x)) := 0$, $\mathsf{sd}(\varphi_1 \cdot \varphi_2) = \mathsf{sd}(\varphi_1 + \varphi_2) := \mathsf{max}\{\mathsf{sd}(\varphi_1), \mathsf{sd}(\varphi_2)\}$, $\mathsf{sd}(a \cdot \varphi_1) := \mathsf{sd}(\varphi_1)$, $\mathsf{sd}(f(\varphi_1, \ldots, \varphi_p)) := \mathsf{max}\{\mathsf{sd}(\varphi_i)|i \in [p]\}$, and $\mathsf{sd}(\sum_x \varphi_1) := \mathsf{sd}(\varphi_1) + 1$. For example, expressions $\psi_s(x_1)$ above have summation depth one. We write $\mathsf{TL}_{k+1}^{(t)}(\Omega)$ for the class of expressions in $\mathsf{TL}_{k+1}(\Omega)$ of summation depth at most $t$, and use $k\text{-MPNN}^{(t)}$ for the corresponding class of $k$-MPNNs. We can now refine Theorem 4.1, taking into account the number of rounds used in $k$-WL.

**Theorem 4.2.** *For all $t \geq 0$, $k \geq 1$ and any collection $\Omega$ of functions,* $\rho_1(\mathsf{vwl}_k^{(t)}) = \rho_1(\mathsf{TL}_{k+1}^{(t)}(\Omega))$.

**Guarded TL and color refinement.** As noted by Barceló et al. (2020), the separation power of vertex embeddings of simple GNNs, which propagate information only through neighboring vertices, is usually weaker than that of 1-WL. For these types of architectures, Barceló et al. (2020) provide a relation with the weaker color refinement algorithm, but only in the special case of *first-order* classifiers. We can recover and extend this result in our general setting, with a *guarded* version of TL which, as we will show, has the same separation power as color refinement.

The *guarded fragment* $\mathsf{GTL}(\Omega)$ of $\mathsf{TL}_2(\Omega)$ is inspired by the use of adjacency matrices in simple GNNs. In $\mathsf{GTL}(\Omega)$ only equality predicates $\mathbf{1}_{x_i=x_i}$ (constant 1) and $\mathbf{1}_{x_i \neq x_i}$ (constant 0) are allowed, addition and multiplication require the component expressions to have the same (single) free index, and summation must occur in a *guarded form* $\sum_{x_j}(E(x_i, x_j) \cdot \varphi(x_j))$, for $i, j \in [2]$. Guardedness means that summation only happens over neighbors. In $\mathsf{GTL}(\Omega)$, all expressions have a single free variable and thus only functions from $\mathcal{G}_1$ can be represented. Our example expressions $\psi_s(x_1)$ are guarded. The fragment $\mathsf{GTL}^{(t)}(\Omega)$ consists of expressions in $\mathsf{GTL}(\Omega)$ of summation depth at most $t$. We denote by MPNNs and MPNNs$^{(t)}$ the corresponding "guarded" classes of 1-MPNNs.[3]

**Theorem 4.3.** *For all $t \geq 0$ and any collection $\Omega$ of functions:* $\rho_1(\mathsf{cr}^{(t)}) = \rho_1(\mathsf{GTL}^{(t)}(\Omega))$.

As an application of this theorem, to detect the existence of paths of length $t$, the number of guarded layers in GNNs should account for a representation in $\mathsf{GTL}(\Omega)$ of summation depth of at least $t$. We recall that $\rho_1(\mathsf{vwl}_1^{(t)}) \subset \rho_1(\mathsf{cr}^{(t)})$ which, combined with our previous results, implies that $\mathsf{TL}_2^{(t)}(\Omega)$ (resp., 1-MPNNs) is strictly more separating than $\mathsf{GTL}^{(t)}(\Omega)$ (resp., MPNNs).

**Graph embeddings.** We next establish connections between the graph versions of $k$-WL and CR, and TL expressions without free index variables. To this aim, we use $\rho_0(\mathcal{F})$, for a set $\mathcal{F}$ of functions $f : \mathcal{G} \rightarrow \mathbb{R}^{\ell_f}$, as the equivalence relation over $\mathcal{G}$ defined in analogy to $\rho_1$: $(G, H) \in \rho_0(\mathcal{F}) \iff \forall f \in \mathcal{F}, f(G) = f(H)$. We thus consider separation power on the graph level. For example, we can consider $\rho_0(\mathsf{gcr}^{(t)})$ and $\rho_0(\mathsf{gwl}_k^{(t)})$ for any $t \geq 0$ and $k \geq 1$. Also here, $\rho_0(\mathsf{gwl}_{k+1}^{(t)}) \subset \rho_0(\mathsf{gwl}_k^{(t)})$ but different from vertex embeddings, $\rho_0(\mathsf{gcr}^{(t)}) = \rho_0(\mathsf{gwl}_1^{(t)})$ (Grohe, 2021). We define $\rho_0(\mathcal{L})$ for a fragment $\mathcal{L}$ of $\mathsf{TL}(\Omega)$ by considering expressions without free index variables.

The connection between the number of index variables in expressions and $k$-WL remains to hold. Apart from $k = 1$, no clean relationship exists between summation depth and rounds, however.[4]

**Theorem 4.4.** *For all $t \geq 0$, $k \geq 1$ and any collection $\Omega$ of functions, we have that:*

*(1)* $\rho_0(\mathsf{gcr}^{(t)}) = \rho_0(\mathsf{TL}_2^{(t+1)}(\Omega)) = \rho_0(\mathsf{gwl}_1^{(t)})$ *(2)* $\rho_0(\mathsf{gwl}_k^{(\infty)}) = \rho_0(\mathsf{TL}_{k+1}(\Omega))$.

Intuitively, in (1) the increase in summation depth by one is incurred by the additional aggregation needed to collect all vertex labels computed by $\mathsf{gwl}_1^{(t)}$.

**Optimality of number of indices.** Our results so far tell that graph functions represented in $\mathsf{TL}_{k+1}(\Omega)$ are at most as separating as $k$-WL. What is left unaddressed is whether all $k + 1$ index variables are needed for the graph functions under consideration. It may well be, for example, that there exists an equivalent expression using less index variables. This would imply a stronger upper bound on the separation power by $\ell$-WL for $\ell < k$. We next identify a large class of $\mathsf{TL}(\Omega)$ expressions, those of *treewidth* $k$, for which the number of index variables can be reduced to $k + 1$.

---

[3]For the connection to classical MPNNs (Gilmer et al., 2017), see Section H in the supplementary material.

[4]Indeed, the best one can obtain for general tensor logic expressions is $\rho_0(\mathsf{TL}_{k+1}^{(t+k)}(\Omega)) \subseteq \rho_0(\mathsf{gwl}_k^{(t)}) \subseteq \rho_0(\mathsf{TL}_{k+1}^{(t+1)}(\Omega))$. This follows from Cai et al. (1992) and connections to finite variable logics.

**Proposition 4.5.** *Expressions in* $\mathsf{TL}(\Omega)$ *of treewidth* $k$ *are equivalent to expressions in* $\mathsf{TL}_{k+1}(\Omega)$.

Treewidth is defined in the supplementary material (Section G) and a treewidth of $k$ implies that the computation of tensor language expressions can be decomposed, by reordering summations, such that each local computation requires at most $k + 1$ indices (see also Aji & McEliece (2000)). As a simple example, consider $\theta(x_1) = \sum_{x_2} \sum_{x_3} E(x_1, x_2) \cdot E(x_2, x_3)$ in $\mathsf{TL}_3^{(2)}$ such that $[\![\theta, v]\!]_G$ counts the number of paths of length two starting from $v$. This expression has a treewidth of one. And indeed, it is equivalent to the expression $\tilde{\theta}(x_1) = \sum_{x_2} E(x_1, x_2) \cdot \left( \sum_{x_1} E(x_2, x_1) \right)$ in $\mathsf{TL}_2^{(2)}$ (and in fact in $\mathsf{GTL}^{(2)}$). As a consequence, no more vertices can be separated by $\theta(x_1)$ than by $\mathsf{cr}^{(2)}$, rather than $\mathsf{vwl}_2^2$ as the original expression in $\mathsf{TL}_3^{(2)}$ suggests.

**On the impact of functions.** All separation results for $\mathsf{TL}(\Omega)$ and fragments thereof hold irregardless of the chosen functions in $\Omega$, including when no functions are present at all. Function applications hence do not add expressive power. While this may seem counter-intuitive, it is due to the presence of summation and multiplication in $\mathsf{TL}$ that are enough to separate graphs or vertices.

## 5 Consequences for GNNs

We next interpret the general results on the separation power from Section 4 in the context of GNNs.

*1. The separation power of any vertex embedding* GNN *architecture which is an* $\mathsf{MPNN}^{(t)}$ *is bounded by the power of* $t$ *rounds of color refinement.*

We consider the Graph Isomorphism Networks (GINs) (Xu et al., 2019) and show that these are MPNNs. To do so, we represent them in $\mathsf{GTL}(\Omega)$. Let gin be such a network; it updates vertex embeddings as follows. Initially, $\mathsf{gin}^{(0)} : \mathcal{G}_1 \to \mathbb{R}^{\ell_0} : (G, v) \mapsto \boldsymbol{F}_{v:}^{(0)} := \mathsf{col}_G(v) \in \mathbb{R}^{\ell_0}$. For layer $t > 0$, $\mathsf{gin}^{(t)} : \mathcal{G}_1 \to \mathbb{R}^{\ell_t}$ is given by: $(G, v) \mapsto \boldsymbol{F}_{v:}^{(t)} := \mathsf{mlp}^{(t)}\big(\boldsymbol{F}_{v:}^{(t-1)}, \sum_{u \in N_G(v)} \boldsymbol{F}_{u:}^{(t-1)}\big)$, with $\boldsymbol{F}^{(t)} \in \mathbb{R}^{n \times \ell_t}$ and $\mathsf{mlp}^{(t)} = (\mathsf{mlp}_1^{(t)}, \ldots, \mathsf{mlp}_{\ell_t}^{(t)}) : \mathbb{R}^{2\ell_{t-1}} \to \mathbb{R}^{\ell_t}$ is an MLP. We denote by $\mathsf{GIN}^{(t)}$ the class of GINs consisting $t$ layers. Clearly, $\mathsf{gin}^{(0)}$ can be represented in $\mathsf{GTL}^{(0)}$ by considering the expressions $\varphi_i^{(0)}(x_1) := P_i(x_1)$ for each $i \in [\ell_0]$. To represent $\mathsf{gin}^{(t)}$, assume that we have $\ell_{t-1}$ expressions $\varphi_i^{(t-1)}(x_1)$ in $\mathsf{GTL}^{(t-1)}(\Omega)$ representing $\mathsf{gin}^{(t-1)}$. That is, we have $[\![\varphi_i^{(t-1)}, v]\!]_G = F_{vi}^{(t-1)}$ for each vertex $v$ and $i \in [\ell_{t-1}]$. Then $\mathsf{gin}^{(t)}$ is represented by $\ell_t$ expressions $\varphi_i^{(t)}(x_1)$ defined as:

$$\mathsf{mlp}_i^{(t)}\Big( \varphi_1^{(t-1)}(x_1), \ldots, \varphi_{\ell_{t-1}}^{(t-1)}(x_1), \sum_{x_2} E(x_1, x_2) \cdot \varphi_1^{(t-1)}(x_2), \ldots, \sum_{x_2} E(x_1, x_2) \cdot \varphi_{\ell_{t-1}}^{(t-1)}(x_2) \Big),$$

which are now expressions in $\mathsf{GTL}^{(t)}(\Omega)$ where $\Omega$ consists of MLPs. We have $[\![\varphi_i^{(t)}, v]\!]_G = F_{v,i}^{(t)}$ for each $v \in V_G$ and $i \in [\ell_t]$, as desired. Hence, Theorem 4.3 tells that $t$-layered GINs cannot be more separating than $t$ rounds of color refinement, in accordance with known results (Xu et al., 2019; Morris et al., 2019). We thus simply cast GINs in $\mathsf{GTL}(\Omega)$ to obtain an upper bound on their separation power. In the supplementary material (Section D) we give similar analyses for GraphSage GNNs with various aggregation functions (Hamilton et al., 2017), GCNs (Kipf & Welling, 2017), simplified GCNs (SGCs) (Wu et al., 2019), Principled Neighborbood Aggregation (PNAs) (Corso et al., 2020), and revisit the analysis of ChebNet (Defferrard et al., 2016) given in Balcilar et al. (2021a).

*2. The separation power of any vertex embedding* GNN *architecture which is an* $k$*-*$\mathsf{MPNN}^{(t)}$ *is bounded by the power of* $t$ *rounds of* $k$*-*$\mathsf{WL}$.

For $k = 1$, we consider extended Graph Isomorphism Networks (eGINs) (Barceló et al., 2020). For an $\mathsf{egin} \in \mathsf{eGIN}$, $\mathsf{egin}^{(0)} : \mathcal{G}_1 \to \mathbb{R}^{\ell_0}$ is defined as for GINs, but for layer $t > 0$, $\mathsf{egin}^{(t)} : \mathcal{G}_1 \to \mathbb{R}^{\ell_t}$ is defined by $(G, v) \mapsto \boldsymbol{F}_{v:}^{(t)} := \mathsf{mlp}^{(t)}\big(\boldsymbol{F}_{v:}^{(t-1)}, \sum_{u \in N_G(v)} \boldsymbol{F}_{u:}^{(t-1)}, \sum_{u \in V_G} \boldsymbol{F}_{u:}^{(t-1)}\big)$, where $\mathsf{mlp}^{(t)}$ is now an MLP from $\mathbb{R}^{3\ell_{t-1}} \to \mathbb{R}^{\ell_t}$. The difference with GINs is the use of $\sum_{u \in V_G} \boldsymbol{F}_{u:}^{(t-1)}$ which corresponds to the *unguarded* summation $\sum_{x_1} \varphi^{(t-1)}(x_1)$. This implies that $\mathsf{TL}$ rather than $\mathsf{GTL}$ needs to be used. In a similar way as for GINs, we can represent eGIN layers in $\mathsf{TL}_2^{(t)}(\Omega)$. That is, each $\mathsf{eGIN}^{(t)}$ is an $1$-$\mathsf{MPNN}^{(t)}$. Theorem 4.2 tells that $t$ rounds of 1-WL bound the separation power of $t$-layered extended GINs, conform to Barceló et al. (2020). More generally, any GNN looking to go beyond CR must use non-guarded aggregations.

For $k \geq 2$, it is straightforward to show that $t$-layered "folklore" GNNs ($k$-FGNNs) (Maron et al., 2019b) are $k$-$\mathsf{MPNN}^{(t)}$ and thus, by Theorem 4.2, $t$ rounds of $k$-WL bound their separation power.

One merely needs to cast the layer definitions in $\mathsf{TL}(\Omega)$ and observe that $k+1$ indices and summation depth $t$ are needed. We thus refine and recover the $k$-WL bound for $k$-FGNNs by Azizian & Lelarge (2021). We also show that the separation power of $(k+1)$-Invariant Graph Networks ($(k+1)$-IGNs) (Maron et al., 2019b) are bounded by $k$-WL, albeit with an increase in the required rounds.

**Theorem 5.1.** *For any $k \geq 1$, the separation power of a $t$-layered $(k+1)$-IGNs is bounded by the separation power of $tk$ rounds of $k$-WL.*

We hereby answer open problem 1 in Maron et al. (2019a). The case $k = 1$ was solved in Chen et al. (2020) by analyzing properties of 1-WL. By contrast, Theorem 4.2 shows that one can focus on expressing $(k+1)$-IGNs in $\mathsf{TL}_{k+1}(\Omega)$ and analyzing the summation depth of expressions. The proof of Theorem 5.1 requires non-trivial manipulations of tensor language expressions; it is a simplified proof of Geerts (2020). The additional rounds ($tk$) are needed because $(k+1)$-IGNs aggregate information in one layer that becomes accessible to $k$-WL in $k$ rounds. We defer detail to Section E in the supplementary material, where we also identify a simple class of $t$-layered $(k+1)$-IGNs that are as powerful as $(k+1)$-IGNs but whose separation power is bounded by $t$ rounds of $k$-WL.

We also consider "augmented" GNNs, which are combined with a preprocessing step in which higher-order graph information is computed. In the supplementary material (Section D.3) we show how TL encodes the preprocessing step, and how this leads to separation bounds in terms of $k$-WL, where $k$ depends on the treewidth of the graph information used. Finally, our approach can also be used to show that the spectral CayleyNets (Levie et al., 2019) are bounded in separation power by 2-WL. This result complements the spectral analysis of CayleyNets given in Balcilar et al. (2021b).

*3. The separation power of any graph embedding GNN architecture which is a $k$-MPNN is bounded by the power of $k$-WL.*

Graph embedding methods are commonly obtained from vertex (tuple) embeddings methods by including a readout layer in which all vertex (tuple) embeddings are aggregated. For example, $\mathsf{mlp}(\sum_{v \in V} \mathsf{egin}^{(t)}(G, v))$ is a typical readout layer for eGINs . Since $\mathsf{egin}^{(t)}$ can be represented in $\mathsf{TL}_2^{(t)}(\Omega)$, the readout layer can be represented in $\mathsf{TL}_2^{(t+1)}(\Omega)$, using an extra summation. So they are 1-MPNNs. Hence, their separation power is bounded by $\mathsf{gwl}_1^{(t)}$, in accordance with Theorem 4.4. This holds more generally. If vertex embedding methods are $k$-MPNNs, then so are their graph versions, which are then bounded by $\mathsf{gwl}_k^{(\infty)}$ by our Theorem 4.4.

*4. To go beyond the separation power of $k$-WL, it is necessary to use GNNs whose layers are represented by expressions of treewidth $> k$.*

Hence, to design expressive GNNs one needs to define the layers such that treewidth of the resulting TL expressions is large enough. For example, to go beyond 1-WL, $\mathsf{TL}_3$ representable linear algebra operations should be used. Treewidth also sheds light on the open problem from Maron et al. (2019a) where it was asked whether polynomial layers (in $\boldsymbol{A}$) increase the separation power. Indeed, consider a layer of the form $\sigma(\boldsymbol{A}^3 \cdot \boldsymbol{F} \cdot \boldsymbol{W})$, which raises the adjacency matrix $\boldsymbol{A}$ to the power three. Translated in $\mathsf{TL}(\Omega)$, layer expressions resemble $\sum_{x_2} \sum_{x_3} \sum_{x_4} E(x_1, x_2) \cdot E(x_2, x_3) \cdot E(x_3, x_4)$, of treewidth one. Proposition 4.5 tells that the layer is bounded by $\mathsf{wl}_1^{(3)}$ (and in fact by $\mathsf{cr}^{(3)}$) in separation power. If instead, the layer is of the form $\sigma(\boldsymbol{C} \cdot \boldsymbol{F} \cdot \boldsymbol{W})$ where $C_{ij}$ holds the number of cliques containing the edge $ij$. Then, in $\mathsf{TL}(\Omega)$ we get expressions containing $\sum_{x_2} \sum_{x_3} E(x_1, x_2) \cdot E(x_1, x_3) \cdot E(x_2, x_3)$. The variables form a 3-clique resulting in expressions of treewidth two. As a consequence, the separation power will be bounded by $\mathsf{wl}_2^{(2)}$. These examples show that it is not the number of multiplications (in both cases two) that gives power, it is how variables are connected to each other.

## 6 FUNCTION APPROXIMATION

We next provide characterizations of functions that can be approximated by TL expressions, when interpreted as functions. We recover and extend results from Azizian & Lelarge (2021) by taking the number of layers of GNNs into account. We also provide new results related to color refinement.

### 6.1 GENERAL TL APPROXIMATION RESULTS

We assume that $\mathcal{G}_s$ is a compact space by requiring that vertex labels come from a compact set $K \subseteq \mathbb{R}^{\ell_0}$. Let $\mathcal{F}$ be a set of functions $f : \mathcal{G}_s \to \mathbb{R}^{\ell_f}$ and define its closure $\overline{\mathcal{F}}$ as all functions $h$ from $\mathcal{G}_s$ for

which there exists a sequence $f_1, f_2, \ldots \in \mathcal{F}$ such that $\lim_{i \to \infty} \sup_{G, \boldsymbol{v}} \| f_i(G, \boldsymbol{v}) - h(G, \boldsymbol{v}) \| = 0$ for some norm $\|.\|$. We assume $\mathcal{F}$ to satisfy two properties. First, $\mathcal{F}$ is *concatenation-closed*: if $f_1 : \mathcal{G}_s \to \mathbb{R}^p$ and $f_2 : \mathcal{G}_s \to \mathbb{R}^q$ are in $\mathcal{F}$, then $g := (f_1, f_2) : \mathcal{G}_s \to \mathbb{R}^{p+q} : (G, \boldsymbol{v}) \mapsto (f_1(G, \boldsymbol{v}), f_2(G, \boldsymbol{v}))$ is also in $\mathcal{F}$. Second, $\mathcal{F}$ is *function-closed*, for a fixed $\ell \in \mathbb{N}$: for any $f \in \mathcal{F}$ such that $f : \mathcal{G}_s \to \mathbb{R}^p$, also $g \circ f : \mathcal{G}_s \to \mathbb{R}^\ell$ is in $\mathcal{F}$ for any continuous function $g : \mathbb{R}^p \to \mathbb{R}^\ell$. For such $\mathcal{F}$, we let $\mathcal{F}_\ell$ be the subset of functions in $\mathcal{F}$ from $\mathcal{G}_s$ to $\mathbb{R}^\ell$. Our next result is based on a generalized Stone-Weierstrass Theorem (Timofte, 2005), also used in Azizian & Lelarge (2021).

**Theorem 6.1.** *For any $\ell$, and any set $\mathcal{F}$ of functions, concatenation and function closed for $\ell$, we have:* $\overline{\mathcal{F}_\ell} = \{ f : \mathcal{G}_s \to \mathbb{R}^\ell \mid \rho_s(\mathcal{F}) \subseteq \rho_s(f) \}$.

This result gives us insight on which functions can be approximated by, for example, a set $\mathcal{F}$ of functions originating from a class of GNNs. In this case, $\overline{\mathcal{F}_\ell}$ represent all functions approximated by instances of such a class and Theorem 6.1 tells us that this set corresponds precisely to the set of all functions that are equally or less separating than the GNNs in this class. If, in addition, $\mathcal{F}_\ell$ is more separating that CR or $k$-WL, then we can say more. Let $\mathsf{alg} \in \{ \mathsf{cr}^{(t)}, \mathsf{gcr}^{(t)}, \mathsf{vwl}_k^{(t)}, \mathsf{gwl}_k^{(\infty)} \}$.

**Corollary 6.2.** *Under the assumptions of Theorem 6.1 and if $\rho(\mathcal{F}_\ell) = \rho(\mathsf{alg})$, then* $\overline{\mathcal{F}_\ell} = \{ f : \mathcal{G}_s \to \mathbb{R}^\ell \mid \rho(\mathsf{alg}) \subseteq \rho(f) \}$.

The properties of being concatenation and function-closed are satisfied for sets of functions representable in our tensor languages, if $\Omega$ contains all continuous functions $g : \mathbb{R}^p \to \mathbb{R}^\ell$, for any $p$, or alternatively, all MLPs (by Lemma 32 in Azizian & Lelarge (2021)). Together with our results in Section 4, the corollary implies that MPNNs$^{(t)}$, 1-MPNNs$^{(t)}$, $k$-MPNNs$^{(t)}$ or $k$-MPNNs can approximate all functions with equal or less separation power than $\mathsf{cr}^{(t)}$, $\mathsf{gcr}^{(t)}$, $\mathsf{vwl}_k^{(t)}$ or $\mathsf{gwl}_k^{(\infty)}$, respectively.

Prop. 3.1 also tells that the closure consists of invariant ($s = 0$) and equivariant ($s > 0$) functions.

## 6.2 CONSEQUENCES FOR GNNs

***All our results combined provide a recipe to guarantee that a given function can be approximated by*** GNN ***architectures.*** Indeed, suppose that your class of GNNs is an MPNN$^{(t)}$ (respectively, 1-MPNN$^{(t)}$, $k$-MPNN$^{(t)}$ or $k$-MPNN, for some $k \geq 1$). Then, since most classes of GNNs are concatenation-closed and allow the application of arbitrary MLPs, this implies that your GNNs can only approximate functions $f$ that are no more separating than $\mathsf{cr}^{(t)}$ (respectively, $\mathsf{gcr}^{(t)}$, $\mathsf{vwl}_k^{(t)}$ or $\mathsf{gwl}_k^{(\infty)}$). To guarantee that that these functions can indeed be approximated, one additionally has to show that your class of GNNs matches the corresponding labeling algorithm in separation power.

For example, GNNs in $\mathsf{GIN}_\ell^{(t)}$ are MPNNs$^{(t)}$, and thus $\overline{\mathsf{GIN}_\ell^{(t)}}$ contains any function $f : \mathcal{G}_1 \to \mathbb{R}^\ell$ satisfying $\rho_1(\mathsf{cr}^{(t)}) \subseteq \rho_1(f)$. Similarly, $\mathsf{eGIN}_\ell^{(t)}$'s are 1-MPNNs$^{(t)}$, so $\overline{\mathsf{eGIN}_\ell^{(t)}}$ contains any function satisfying $\rho_1(\mathsf{wl}_1^{(t)}) \subseteq \rho_1(f)$; and when extended with a readout layer, their closures consist of functions $f : \mathcal{G}_0 \to \mathbb{R}^\ell$ satisfying $\rho_0(\mathsf{gcr}^{(t)}) = \rho_0(\mathsf{vwl}_1^{(t)}) \subseteq \rho_0(f)$. Finally, $k$-$\mathsf{FGNN}_\ell^{(t)}$'s are $k$-MPNNs$^{(t)}$, so $\overline{k\text{-}\mathsf{FGNN}_\ell^{(t)}}$ consist of functions $f$ such that $\rho_1(\mathsf{vwl}_k^{(t)}) \subseteq \rho_1(f)$. We thus recover and extend results by Azizian & Lelarge (2021) by including layer information ($t$) and by treating color refinement separately from 1-WL for vertex embeddings. Furthermore, Theorem 5.1 implies that $\overline{(k+1)\text{-}\mathsf{IGN}_\ell}$ consists of functions $f$ satisfying $\rho_1(\mathsf{vwl}_k^{(\infty)}) \subseteq \rho_1(f)$ and $\rho_0(\mathsf{gwl}_k^{(\infty)}) \subseteq \rho_0(f)$, a case left open in Azizian & Lelarge (2021).

These results follow from Corollary 6.2, that the respective classes of GNNs can simulate CR or $k$-WL on either graphs with discrete (Xu et al., 2019; Barceló et al., 2020) or continuous labels (Maron et al., 2019b), and that they are $k$-MPNNs of the appropriate form.

## 7 CONCLUSION

Connecting GNNs and tensor languages allows us to use our analysis of tensor languages to understand the separation and approximation power of GNNs. The number of indices and summation depth needed to represent the layers in GNNs determine their separation power in terms of color refinement and Weisfeiler-Leman tests. The framework of $k$-MPNNs provides a handy toolbox to understand existing and new GNN architectures, and we demonstrate this by recovering several results about the power of GNNs presented recently in the literature, as well as proving new results.

## 8 AKNOWLEDGEMENTS & DISCLOSURE FUNDING

This work is partially funded by ANID–Millennium Science Initiative Program–Code ICN17_002, Chile.

## ETHICS STATEMENT

The results in this paper do not include misleading claims; their correctness is theoretically verified. Related work is accurately represented.

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

SUPPLEMENTARY MATERIAL

## A  RELATED WORK CNT'D

We provide additional details on how the tensor language $\mathsf{TL}(\Omega)$ considered in this paper relates to recent work on other matrix query languages. Closest to $\mathsf{TL}(\Omega)$ is the matrix query language sum-MATLANG (Geerts et al., 2021b) whose syntax is close to that of $\mathsf{TL}(\Omega)$. There are, however, key differences. First, although sum-MATLANG uses index variables (called vector variables), they all must occur under a summation. In other words, the concept of free index variables is missing, which implies that no general tensors can be represented. In $\mathsf{TL}(\Omega)$, we can represent arbitrary tensors and the presence of free index variables is crucial to define vertex, or more generally, $k$-tuple embeddings in the context of GNNs. Furthermore, no notion of summation depth was introduced for sum-MATLANG. In $\mathsf{TL}(\Omega)$, the summation depth is crucial to assess the separation power in terms of the number of rounds of color refinement and $k$-WL. And in fact, the separation power of sum-MATLANG was not considered before, and neither are finite variable fragments of sum-MATLANG and connections to color refinement and $k$-WL studied before. Finally, no other aggregation functions were considered for sum-MATLANG. We detail in Section C.5 that $\mathsf{TL}(\Omega)$ can be gracefully extended to $\mathsf{TL}(\Omega, \Theta)$ for some arbitrary set $\Theta$ of aggregation functions.

Connections to 1-WL and 2-WL and the separation power of another matrix query language, MATLANG (Brijder et al., 2019) were established in Geerts (2021). Yet, the design of MATLANG is completely different in spirit than that of $\mathsf{TL}(\Omega)$. Indeed, MATLANG does not have index variables or explicit summation aggregation. Instead, it only supports matrix multiplication, matrix transposition, function applications, and turning a vector into a diagonal matrix. As such, MATLANG can be shown to be included in $\mathsf{TL}_3(\Omega)$. Similarly as for sum-MATLANG, MATLANG cannot represent general tensors, has no (free) index variables and summation depth is not considered (in view of the absence of an explicit summation).

We also emphasize that neither for MATLANG nor for sum-MATLANG a guarded fragment was considered. The guarded fragment is crucial to make connections to color refinement (Theorem 4.3). Furthermore, the analysis in terms of the number of index variables, summation depth and treewidth (Theorems 4.1,4.2 and Proposition 4.5), were not considered before in the matrix query language literature. For none of these matrix query languages, approximation results were considered (Section 6.1).

Matrix query languages are used to assess the expressive power of linear algebra. Balcilar et al. (2021a) use MATLANG and the above mentioned connections to 1-WL and 2-WL, to assess the separation power of GNNs. More specifically, similar to our work, they show that several GNN architectures can be represented in MATLANG, or fragments thereof. As a consequence, bounds on their separation power easily follow. Furthermore, Balcilar et al. (2021a) propose new architectures inspired by special operators in MATLANG. The use of $\mathsf{TL}(\Omega)$ can thus been seen as a continuation of their approach. We note, however, that $\mathsf{TL}(\Omega)$ is more general than MATLANG (which is included in $\mathsf{TL}_3(\Omega)$), allows to represent more complex linear algebra computations by means summation (or other) aggregation, and finally, provides insights in the number of iterations needed for color refinement and $k$-WL. The connection between the number of variables (or treewidth) and $k$-WL is not present in the work by Balcilar et al. (2021a), neither is the notion of guarded fragment, needed to connect to color refinement. We believe that it is precisely these latter two insights that make the tensor language approach valuable for any GNN designer who wishes to upper bound their GNN architecture.

## B  DETAILS OF SECTION 3

### B.1  PROOF OF PROPOSITION 3.1

Let $G = (V, E, \mathsf{col})$ be a graph and let $\sigma$ be a permutation of $V$. As usual, we define $\sigma \star G = (V^\sigma, E^\sigma, \mathsf{col}^\sigma)$ as the graph with vertex set $V^\sigma := V$, edge set $vw \in E^\sigma$ if and only if $\sigma^{-1}(v)\sigma^{-1}(w) \in E$, and $\mathsf{col}^\sigma(v) := \mathsf{col}(\sigma^{-1}(v))$. We need to show that for any expression $\varphi(\boldsymbol{x})$ in $\mathsf{TL}(\Omega)$ either $[\![\varphi, \sigma \star \boldsymbol{v}]\!]_{\sigma \star G} = [\![\varphi, \boldsymbol{v}]\!]_G$, or when $\varphi$ has no free index variables, $[\![\varphi]\!]_{\sigma \star G} = [\![\varphi]\!]_G$. We verify this by a simple induction on the structure of expressions in $\mathsf{TL}(\Omega)$.

- If $\varphi(x_i, x_j) = \mathbf{1}_{x_i \text{ op } x_j}$, then for a valuation $\nu$ mapping $x_i$ to $v_i$ and $x_j$ to $v_j$ in $V$:

$$[\![\mathbf{1}_{x_i \text{ op } x_j}, \nu]\!]_G = \mathbf{1}_{v_i \text{ op } v_j} = \mathbf{1}_{\sigma(v_i) \text{ op } \sigma(v_j)} = [\![\mathbf{1}_{x_i \text{ op } x_j}, \sigma \star \nu]\!]_{\sigma \star G},$$

where we used that $\sigma$ is a permutation.

- If $\varphi(x_i) = P_\ell(x_i)$, then for a valuation $\mu$ mapping $x_i$ to $v_i$ in $V$:

$$[\![P_\ell, \mu]\!]_G = (\mathsf{col}(v_i))_\ell = (\mathsf{col}^\sigma(\sigma(v_i)))_\ell = [\![P_\ell, \sigma \star \nu]\!]_{\sigma \star G},$$

where we used the definition of $\mathsf{col}^\sigma$.

- Similarly, if $\varphi(x_i, x_j) = E(x_i, x_j)$, then for a valuation $\nu$ assigning $x_i$ to $v_i$ and $x_j$ to $v_j$:

$$[\![\varphi, \nu]\!]_G = \mathbf{1}_{v_i v_j \in E} = \mathbf{1}_{\sigma(v_i)\sigma(v_j) \in E^\sigma} = [\![\varphi, \sigma \star \nu]\!]_{\sigma \star G},$$

where we used the definition of $E^\sigma$.

- If $\varphi(\boldsymbol{x}) = \varphi_1(\boldsymbol{x}_1) \cdot \varphi_2(\boldsymbol{x}_2)$, then for a valuation $\nu$ from $\boldsymbol{x}$ to $V$:

$$[\![\varphi, \nu]\!]_G = [\![\varphi_1, \nu]\!]_G \cdot [\![\varphi_2, \nu]\!]_G = [\![\varphi_1, \sigma \star \nu]\!]_{\sigma \star G} \cdot [\![\varphi_2, \sigma \star \nu]\!]_{\sigma \star G} = [\![\varphi, \sigma \star \nu]\!]_{\sigma \star G},$$

where we used the induction hypothesis for $\varphi_1$ and $\varphi_2$. The cases $\varphi(\boldsymbol{x}) = \varphi_1(\boldsymbol{x}_1) + \varphi_2(\boldsymbol{x}_2)$ and $\varphi(\boldsymbol{x}) = a \cdot \varphi_1(\boldsymbol{x})$ are dealt with in a similar way.

- If $\varphi(\boldsymbol{x}) = f(\varphi_1(\boldsymbol{x}_1), \ldots, \varphi_p(\boldsymbol{x}_p))$, then

$$\begin{aligned}
[\![\varphi, \nu]\!]_G &= f([\![\varphi_1, \nu]\!]_G, \ldots, [\![\varphi_p, \nu]\!]_G) \\
&= f([\![\varphi_1, \sigma \star \nu]\!]_{\sigma \star G}, \ldots, [\![\varphi_p, \sigma \star \nu]\!]_{\sigma \star G}) \\
&= [\![\varphi, \sigma \star \nu]\!]_{\sigma \star G},
\end{aligned}$$

where we used again the induction hypothesis for $\varphi_1, \ldots, \varphi_p$.

- Finally, if $\varphi(\boldsymbol{x}) = \sum_y \varphi_1(\boldsymbol{x}, y)$ then for a valuation $\nu$ of $\boldsymbol{x}$ to $V$:

$$\begin{aligned}
[\![\varphi, \nu]\!]_G &= \sum_{v \in V} [\![\varphi_1, \nu[y \mapsto v]]\!]_G = \sum_{v \in V} [\![\varphi_1, \sigma \star \nu[y \mapsto v]]\!]_{\sigma \star G} \\
&= \sum_{v \in V^\sigma} [\![\varphi_1, \sigma \star \nu[y \mapsto v]]\!]_{\sigma \star G} = [\![\varphi, \sigma \star \nu]\!]_{\sigma \star G},
\end{aligned}$$

where we used the induction hypothesis for $\varphi_1$ and that $V^\sigma = V$ because $\sigma$ is a permutation.

We remark that when $\varphi$ does not contain free index variables, then $[\![\varphi, \nu]\!]_G = [\![\varphi]\!]_G$ for any valuation $\nu$, from which invariance follows from the previous arguments. This concludes the proof of Proposition 3.1.

## C    DETAILS OF SECTION 4

In the following sections we prove Theorem 4.1, 4.2, 4.3 and 4.4. More specifically, we start by showing these results in the setting that $\mathsf{TL}(\Omega)$ only supports summation aggregation ($\sum_x e$) and in which the vertex-labellings in graphs take values in $\{0, 1\}^\ell$. In this context, we introduce classical logics in Section C.1 and recall and extend connections between the separation power of these logics and the separation power of color refinement and $k$-WL in Section C.2. We connect $\mathsf{TL}(\Omega)$ and logics in Section C.3, to finally obtain the desired proofs in Section C.4. We then show how these results can be generalized in the presence of general aggregation operators in Section C.5, and to the setting where vertex-labellings take values in $\mathbb{R}^\ell$ in Section C.6.

### C.1    CLASSICAL LOGICS

In what follows, we consider graphs $G = (V_G, E_G, \mathsf{col}_G)$ with $\mathsf{col}_G : V_G \to \{0, 1\}^\ell$. We start by defining the $k$-*variable fragment* $\mathsf{C}^k$ of first-order logic with counting quantifiers, followed by the definition of the *guarded fragment* $\mathsf{GC}$ of $\mathsf{C}^2$. Formulae $\varphi$ in $\mathsf{C}^k$ are defined over the set $\{x_1, \ldots, x_k\}$ of variables and are formed by the following grammar:

$$\varphi := (x_i = x_j) \mid E(x_i, x_j) \mid P_s(x_i) \mid \neg\varphi \mid \varphi \wedge \varphi \mid \exists^{\geq m} x_i\, \varphi,$$

where $i, j \in [k]$, $E$ is a binary predicate, $P_s$ for $s \in [\ell]$ are unary predicates for some $\ell \in \mathbb{N}$, and $m \in \mathbb{N}$. The semantics of formulae in $\mathsf{C}^k$ is defined in terms of interpretations relative to a given graph $G$ and a (partial) valuation $\mu : \{x_1, \dots, x_k\} \to V_G$. Such an interpretation maps formulae, graphs and valuations to Boolean values $\mathbb{B} := \{\bot, \top\}$, in a similar way as we did for tensor language expressions.

More precisely, given a graph $G = (V_G, E_G, \mathsf{col}_G)$ and partial valuation $\mu : \{x_1, \dots, x_k\} \to V_G$, we define $[\![\varphi, \mu]\!]_G^{\mathbb{B}} \in \mathbb{B}$ for valuations defined on the free variables in $\varphi$. That is, we define:

$$[\![x_i = x_j, \mu]\!]_G^{\mathbb{B}} := \text{if } \mu(x_i) = \mu(x_j) \text{ then } \top \text{ else } \bot;$$
$$[\![E(x_i, x_j), \mu]\!]_G^{\mathbb{B}} := \text{if } \mu(x_i)\mu(x_j) \in E_G \text{ then } \top \text{ else } \bot;$$
$$[\![P_s(x_i), \mu]\!]_G^{\mathbb{B}} := \text{if } \mathsf{col}_G(\mu(x_i))_s = 1 \text{ then } \top \text{ else } \bot;$$
$$[\![\neg\varphi, \mu]\!]_G^{\mathbb{B}} := \neg[\![\varphi, \mu]\!]_G^{\mathbb{B}};$$
$$[\![\varphi_1 \wedge \varphi_2, \mu]\!]_G^{\mathbb{B}} := [\![\varphi_1, \mu]\!]_G^{\mathbb{B}} \wedge [\![\varphi_2, \mu]\!]_G^{\mathbb{B}};$$
$$[\![\exists^{\geq m} x_i \, \varphi_1, \mu]\!]_G^{\mathbb{B}} := \text{ if } |\{v \in V_G \mid [\![\varphi, \mu[x_i \mapsto v]]\!]_G^{\mathbb{B}} = \top\}| \geq m \text{ then } \top \text{ else } \bot.$$

In the last expression, $\mu[x_i \mapsto v]$ denotes the valuation $\mu$ modified such that it maps $x_i$ to vertex $v$.

We will also need the *guarded fragment* $\mathsf{GC}$ of $\mathsf{C}^2$ in which we only allow equality conditions of the form $x_i = x_i$, component expressions of conjunction and disjunction should have the same single free variable, and counting quantifiers can only occur in guarded form: $\exists^{\geq m} x_2(E(x_1, x_2) \wedge \varphi(x_2))$ or $\exists^{\geq m} x_1(E(x_2, x_1) \wedge \varphi(x_1))$. The semantics of formulae in $\mathsf{GC}$ is inherited from formulae in $\mathsf{C}^2$.

Finally, we will also consider $\mathsf{C}^k_{\infty\omega}$, that is, the logic $\mathsf{C}^k$ extended with *infinitary disjunctions* and *conjunctions*. More precisely, we add to the grammar of formulae the following constructs:

$$\bigvee_{\alpha \in A} \varphi_\alpha \quad \text{and} \quad \bigwedge_{\alpha \in A} \varphi_\alpha$$

where the index set $A$ can be arbitrary, even containing uncountably many indices. We define $\mathsf{GC}_{\infty\omega}$ in the same way by relaxing the finite variable conditions. The semantics is, as expected: $[\![\bigvee_{\alpha \in A} \varphi_\alpha, \mu]\!]_G^{\mathbb{B}} = \top$ if for at least one $\alpha \in A$, $[\![\varphi_\alpha, \mu]\!]_G^{\mathbb{B}} = \top$, and $[\![\bigwedge_{\alpha \in A} \varphi_\alpha, \mu]\!]_G^{\mathbb{B}} = \top$ if for all $\alpha \in A$, $[\![\varphi_\alpha, \mu]\!]_G^{\mathbb{B}} = \top$.

We define the *free variables* of formulae just as for $\mathsf{TL}$, and similarly, *quantifier rank* is defined as summation depth (only existential quantifications increase the quantifier rank). For any of the above logics $\mathcal{L}$ we define $\mathcal{L}^{(t)}$ as the set of formulae in $\mathcal{L}$ of quantifier rank at most $t$.

To capture the *separation power of logics*, we define $\rho_1(\mathcal{L}^{(t)})$ as the equivalence relation on $\mathcal{G}_1$ defined by

$$((G, v), (H, w)) \in \rho_1(\mathcal{L}^{(t)}) \iff \forall \varphi(x) \in \mathcal{L}^{(t)} : [\![\varphi, \mu_v]\!]_G^{\mathbb{B}} = [\![\varphi, \mu_w]\!]_H^{\mathbb{B}},$$

where $\mu_v$ is any valuation such that $\mu(x) = v$, and likewise for $w$. The relation $\rho_0$ is defined in a similar way, except that now the relation is only over pairs of graphs, and the characterization is over all formulae with no free variables (also called sentences). Finally, we also use, and define, the relation $\rho_s$, which relates pairs from $\mathcal{G}_s$: consisting of a graph and an $s$-tuple of vertices. The relation is defined as

$$((G, \boldsymbol{v}), (H, \boldsymbol{w})) \in \rho_s(\mathcal{L}^{(t)}) \iff \forall \varphi(\boldsymbol{x}) \in \mathcal{L}^{(t)} : [\![\varphi, \mu_{\boldsymbol{v}}]\!]_G^{\mathbb{B}} = [\![\varphi, \mu_{\boldsymbol{w}}]\!]_H^{\mathbb{B}},$$

where $\boldsymbol{x}$ consist of $s$ free variables and $\mu_{\boldsymbol{v}}$ is a valuation assigning the $i$-th variable of $\boldsymbol{x}$ to the $i$-th value of $\boldsymbol{v}$, for any $i \in [s]$.

## C.2 CHARACTERIZATION OF SEPARATION POWER OF LOGICS

We first connect the separation power of the color refinement and $k$-dimensional Weisfeiler-Leman algorithms to the separation power of the logics we just introduced. Although most of these connections are known, we present them in a bit of a more fine-grained way. That is, we connect the number of rounds used in the algorithms to the quantifier rank of formulae in the above logics.

**Proposition C.1.** *For any $t \geq 0$, we have the following identities:*

*(1)* $\rho_1\big(\mathsf{cr}^{(t)}\big) = \rho_1\big(\mathsf{GC}^{(t)}\big)$ *and* $\rho_0\big(\mathsf{gcr}^{(t)}\big) = \rho_0\big(\mathsf{gwl}_1^{(t)}\big) = \rho_0\big(\mathsf{C}^{2,(t+1)}\big)$;

*(2) For* $k \geq 1$, $\rho_1\big(\mathsf{vwl}_k^{(t)}\big) = \rho_1\big(\mathsf{C}^{k+1,(t)}\big)$ *and*

$$\rho_0\big(\mathsf{C}^{k+1,(t+k)}\big) \subseteq \rho_0\big(\mathsf{gwl}_k^{(t)}\big) \subseteq \rho_0\big(\mathsf{C}^{k+1,(t+1)}\big).$$

*As a consequence,* $\rho_0\big(\mathsf{gwl}_k^{(\infty)}\big) = \rho_0\big(\mathsf{C}^{k+1}\big)$.

*Proof.* For (1), the identity $\rho_1\big(\mathsf{cr}^{(t)}\big) = \rho_1\big(\mathsf{GC}^{(t)}\big)$ is known and can be found, for example, in Theorem V.10 in Grohe (2021). The identity $\rho_0\big(\mathsf{gcr}^{(t)}\big) = \rho_0\big(\mathsf{gwl}_1^{(t)}\big)$ can be found in Proposition V.4 in Grohe (2021). The identity $\rho_0\big(\mathsf{gwl}_1^{(t)}\big) = \rho_0\big(\mathsf{C}^{2,(t+1)}\big)$ is a consequence of the inclusion shown in (2) for $k = 1$.

For (2), we use that $\rho_k\big(\mathsf{wl}_k^{(t)}\big) = \rho_k\big(\mathsf{C}^{k+1,(t)}\big)$, see e.g., Theorem V.8 in Grohe (2021). We argue that this identity holds for $\rho_1\big(\mathsf{vwl}_k^{(t)}\big) = \rho_1\big(\mathsf{C}^{k+1,(t)}\big)$. Indeed, suppose that $(G, v)$ and $(H, w)$ are not in $\rho_1\big(\mathsf{C}^{k+1,(t)}\big)$. Let $\varphi(x_1)$ be a formula in $\mathsf{C}^{k+1,(t)}$ such that $[\![\varphi, v]\!]_G^{\mathbb{B}} \neq [\![\varphi, w]\!]_H^{\mathbb{B}}$. Consider the formula $\varphi^+(x_1, \ldots, x_k) = \varphi(x_1) \wedge \bigwedge_{i=1}^k (x_1 = x_i)$. Then, $[\![\varphi^+, (v, \ldots, v)]\!]_G^{\mathbb{B}} \neq [\![\varphi^+, (w, \ldots, w)]\!]_H^{\mathbb{B}}$, and hence $(G, (v, \ldots, v))$ and $(H, (w, \ldots, w))$ are not in $\rho_k\big(\mathsf{C}^{k+1,(t)}\big)$ either. This implies that $\mathsf{wl}_k^{(t)}(G, (v, \ldots, v)) \neq \mathsf{wl}_k^{(t)}(H, (w, \ldots, w))$, and thus, by definition, $\mathsf{vwl}_k^{(t)}(G, v) \neq \mathsf{vwl}_k^{(t)}(H, w)$. In other words, $(G, v)$ and $(H, w)$ are not in $\rho_1\big(\mathsf{vwl}_k^{(t)}\big)$, from which the inclusion $\rho_1\big(\mathsf{vwl}_k^{(t)}\big) \subseteq \rho_1\big(\mathsf{C}^{k+1,(t)}\big)$ follows. Conversely, if $(G, v)$ and $(H, w)$ are not in $\rho_1\big(\mathsf{vwl}_k^{(t)}\big)$, then $\mathsf{wl}_k^{(t)}(G, (v, \ldots, v)) \neq \mathsf{wl}_k^{(t)}(H, (w, \ldots, w))$. As a consequence, $(G, (v, \ldots, v))$ and $(H, (w, \ldots, w))$ are not in $\rho_k\big(\mathsf{C}^{k+1,(t)}\big)$ either. Let $\varphi(x_1, \ldots, x_k)$ be a formula in $\mathsf{C}^{k+1,(t)}$ such that $[\![\varphi, (v, \ldots, v)]\!]_G^{\mathbb{B}} \neq [\![\varphi, (w, \ldots, w)]\!]_H^{\mathbb{B}}$. Then it is readily shown that we can convert $\varphi(x_1, \ldots, x_k)$ into a formula $\varphi^-(x_1)$ in $\mathsf{C}^{k+1,(t)}$ such that $[\![\varphi^-, v]\!]_G^{\mathbb{B}} \neq [\![\varphi^-, w]\!]_H^{\mathbb{B}}$, and thus $(G, v)$ and $(H, w)$ are not in $\rho_1\big(\mathsf{C}^{k+1,(t)}\big)$. Hence, we also have the inclusion $\rho_1\big(\mathsf{vwl}_k^{(t)}\big) \supseteq \rho_1\big(\mathsf{C}^{k+1,(t)}\big)$, form which the first identity in (2) follows.

It remains to show $\rho_0\big(\mathsf{C}^{k+1,(t+k)}\big) \subseteq \rho_0\big(\mathsf{gwl}_k^{(t)}\big) \subseteq \rho_0\big(\mathsf{C}^{k+1,(t+1)}\big)$. Clearly, if $(G, H)$ is not in $\rho_0\big(\mathsf{gwl}_k^{(t)}\big)$ then the multisets of labels $\mathsf{wl}_k^{(t)}(G, \boldsymbol{v})$ and $\mathsf{wl}_k^{(t)}(H, \boldsymbol{w})$ differ. It is known that with each label $c$ one can associate a formula $\varphi^c$ in $\mathsf{C}^{k+1,(t)}$ such that $[\![\varphi^c, \boldsymbol{v}]\!]_G^{\mathbb{B}} = \top$ if and only if $\mathsf{wl}_k^{(t)}(G, \boldsymbol{v}) = c$. So, if the multisets are different, there must be a $c$ that occurs more often in one multiset than in the other one. This can be detected by a fomulae of the form $\exists^{=m}(x_1, \ldots, x_k)\varphi^c(x_1, \ldots, x_k)$ which is satisfied if there are $m$ tuples $\boldsymbol{v}$ with label $c$. It is now easily verified that the latter formula can be converted into a formula in $\mathsf{C}^{k+1,(t+k)}$. Hence, the inclusion $\rho_0\big(\mathsf{C}^{k+1,(t+k)}\big) \subseteq \rho_0\big(\mathsf{gwl}_k^{(t)}\big)$ follows.

For $\rho_0\big(\mathsf{gwl}_k^{(t)}\big) \subseteq \rho_0\big(\mathsf{C}^{k+1,(t+1)}\big)$, we show that if $(G, H)$ is in $\rho_0\big(\mathsf{gwl}_k^{(t)}\big)$, then this implies that $[\![\varphi, \mu]\!]_G^{\mathbb{B}} = [\![\varphi, \mu]\!]_H^{\mathbb{B}}$ for all formulae in $\mathsf{C}^{k+1,(t+1)}$ and any valuation $\mu$ (notice that $\mu$ is superfluous in this definition when formulas have no free variables). Assume that $(G, H)$ is in $\rho_0\big(\mathsf{gwl}_k^{(t)}\big)$. Since any formula of quantifier rank $t+1$ is a Boolean combination of formulas of less rank or a formula of the form $\varphi = \exists^{\geq m} x_i\, \psi$ where $\psi$ is of quantifier rank $t$, without loss of generality consider a formula of the latter form, and assume for the sake of contradiction that $[\![\varphi, \mu]\!]_G^{\mathbb{B}} = \top$ but $[\![\varphi, \mu]\!]_H^{\mathbb{B}} = \bot$. Since $[\![\varphi, \mu]\!]_G^{\mathbb{B}} = \top$, there must be at least $m$ elements satisfying $\psi$. More precisely, let $v_1, \ldots, v_p$ in $G$ be all vertices in $G$ such that for each valuation $\mu[x \mapsto v_i]$ it holds that $[\![\psi, \mu[x \mapsto v_i]]\!]_G^{\mathbb{B}} = \top$. As mentioned, it must be that $p$ is at least $m$. Using again the fact that $\rho_k\big(\mathsf{wl}_k^{(t)}\big) = \rho_k\big(\mathsf{C}^{k+1,(t)}\big)$, we infer that the color $\mathsf{wl}_k^{(t-1)}(G, (v_i, \ldots, v_i))$ is the same, for each such $v_i$.

Now since $\mathsf{gwl}_k^{(t-1)}(G) = \mathsf{gwl}_k^{(t-1)}(H)$, it is not difficult to see that there must be exactly $p$ vertices $w_1, \ldots, w_p$ in $H$ such that $\mathsf{wl}_k^{(t-1)}(G, (v_i, \ldots, v_i)) = \mathsf{wl}_k^{(t-1)}(H, (w_i, \ldots, w_i))$. Otherwise, it would simply not be the case that the aggregation step of the colors, assigned by $k$-WL is the same in $G$ and $H$. By the connection to logic, we again know that for valuation $\mu[x \mapsto w_i]$ it holds that $[\![\psi, \mu[x \mapsto w_i]]\!]_H^{\mathbb{B}} = \top$. It then follows that $[\![\varphi, \mu]\!]_H^{\mathbb{B}} = \top$ for any valuation $\mu$, which was to be shown.

Finally, we remark that $\rho_0\big(\mathsf{gwl}_k^{(\infty)}\big) = \rho_0\big(\mathsf{C}^{k+1}\big)$ follows from the preceding inclusions in (2). $\qquad\square$

Before moving to tensor languages, where we will use infinitary logics to simulate expressions in $\mathsf{TL}_k(\Omega)$ and $\mathsf{GTL}(\Omega)$, we recall that, when considering the separation power of logics, we can freely move between the logics and their infinitary counterparts:

**Theorem C.2.** *The following identities hold for any $t \geq 0$, $k \geq 2$ and $s \geq 0$:*

*(1)* $\rho_1\big(\mathsf{GC}^{(t)}_{\infty\omega}\big) := \rho_1\big(\mathsf{GC}^{(t)}\big)$;

*(2)* $\rho_s\big(\mathsf{C}^{k,(t)}_{\infty\omega}\big) = \rho_s\big(\mathsf{C}^{k,(t)}\big)$.

*Proof.* For identity (1), notice that we only need to prove that $\rho_1\big(\mathsf{GC}^{(t)}\big) \subseteq \rho_1\big(\mathsf{GC}^{(t)}_{\infty\omega}\big)$, the other direction follows directly from the definition. We point out the well-known fact that two tuples $(G, v)$ and $(H, w)$ belong to $\rho_1\big(\mathsf{GC}^{(t)}\big)$ if and only if the *unravelling* of $G$ rooted at $v$ up to depth $t$ is isomorphic to the *unravelling* of $H$ rooted at $w$ up to root $t$. Here the *unravelling* is the infinite tree whose root is the root node, and whose children are the neighbors of the root node (see e.g. Barceló et al. (2020); Otto (2019). Now for the connection with infinitary logic. Assume that the unravellings of $G$ rooted at $v$ and of $H$ rooted at $w$ up to level $t$ are isomorphic, but assume for the sake of contradiction that there is a formula $\varphi(x)$ in $\mathsf{GC}^{(t)}_{\infty\omega}$ such that $[\![\varphi, \mu_v]\!]^{\mathbb{B}}_G \neq [\![\varphi, \mu_w]\!]^{\mathbb{B}}_H$, where $\mu_v$ and $\mu_w$ are any valuation mapping variable $x$ to $v$ and $w$, respectively. Now since $G$ and $H$ are finite graphs, one can construct, from formula $\phi$, a formula $\phi'$ in $\mathsf{GC}^{(t)}$ such that $[\![\psi, \mu_v]\!]^{\mathbb{B}}_G \neq [\![\psi, \mu_w]\!]^{\mathbb{B}}_H$. Notice that this is in contradiction with our assumption that unravellings where isomorphic and therefore indistinguishable by formulae in $\mathsf{GC}^{(t)}$. To construct $\psi$, consider an infinitary disjunction $\bigvee_{a \in A} \alpha_a$. Since $G$ and $H$ have a finite number of vertices, and the formulae have a finite number of variables, the number of different valuations from the variables to the vertices in $G$ or $H$ is also finite. Thus, one can replace any extra copy of $\alpha_a$, $\alpha_{a'}$ such that their value is the same in $G$ and $H$. The final result is a finite disjunction, and the truth value over $G$ and $H$ is equivalent to the original infinitary disjunction.

For identity (2) we refer to Corollary 2.4 in Otto (2017). $\qquad\square$

## C.3   From $\mathsf{TL}(\Omega)$ to $\mathsf{C}^k_{\infty\omega}$ and $\mathsf{GC}_{\infty\omega}$

We are now finally ready to make the connection between expressions in $\mathsf{TL}(\Omega)$ and the infinitary logics introduced earlier.

**Proposition C.3.** *For any expression $\varphi(\boldsymbol{x})$ in $\mathsf{TL}_k(\Omega)$ and $c \in \mathbb{R}$, there exists an expression $\tilde{\varphi}^c(\boldsymbol{x})$ in $\mathsf{C}^k_{\infty\omega}$ such that $[\![\varphi, \boldsymbol{v}]\!]_G = c$ if and only if $[\![\tilde{\varphi}^c, \boldsymbol{v}]\!]^{\mathbb{B}}_G = \top$ for any graph $G = (V_G, E_G, \mathsf{col}_G)$ in $\mathcal{G}$ and $\boldsymbol{v} \in V^k_G$. Furthermore, if $\varphi(x) \in \mathsf{GTL}(\Omega)$ then $\tilde{\varphi}^c \in \mathsf{GC}_{\infty\omega}$. Finally, if $\varphi$ has summation depth $t$ then $\tilde{\varphi}^c$ has quantifier rank $t$.*

*Proof.* We define $\tilde{\varphi}^c$ inductively on the structure of expressions in $\mathsf{TL}_k(\Omega)$.

- $\varphi(x_i, x_j) := \mathbf{1}_{x_i \, \mathsf{op} \, x_j}$. Assume first that op is "$=$". We distinguish between (a) $i \neq j$ and (b) $i = j$. For case (a), if $c = 1$, then we define $\tilde{\varphi}^1(x_i, x_j) := (x_i = x_j)$, if $c = 0$, then we define $\tilde{\varphi}^0(x_i, x_j) := \neg(x_i = x_j)$, and if $c \neq 0, 1$, then we define $\tilde{\varphi}^c(x_i, x_j) := x_i \neq x_i$. For case (b), if $c = 1$, then we define $\tilde{\varphi}^1(x_i, x_j) := (x_i = x_i)$, and for any $c \neq 1$, we define $\tilde{\varphi}^c(x_i, x_j) := \neg(x_i = x_i)$. The case when op is "$\neq$" is treated analogously.

- $\varphi(x_i) := P_\ell(x_i)$. If $c = 1$, then we define $\tilde{\varphi}^1(x_i) := P_\ell(x_i)$, if $c = 0$, then we define $\tilde{\varphi}^0(x_i) := \neg P_j(x_i)$. For all other $c$, we define $\tilde{\varphi}^c(x_i, x_j) := \neg(x_i = x_i)$.

- $\varphi(x_i, x_j) := E(x_i, x_j)$. If $c = 1$, then we define $\tilde{\varphi}^1(x_i, x_j) := E(x_i, x_j)$, if $c = 0$, then we define $\tilde{\varphi}^0(x_i, x_j) := \neg E(x_i, x_j)$. For all other $c$, we define $\tilde{\varphi}^c(x_i, x_j) := \neg(x_i = x_i)$.

- $\varphi := \varphi_1 + \varphi_2$. We observe that $[\![\varphi, \boldsymbol{v}]\!]_G = c$ if and only if there are $c_1, c_2 \in \mathbb{R}$ such that $[\![\varphi_1, \boldsymbol{v}]\!]_G = c_1$ and $[\![\varphi_2, \boldsymbol{v}]\!]_G = c_2$ and $c = c_1 + c_2$. Hence, it suffices to define

$$\tilde{\varphi}^c := \bigvee_{\substack{c_1, c_2 \in \mathbb{R} \\ c = c_1 + c_2}} \tilde{\varphi}_1^{c_1} \wedge \tilde{\varphi}_2^{c_2},$$

where $\tilde{\varphi}_1^{c_1}$ and $\tilde{\varphi}_2^{c_2}$ are the expressions such that $[\![\varphi_1, \boldsymbol{v}]\!]_G = c_1$ if and only if $[\![\tilde{\varphi}_1^{c_1}, \boldsymbol{v}]\!]_G^{\mathbb{B}} = \top$ and $[\![\varphi_2, \boldsymbol{v}]\!]_G = c_2$ if and only if $[\![\tilde{\varphi}_2^{c_2}, \boldsymbol{v}]\!]_G^{\mathbb{B}} = \top$, which exist by induction.

- $\varphi := \varphi_1 \cdot \varphi_2$. This is case is analogous to the previous one. Indeed, $[\![\varphi, \boldsymbol{v}]\!]_G = c$ if and only if there are $c_1, c_2 \in \mathbb{R}$ such that $[\![\varphi_1, \boldsymbol{v}]\!]_G = c_1$ and $[\![\varphi_2, \boldsymbol{v}]\!]_G = c_2$ and $c = c_1 \cdot c_2$. Hence, it suffices to define

$$\tilde{\varphi}^c := \bigvee_{\substack{c_1, c_2 \in \mathbb{R} \\ c = c_1 \cdot c_2}} \tilde{\varphi}_1^{c_1} \wedge \tilde{\varphi}_2^{c_2}.$$

- $\varphi := a \cdot \varphi_1$. This is case is again dealt with in a similar way. Indeed, $[\![\varphi, \boldsymbol{v}]\!]_G = c$ if and only if there is a $c_1 \in \mathbb{R}$ such that $[\![\varphi_1, \boldsymbol{v}]\!]_G = c_1$ and $c = a \cdot c_1$. Hence, it suffices to define

$$\tilde{\varphi}^c := \bigvee_{\substack{c_1 \in \mathbb{R} \\ c = a \cdot c_1}} \tilde{\varphi}_1^{c_1}.$$

- $\varphi := f(\varphi_1, \ldots, \varphi_p)$ with $f : \mathbb{R}^p \to \mathbb{R}$. We observe that $[\![\varphi, \boldsymbol{v}]\!]_G = c$ if and only if there are $c_1, \ldots, c_p \in \mathbb{R}$ such that $c = f(c_1, \ldots, c_p)$ and $[\![\varphi_i, \boldsymbol{v}]\!]_G = c_i$ for $i \in [p]$. Hence, it suffices to define

$$\tilde{\varphi}^c := \bigvee_{\substack{c_1, \ldots, c_p \in \mathbb{R} \\ c = f(c_1, \ldots, c_p)}} \tilde{\varphi}_1^{c_1} \wedge \cdots \wedge \tilde{\varphi}_p^{c_p}.$$

- $\varphi := \sum_{x_i} \varphi_1$. We observe that $[\![\varphi, \mu]\!]_G = c$ implies that we can partition $V_G$ into $\ell$ parts $V_1, \ldots, V_\ell$, of sizes $m_1, \ldots, m_\ell$, respectively, such that $[\![\varphi_1, \mu[x_i \to v]]\!]_G = c_i$ for each $v \in V_i$, and such that all $c_i$'s are pairwise distinct and $c = \sum_{i=1}^\ell c_i \cdot m_i$. It now suffices to consider the following formula

$$\tilde{\varphi}^c := \bigvee_{\substack{\ell, m_1, \ldots, m_\ell \in \mathbb{N} \\ c_1, \ldots, c_\ell \in \mathbb{R} \\ c = \sum_{i=1}^\ell m_i c_i}} \bigwedge_{i=1}^\ell \exists^{=m_i} x_i \, \tilde{\varphi}_1^{c_i} \wedge \forall x_i \bigvee_{i=1}^\ell \tilde{\varphi}_1^{c_i},$$

where $\exists^{=m_i} x_i \, \psi$ is shorthand notation for $\exists^{\geq m_i} x_i \, \psi \wedge \neg \exists^{\geq m_i + 1} x_i \, \psi$, and $\forall x_i \, \psi$ denotes $\neg \exists^{\geq 1} x_i \, \neg \psi$.

This concludes the construction of $\tilde{\varphi}^c$. We observe that we only introduce a quantifiers when $\varphi = \sum_{x_i} \varphi_1$ and hence if we assume by induction that summation depth and quantifier rank are in sync, then if $\varphi_1$ has summation depth $t-1$ and thus $\tilde{\varphi}_1^c$ has quantifier rank $t-1$ for any $c \in \mathbb{R}$, then $\varphi$ has summation depth $t$, and as can be seen from the definition of $\tilde{\varphi}^c$, this formula has quantifier rank $t$, as desired.

It remains to verify the claim about guarded expressions. This is again verified by induction. The only case requiring some attention is $\varphi(x_1) := \sum_{x_2} E(x_1, x_2) \wedge \varphi_1(x_2)$ for which we can define

$$\tilde{\varphi}^c := \bigvee_{\substack{\ell, m_1, \ldots, m_\ell \in \mathbb{N} \\ c_1, \ldots, c_\ell \in \mathbb{R} \\ c = \sum_{i=1}^\ell m_i c_i \\ m = \sum_{i=1}^\ell m_i}} \exists^{=m} x_2 E(x_1, x_2) \wedge \bigwedge_{i=1}^\ell \exists^{=m_i} x_2 \, E(x_1, x_2) \wedge \tilde{\varphi}_1^{c_i}(x_2),$$

which is a formula in GC again only adding one to the quantifier rank of the formulae $\tilde{\varphi}_1^c$ for $c \in \mathbb{R}$. So also here, we have the one-to-one correspondence between summation depth and quantifier rank. $\qquad\square$

## C.4 Proof of Theorem 4.1, 4.2, 4.3 and 4.4

**Proposition C.4.** *We have the following inclusions: For any $t \geq 0$ and any collection $\Omega$ of functions:*

- $\rho_1\big(\mathsf{cr}^{(t)}\big) \subseteq \rho_1\big(\mathsf{GTL}^{(t)}(\Omega)\big);$

- $\rho_1\big(\mathsf{vwl}_k^{(t)}\big) \subseteq \rho_1\big(\mathsf{TL}_{k+1}^{(t)}(\Omega)\big)$; and

- $\rho_0\big(\mathsf{gwl}_k^{(t)}\big) \subseteq \rho_0\big(\mathsf{TL}_{k+1}^{(t+1)}(\Omega)\big)$.

*Proof.* We first show the second bullet by contraposition. That is, we show that if $(G, v)$ and $(H, w)$ are not in $\rho_1\big(\mathsf{TL}_{k+1}^{(t)}(\Omega)\big)$, then neither are they in $\rho_1\big(\mathsf{vwl}_k^{(t)}\big)$. Indeed, suppose that there exists an expression $\varphi(x_1)$ in $\mathsf{TL}_{k+1}^{(t)}(\Omega)$ such that $[\![\varphi, v]\!]_G = c \neq c' = [\![\varphi, w]\!]_H$. From Proposition C.3 we know that there exists a formula $\tilde{\varphi}^c$ in $\mathsf{C}_{\infty\omega}^{k+1,(t)}$ such that $[\![\tilde{\varphi}^c, v]\!]_G^\mathbb{B} = \top$ and $[\![\tilde{\varphi}^c, w]\!]_H^\mathbb{B} = \bot$. Hence, $(G, v)$ and $(H, w)$ do no belong to $\rho_1\big(\mathsf{C}_{\infty\omega}^{k+1,(t)}\big)$. Theorem C.2 implies that $(G, v)$ and $(H, w)$ also do not belong to $\rho_1\big(\mathsf{C}^{k+1,(t)}\big)$. Finally, Proposition C.1 implies that $(G, v)$ and $(H, w)$ do not belong to $\rho_1\big(\mathsf{vwl}_k^{(t)}\big)$, as desired. The third bullet is shown in precisely the same, but using the identities for $\rho_0$ rather than $\rho_1$, and $\mathsf{gwl}_k^{(t)}$ rather than $\mathsf{vwl}_k^{(t)}$.

Also the first bullet is shown in the same way, using the connection between $\mathsf{GTL}^{(t)}(\Omega)$, $\mathsf{GC}_{\infty\omega}^{2,(t)}$, $\mathsf{GC}^{(t)}$ and $\mathsf{cr}^{(t)}$, as given by Proposition C.1, Theorem C.2, and Proposition C.3. □

We next show that our tensor languages are also more separating than the color refinement and $k$-dimensional Weisfeiler-Leman algorithms.

**Proposition C.5.** *We have the following inclusions: For any $t \geq 0$ and any collection $\Omega$ of functions:*

- $\rho_1\big(\mathsf{GTL}^{(t)}(\Omega)\big) \subseteq \rho_1\big(\mathsf{cr}^{(t)}\big)$;

- $\rho_1\big(\mathsf{TL}_{k+1}^{(t)}(\Omega)\big) \subseteq \rho_1\big(\mathsf{vwl}_k^{(t)}\big)$; and

- $\rho_0\big(\mathsf{TL}_{k+1}^{(t+k)}(\Omega)\big) \subseteq \rho_0\big(\mathsf{gwl}_k^{(t)}\big)$.

*Proof.* For any of these inclusions to hold, for any $\Omega$, we need to show the inclusion without the use of any functions. We again use the connections between the color refinement and $k$-dimensional Weisfeiler-Leman algorithms and finite variable logics as stated in Proposition C.1. More precisely, we show for any formula $\varphi(\boldsymbol{x}) \in \mathsf{C}^{k,(t)}$ there exists an expression $\hat{\varphi}(\boldsymbol{x}) \in \mathsf{TL}_k^{(t)}$ such that for any graph $G$ in $\mathcal{G}$, $[\![\varphi, \boldsymbol{v}]\!]_G^\mathbb{B} = \top$ implies $[\![\hat{\varphi}, \boldsymbol{v}]\!]_G = 1$ and $[\![\varphi, \boldsymbol{v}]\!]_G^\mathbb{B} = \bot$ implies $[\![\hat{\varphi}, \boldsymbol{v}]\!]_G = 0$. By appropriately selecting $k$ and $t$ and by observing that when $\varphi(x) \in \mathsf{GC}$ then $\hat{\varphi}(x) \in \mathsf{GTL}$, the inclusions follow.

The construction of $\hat{\varphi}(\boldsymbol{x})$ is by induction on the structure of formulae in $\mathsf{C}^k$.

- $\varphi := (x_i = x_j)$. Then, we define $\hat{\varphi} := \mathbf{1}_{x_i = x_j}$.

- $\varphi := P_\ell(x_i)$. Then, we define $\hat{\varphi} := P_\ell(x_i)$.

- $\varphi := E(x_i, x_j)$. Then, we define $\hat{\varphi} := E(x_i, x_j)$.

- $\varphi := \neg\varphi_1$. Then, we define $\hat{\varphi} := \mathbf{1}_{x_i = x_i} - \hat{\varphi}_1$.

- $\varphi := \varphi_1 \wedge \varphi_2$. Then, we define $\hat{\varphi} := \hat{\varphi}_1 \cdot \hat{\varphi}_2$.

- $\varphi := \exists^{\geq m} x_i\, \varphi_1$. Consider a polynomial $p(x) := \sum_j a_j x^j$ such that $p(x) = 0$ for $x \in \{0, 1, \ldots, m-1\}$ and $p(x) = 1$ for $x \in \{m, m+1, \ldots, n\}$. Such a polynomial exists by interpolation. Then, we define $\hat{\varphi} := \sum_j a_j \big(\sum_{x_i} \hat{\varphi}_1\big)^j$.

We remark that we here crucially rely on the assumption that $\mathcal{G}$ contains graphs of fixed size $n$ and that $\mathsf{TL}_k$ is closed under linear combinations and product. Clearly, if $\varphi \in \mathsf{GC}$, then the above translations results in an expression $\hat{\varphi} \in \mathsf{GTL}(\Omega)$. Furthermore, the quantifier rank of $\varphi$ is in one-to-one correspondence to the summation depth of $\hat{\varphi}$.

We can now apply Proposition C.1. That is, if $(G, v)$ and $(H, w)$ are not in $\rho_1\big(\mathsf{cr}^{(t)}\big)$ then by Proposition C.1, there exists a formula $\varphi(x)$ in $\mathsf{GC}^{(t)}$ such that $[\![\varphi, v]\!]_G^\mathbb{B} = \top \neq [\![\varphi, w]\!]_H^\mathbb{B} = \bot$. We have just shown when we consider $\tilde{\varphi}$, in $\mathsf{GTL}^{(t)}$, also $[\![\tilde{\varphi}, v]\!]_G \neq [\![\tilde{\varphi}, w]\!]_H$ holds. Hence, $(G, v)$ and $(H, w)$

are not in $\rho_1\big(\mathsf{GTL}^{(t)}(\Omega)\big)$ either, for any $\Omega$. Hence, $\rho_1\big(\mathsf{GTL}^{(t)}(\Omega)\big) \subseteq \rho_1\big(\mathsf{cr}^{(t)}\big)$ holds. The other bullets are shown in the same way, again by relying on Proposition C.1 and using that we can move from $\mathsf{vwl}|_k^{(t)}$ and $\mathsf{gwl}|_k^{(t)}$ to logical formulae, and to expressions in $\mathsf{TL}_{k+1}^{(t)}$ and $\mathsf{TL}_{k+1}^{(t+k)}$, respectively, to separate $(G, v)$ from $(H, w)$ or $G$ from $H$, respectively. □

Theorems 4.1, 4.2, 4.3 and 4.4 now follow directly from Propositions C.4 and C.5.

## C.5 OTHER AGGREGATION FUNCTIONS

As is mentioned in the main paper, our upper bound results on the separation power of tensor languages (and hence also of GNNs represented in those languages) generalize easily when other aggregation functions than summation are used in $\mathsf{TL}$ expressions.

To clarify what we understand by an aggregation function, let us first recall the semantics of summation aggregation. Let $\varphi := \sum_{x_i} \varphi_1$, where $\sum_{x_i}$ represents summation aggregation, let $G = (V_G, E_G, \mathsf{col}_G)$ be a graph, and let $\nu$ be a valuation assigning index variables to vertices in $V_G$. The semantics is then given by:

$$[\![\textstyle\sum_{x_i}\varphi_1, \nu]\!]_G := \sum_{v \in V_G}[\![\varphi_1, \nu[x_i \mapsto v]]\!]_G,$$

as explained in Section 3. Semantically, we can alternatively view $\sum_{x_i} \varphi_1$ as a function which takes the sum of the elements in the following multiset of real values:

$$\{\!\{[\![\varphi_1, \nu[x_i \mapsto v]]\!]_G \mid v \in V_G\}\!\}.$$

One can now consider, more generally, an *aggregation function* $F$ as a function which assigns to any multiset of values in $\mathbb{R}$ a single real value. For example, $F$ could be $\mathsf{max}$, $\mathsf{min}$, $\mathsf{mean}$, .... Let $\Theta$ be such a collection of aggregation functions. We next incorporate general aggregation function in tensor language.

First, we extend the syntax of expressions in $\mathsf{TL}(\Omega)$ by generalizing the construct $\sum_{x_i}\varphi$ in the grammar of $\mathsf{TL}(\Omega)$ expression. More precisely, we define $\mathsf{TL}(\Omega, \Theta)$ as the class of expressions, formed just like tensor language expressions, but in which two additional constructs, *unconditional* and *conditional aggregation*, are allowed. For an aggregation function $F$ we define:

$$\mathsf{aggr}_{x_j}^F(\varphi) \quad \text{and} \quad \mathsf{aggr}_{x_j}^F\big(\varphi(x_j) \mid E(x_i, x_j)\big),$$

where in the latter construct (conditional aggregation) the expression $\varphi(x_j)$ represents a $\mathsf{TL}(\Omega, \Theta)$ expression whose only free variable is $x_j$. The intuition behind these constructs is that unconditional aggregation $\mathsf{aggr}_{x_j}^F(\varphi)$ allows for aggregating, using aggregate function $F$, over the values of $\varphi$ where $x_j$ ranges unconditionally over *all* vertices in the graph. In contrast, for conditional aggregation $\mathsf{aggr}_{x_j}^F\big(\varphi(x_j) \mid E(x_i, x_j)\big)$, aggregation by $F$ of the values of $\varphi(x_j)$ is conditioned on the neighbors of the vertex assigned to $x_i$. That is, the vertices for $x_j$ range only among the neighbors of the vertex assigned to $x_i$.

More specifically, the semantics of the aggregation constructs is defined as follows:

$$[\![\mathsf{aggr}_{x_j}^F(\varphi), \nu]\!]_G := F\big(\{\!\{[\![\varphi, \nu[x_j \mapsto v]]\!]_G \mid v \in V_G\}\!\}\big) \in \mathbb{R}.$$

$$[\![\mathsf{aggr}_{x_j}^F\big(\varphi(x_j) \mid E(x_i, x_j)\big), \nu]\!]_G := F\big(\{\!\{[\![\varphi, \nu[x_j \mapsto v]]\!]_G \mid v \in V_G, (\nu(x_i), v) \in E_G\}\!\}\big) \in \mathbb{R}.$$

We remark that we can also consider aggregations functions $F$ over multisets of values in $\mathbb{R}^\ell$ for some $\ell \in \mathbb{N}$. This requires extending the syntax with $\mathsf{aggr}_{x_j}^F(\varphi_1, \ldots, \varphi_\ell)$ for unconditional aggregation and with $\mathsf{aggr}_{x_j}^F\big(\varphi_1(x_j), \ldots, \varphi_\ell(x_j) \mid E(x_i, x_j)\big)$ for conditional aggregation. The semantics is as expected: $F\big(\{\!\{\big(([\![\varphi_1, \nu[x_j \mapsto v]]\!]_G, \ldots, [\![\varphi_\ell, \nu[x_j \mapsto v]]\!]_G\big) \mid v \in V_G\}\!\}\big) \in \mathbb{R}$ and $F\big(\{\!\{\big(([\![\varphi_1, \nu[x_j \mapsto v]]\!]_G, \ldots, [\![\varphi_\ell, \nu[x_j \mapsto v]]\!]_G\big) \mid v \in V_G, (\nu(x_i), v) \in E_G\}\!\}\big) \in \mathbb{R}$.

The need for considering conditional and unconditional aggregation separately is due to the use of arbitrary aggregation functions. Indeed, suppose that one uses an aggregation function $F$ for which $0 \in \mathbb{R}$ is a *neutral value*. That is, for any multiset $X$ of real values, the equality $F(X) = F(X \uplus \{0\})$ holds. For example, the summation aggregation function satisfies this property. We then observe:

$$[\![\mathsf{aggr}_{x_j}^F\big(\varphi(x_j) \mid E(x_i, x_j)\big), \nu]\!]_G = F\big(\{\!\{[\![\varphi, \nu[x_j \mapsto v]]\!] \mid v \in V_G, (\nu(x_i), v) \in E_G\}\!\}\big)$$

$$= F(\{\{[\![\varphi \cdot E(x_i, x_j), \nu[x_j \mapsto v]]\!] \mid v \in V_G\}\})$$
$$= [\![\mathsf{aggr}^F_{x_j}(\varphi(x_j) \cdot E(x_i, x_j)), \nu]\!]_G.$$

In other words, unconditional aggregation can simulate conditional aggregation. In contrast, when $0$ is not a neutral value of the aggregation function $F$, conditional and unconditional aggregation behave differently. Indeed, in such cases $\mathsf{aggr}^F_{x_j}(\varphi(x_j) \mid E(x_i, x_j))$ and $\mathsf{aggr}^F_{x_j}(\varphi(x_j) \cdot E(x_i, x_j))$ may evaluate to different values, as illustrated in the following example.

As aggregation function $F$ we take the average $\mathsf{avg}(X) := \frac{1}{|X|} \sum_{x \in X} x$ for multisets $X$ of real values. We remark that $0$'s in $X$ contribute to the size of $X$ and hence $0$ is *not* a neutral element of avg. Now, let us consider the expressions

$$\varphi_1(x_i) := \mathsf{aggr}^{\mathsf{avg}}_{x_j}(\mathbf{1}_{x_j = x_j} \cdot E(x_i, x_j)) \text{ and } \varphi_2(x_i) := \mathsf{aggr}^{\mathsf{avg}}_{x_j}(\mathbf{1}_{x_j = x_j} \mid E(x_i, x_j)).$$

Let $\nu$ be such that $\nu(x_i) = v$. Then, $[\![\varphi_1, \nu]\!]_G$ results in applying the average to the multiset $\{\{\mathbf{1}_{w=w} \cdot E(v, w) \mid w \in V_G\}\}$ which includes the value 1 for every $w \in N_G(v)$ and a 0 for every non-neighbor $w \notin N_G(v)$. In other words, $[\![\varphi_1, \nu]\!]_G$ results in $|N_G(v)|/|V_G|$. In contrast, $[\![\varphi_2, \nu]\!]_G$ results in applying the average to the multiset $\{\{\mathbf{1}_{w=w} \mid w \in V_G, (v, w) \in E_G\}\}$. In other words, this multiset only contains the value 1 for each $w \in N_G(v)$, ignoring any information about the non-neighbors of $v$. In other words, $[\![\varphi_2, \nu]\!]_G$ results in $|N_G(v)|/|N_G(v)| = 1$. Hence, conditional and unconditional aggregation behave differently for the average aggregation function.

This said, one could alternative use a more general variant of conditional aggregation of the form $\mathsf{aggr}^F_{x_j}(\varphi|\psi)$ with as semantics $[\![\mathsf{aggr}^F_{x_j}(\varphi|\psi), \nu]\!]_G := F(\{\{[\![\varphi, \nu[x_j \to v]]\!]_G \mid v \in V_G, [\![\psi, \nu[x_j \to v]]\!]_G \neq 0\}\})$ where one creates a multiset only for those valuations $\nu[x_j \to v]$ for which the condition $\psi$ evaluates to a non-zero value. This general form of aggregation includes conditional aggregation, by replacing $\psi$ with $E(x_i, x_j)$ and restricting $\varphi$, and unconditional aggregation, by replacing $\psi$ with the constant function 1, e.g., $\mathbf{1}_{x_j = x_j}$. In order not to overload the syntax of TL expressions, we will not discuss this general form of aggregation further.

The notion of free index variables for expressions in $\mathsf{TL}(\Omega, \Theta)$ is defined as before, where now $\mathsf{free}(\mathsf{aggr}^F_{x_j}(\varphi)) := \mathsf{free}(\varphi) \setminus \{x_j\}$, and where $\mathsf{free}(\mathsf{aggr}^F_{x_j}(\varphi(x_j) \mid E(x_i, x_j)) := \{x_i\}$ (recall that $\mathsf{free}(\varphi(x_j)) = \{x_j\}$ in conditional aggregation). Moreover, summation depth is replaced by the notion of *aggregation depth*, $\mathsf{agd}(\varphi)$, defined in the same way as summation depth except that $\mathsf{agd}(\mathsf{aggr}^F_{x_j}(\varphi)) := \mathsf{agd}(\varphi) + 1$ and $\mathsf{agd}(\mathsf{aggr}^F_{x_j}(\varphi(x_j) \mid E(x_i, x_j)) := \mathsf{agd}(\varphi) + 1$. Similarly, the fragments $\mathsf{TL}_k(\Omega, \Theta)$ and its aggregation depth restricted fragment $\mathsf{TL}^{(t)}_k(\Omega, \Theta)$ are defined as before, using aggregation depth rather than summation depth.

For the guarded fragment, $\mathsf{GTL}(\Omega, \Theta)$, expressions are now restricted such that aggregations must occur only in the form $\mathsf{aggr}^F_{x_j}(\varphi(x_j) \mid E(x_i, x_j))$, for $i, j \in [2]$. In other words, aggregation only happens on multisets of values obtained from neighboring vertices.

We now argue that our upper bound results on the separation power remain valid for the extension $\mathsf{TL}(\Omega, \Theta)$ of $\mathsf{TL}(\Omega)$ with arbitrary aggregation functions $\Theta$.

**Proposition C.6.** *We have the following inclusions: For any $t \geq 0$, any collection $\Omega$ of functions and any collection $\Theta$ of aggregation functions:*

- $\rho_1\big(\mathsf{cr}^{(t)}\big) \subseteq \rho_1\big(\mathsf{GTL}^{(t)}(\Omega, \Theta)\big)$;

- $\rho_1\big(\mathsf{vwl}^{(t)}_k\big) \subseteq \rho_1\big(\mathsf{TL}^{(t)}_{k+1}(\Omega, \Theta)\big)$; *and*

- $\rho_0\big(\mathsf{gwl}^{(t)}_k\big) \subseteq \rho_0\big(\mathsf{TL}^{(t+1)}_{k+1}(\Omega, \Theta)\big)$.

*Proof.* It suffices to show that Proposition C.3 also holds for expressions in the fragments of $\mathsf{TL}(\Omega, \Theta)$ considered. In particular, we only need to revise the case of summation aggregation (that is, $\varphi := \sum_{x_i} \varphi_1$) in the proof of Proposition C.3. Indeed, let us consider the more general case when one of the two aggregating functions are used.

- $\varphi := \mathsf{aggr}^F_{x_i}(\varphi_1)$. We then define

$$\tilde{\varphi}^c := \bigvee_{\ell \in \mathbb{N}} \bigvee_{(m_1,\ldots,m_\ell) \in \mathbb{N}^\ell} \bigvee_{(c,c_1,\ldots,c_\ell) \in \mathcal{C}(m_1,\ldots,m_\ell,F)} \bigwedge_{s=1}^{\ell} \exists^{=m_s} x_i \, \tilde{\varphi}_1^{c_s} \wedge \forall x_i \bigvee_{s=1}^{\ell} \tilde{\varphi}_1^{c_s},$$

where $\mathcal{C}(m_1,\ldots,m_\ell,F)$ now consists of all $(c,c_1,\ldots,c_\ell) \in \mathbb{R}^{\ell+1}$ such that

$$c = F\Big(\{\{\underbrace{c_1,\ldots,c_1}_{m_1 \text{ times}},\ldots,\underbrace{c_\ell,\ldots,c_\ell}_{m_\ell \text{ times}}\}\}\Big).$$

- $\varphi := \mathsf{aggr}^F_{x_i}(\varphi_1(x_i) \mid E(x_j,x_i))$. We then define

$$\tilde{\varphi}^c := \bigvee_{\ell \in \mathbb{N}} \bigvee_{(m_1,\ldots,m_\ell) \in \mathbb{N}^\ell} \bigvee_{(c,c_1,\ldots,c_\ell) \in \mathcal{C}(m_1,\ldots,m_\ell,F)} \exists^{=m} x_i \, E(x_j,x_i) \wedge$$

$$\bigwedge_{s=1}^{\ell} \exists^{=m_s} x_i \, E(x_j,x_i) \wedge \tilde{\varphi}_1^{c_s}(x_i)$$

where $\mathcal{C}(m_1,\ldots,m_\ell,F)$ again consists of all $(c,c_1,\ldots,c_\ell) \in \mathbb{R}^{\ell+1}$ such that

$$c = F\Big(\{\{\underbrace{c_1,\ldots,c_1}_{m_1 \text{ times}},\ldots,\underbrace{c_\ell,\ldots,c_\ell}_{m_\ell \text{ times}}\}\}\Big) \text{ and } m = \sum_{s=1}^{\ell} m_s.$$

It is readily verified that $[\![\mathsf{aggr}^F_{x_i}(\varphi),\boldsymbol{v}]\!]_G = c$ iff $[\![\tilde{\varphi}^c,\boldsymbol{v}]\!]_G^{\mathbb{B}} = \top$, and $[\![\mathsf{aggr}^F_{x_i}(\varphi(x_i) \mid E(x_j,x_i)),\boldsymbol{v}]\!]_G = c$ iff $[\![\tilde{\varphi}^c,\boldsymbol{v}]\!]_G^{\mathbb{B}} = \top$, as desired.

For the guarded case, we note that the expression $\tilde{\varphi}^c$ above yields a guarded expression as long conditional aggregation is used of the form $\mathsf{aggr}^F_{x_i}(\varphi(x_i) \mid E(x_j,x_i))$ with $i,j \in [2]$, so we can reuse the argument in the proof of Proposition C.3 for the guarded case. $\square$

We will illustrate later on (Section D) that this generalization allows for assessing the separation power of GNNs that use a variety of aggregation functions.

The choice of supported aggregation functions has, of course, an impact on the ability of $\mathsf{TL}(\Omega,\Theta)$ to match color refinement or the $k$-WL procedures in separation power. The same holds for GNNs, as shown by Xu et al. (2019). And indeed, the proof of Proposition C.5 relies on the presence of summation aggregation. We note that most lower bounds on the separation power of GNNs in terms of color refinement or the $k$-WL procedures assume summation aggregation since summation suffices to construct injective sum-decomposable functions on multisets (Xu et al., 2019; Zaheer et al., 2017), which are used to simulate color refinement and $k$-WL. A more in-depth analysis of lower bounding GNNs with less expressive aggregation functions, possibly using weaker versions of color refinement and $k$-WL is left as future work.

### C.6 GENERALIZATION TO GRAPHS WITH REAL-VALUED VERTEX LABELS

We next consider the more general setting in which $\mathsf{col}_G : V_G \to \mathbb{R}^\ell$ for some $\ell \in \mathbb{N}$. That is, vertices in a graph can carry real-valued vectors. We remark that no changes to neither the syntax nor the semantics of $\mathsf{TL}$ expressions are needed, yet note that $[\![P_s(x),\nu]\!]_G := \mathsf{col}_G(\nu)_s$ is now an element in $\mathbb{R}$ rather than 0 or 1, for each $s \in [\ell]$.

A first observation is that the color refinement and $k$-WL procedures treat each real value as a separate label. That is, two values that differ only by any small $\epsilon > 0$, are considered different. The proofs of Theorem 4.1, 4.2, 4.3 and 4.4 rely on connections between color refinement and $k$-WL and the finite variable logics $\mathsf{GC}$ and $\mathsf{C}^{k+1}$, respectively. In the discrete context, the unary predicates $P_s(x)$ used in the logical formulas indicate which label vertices have. That is, $[\![P_s,v]\!]_G^{\mathbb{B}} = \top$ iff $\mathsf{col}_G(v)_s = 1$. To accommodate for real values in the context of separation power, these logics now need to be able to differentiate between different labels, that is, different real numbers. We therefore

extend the unary predicates allowed in formulas. More precisely, for each dimension $s \in [\ell]$, we now have *uncountably* many predicates of the form $P_{s,r}$, one for each $r \in \mathbb{R}$. In any formula in GC or $\mathsf{C}^{k+1}$ only a finite number of such predicates may occur. The Boolean semantics of these new predicates is as expected:

$$[\![P_{s,r}(x), \nu]\!]_G^{\mathbb{B}} := \text{if } \mathsf{col}_G(\mu(x_i))_s = r \text{ then } \top \text{ else } \bot.$$

In other words, in our logics, we can now detect which real-valued labels vertices have. Although, in general, the introduction of infinite predicates may cause problems, we here consider a specific setting in which the vertices in a graph have a unique label. This is commonly assumed in graph learning. Given this, it is easily verified that all results in Section C.2 carry over, where all logics involved now use the unary predicates $P_{s,r}$ with $s \in [\ell]$ and $r \in \mathbb{R}$.

The connection between TL and logics also carries over. First, for Proposition C.3 we now need to connect TL expressions, that use a finite number of predicates $P_s$, for $s \in [\ell]$, with the extended logics having uncountably many predicates $P_{s,r}$, for $s \in [\ell]$ and $r \in \mathbb{R}$, at their disposal. It suffices to reconsider the case $\varphi(x_i) = P_s(x_i)$ in the proof of Proposition C.3. More precisely, $[\![P_s(x_i), \nu]\!]_G$ can now be an arbitrary value $c \in \mathbb{R}$. We now simply define $\tilde{\varphi}^c(x_i) := P_{s,c}(x_i)$. By definition $[\![P_s(x_i), \nu]\!]_G = c$ if and only if $[\![P_{s,c}(x_i), \nu]\!]_G^{\mathbb{B}} = \top$, as desired.

The proof for the extended version of proposition C.5 now needs a slightly different strategy, where we build the relevant TL formula after we construct the contrapositive of the Proposition. Let us first show how to construct a TL formula that is equivalent to a logical formula on any graph using only labels in a specific (finite) set $R$ of real numbers.

In other words, given a set $R$ of real values, we show that for any formula $\varphi(\boldsymbol{x}) \in \mathsf{C}^{k,(t)}$ using unary predicates $P_{s,r}$ such that $r \in R$, we can construct the desired $\hat{\varphi}$. As mentioned, we only need to reconsider the case $\varphi(x_i) := P_{s,r}(x_i)$. We define

$$\hat{\varphi} := \frac{1}{\prod_{r' \in R, r \neq r'} r - r'} \prod_{r' \in R, r \neq r'} (P_s(x_i) - r' \mathbf{1}_{x_i=x_i}).$$

Then, $[\![\hat{\varphi}, \nu]\!]_G$ evaluates to

$$\frac{\prod_{r' \in R, r \neq r'}(r - r')}{\prod_{r' \in R, r \neq r'}(r - r')} = \begin{cases} 1 & [\![P_{s,r}, \nu]\!] = \top \\ 0 & [\![P_{s,r}, \nu]\!] = \bot \end{cases}.$$

Indeed, if $[\![P_{s,r}, \nu]\!] = \top$, then $\mathsf{col}_G(v)_s = r$ and hence $[\![P_s, v]\!]_G = r$, resulting in the same nominator and denominator in the above fraction. If $[\![P_{s,r}, \nu]\!] = \bot$, then $\mathsf{col}_G(v)_s = r'$ for some value $r' \in R$ with $r \neq r'$. In this case, the nominator in the above fraction becomes zero. We remark that this revised construction still results in a guarded TL expression, when the input logical formula is guarded as well.

Coming back to the proof of the extended version of Proposition C.5, let us show the proof for the the fact that $\rho_1\big(\mathsf{GTL}^{(t)}(\Omega)\big) \subseteq \rho_1\big(\mathsf{cr}^{(t)}\big)$, the other two items being analogous. Assume that there is a pair $(G, v)$ and $(H, w)$ which is not in $\rho_1\big(\mathsf{cr}^{(t)}\big)$. Then, by Proposition C.1, applied on graphs with real-valued labels, there exists a formula $\varphi(x)$ in $\mathsf{GC}^{(t)}$ such that $[\![\varphi, v]\!]_G^{\mathbb{B}} = \top \neq [\![\varphi, w]\!]_H^{\mathbb{B}} = \bot$. We remark that $\varphi(x)$ uses finitely many $P_{s,r}$ predicates. Let $R$ be the set of real values used in both $G$ and $H$ (and $\varphi(x)$). We note that $R$ is finite. We invoke the construction sketched above, and obtain a formula $\hat{\varphi}$ in $\mathsf{GTL}^{(t)}$ such that $[\![\tilde{\varphi}, v]\!]_G \neq [\![\tilde{\varphi}, w]\!]_H$. Hence, $(G, v)$ and $(H, w)$ is not in $\rho_1\big(\mathsf{GTL}^{(t)}(\Omega)\big)$ either, for any $\Omega$, which was to be shown.

## D  DETAILS OF SECTION 5

We here provide some additional details on the encoding of layers of GNNs in our tensor languages, and how, as a consequence of our results from Section 4, one obtains a bound on their separation power. This section showcases that it is relatively straightforward to represent GNNs in our tensor languages. Indeed, often, a direct translation of the layers, as defined in the literature, suffices.

### D.1  COLOR REFINEMENT

We start with GNN architectures related to color refinement, or in other words, architectures which can be represented in our guarded tensor language.

**GraphSage.** We first consider a "basic" GNN, that is, an instance of GraphSage (Hamilton et al., 2017) in which sum aggregation is used. The initial features are given by $\boldsymbol{F}^{(0)} = (\boldsymbol{f}_1^{(0)}, \ldots, \boldsymbol{f}_{d_0}^{(0)})$ where $\boldsymbol{f}_i^{(0)} \in \mathbb{R}^{n \times 1}$ is a hot-one encoding of the $i$th vertex label in $G$. We can represent the initial embedding easily in $\mathsf{GTL}^{(0)}$, without the use of any summation. Indeed, it suffices to define $\varphi_i^{(0)}(x_1) := P_i(x_1)$ for $i \in [d_0]$. We have $F_{vj}^{(0)} = [\![\varphi_j^{(0)}, v]\!]_G$ for $j \in [d_0]$, and thus the initial features can be represented by simple expressions in $\mathsf{GTL}^{(0)}$.

Assume now, by induction, that we can also represent the features computed by a basic GNN in layer $t - 1$. That is, let $\boldsymbol{F}^{(t-1)} \in \mathbb{R}^{n \times d_{t-1}}$ be those features and for each $i \in [d_{t-1}]$ let $\varphi_i^{(t-1)}(x_1)$ be expressions in $\mathsf{GTL}^{(t-1)}(\sigma)$ representing them. We assume that, for each $i \in [d_{t-1}]$, $F_{vi}^{(t-1)} = [\![\varphi_i^{(t-1)}, v]\!]_G$. We remark that we assume that a summation depth of $t - 1$ is needed for layer $t - 1$.

Then, in layer $t$, a basic GNN computes the next features as

$$\boldsymbol{F}^{(t)} := \sigma \left( \boldsymbol{F}^{(t-1)} \cdot \boldsymbol{V}^{(t)} + \boldsymbol{A} \cdot \boldsymbol{F}^{(t-1)} \cdot \boldsymbol{W}^{(t)} + \boldsymbol{B}^{(t)} \right),$$

where $\boldsymbol{A} \in \mathbb{R}^{n \times n}$ is the adjacency matrix of $G$, $\boldsymbol{V}^{(t)}$ and $\boldsymbol{W}^{(t)}$ are weight matrices in $\mathbb{R}^{d_{t-1} \times d_t}$, $\boldsymbol{B}^{(t)} \in \mathbb{R}^{n \times d_t}$ is a (constant) bias matrix consist of $n$ copies of $\boldsymbol{b}^{(t)} \in \mathbb{R}^{d_t}$, and $\sigma$ is some activation function. We can simply use the following expressions $\varphi_j^{(t)}(x_1)$, for $j \in [d_t]$:

$$\sigma \left( \left( \sum_{i=1}^{d_{t-1}} V_{ij}^{(t)} \cdot \varphi_i^{(t-1)}(x_1) \right) + \sum_{x_2} \left( E(x_1, x_2) \cdot \left( \sum_{i=1}^{d_{t-1}} W_{ij}^{(t)} \cdot \varphi_i^{(t-1)}(x_2) \right) \right) + b_j^{(t)} \cdot \mathbf{1}_{x_1 = x_1} \right).$$

Here, $W_{ij}^{(t)}$, $V_{ij}^{(t)}$ and $b_j^{(t)}$ are real values corresponding the weight matrices and bias vector in layer $t$. These are expressions in $\mathsf{GTL}^{(t)}(\sigma)$ since the additional summation is guarded, and combined with the summation depth of $t - 1$ of $\varphi_i^{(t-1)}$, this results in a summation depth of $t$ for layer $t$. Furthermore, $F_{vi}^{(t)} = [\![\varphi_i^{(t)}, v]\!]_G$, as desired. If we denote by $\mathsf{bGNN}^{(t)}$ the class of $t$-layered basic GNNs, then our results imply

$$\rho_1\big(\mathsf{cr}^{(t)}\big) \subseteq \rho_1\big((\mathsf{GTL}^{(t)}(\Omega)\big) \subseteq \rho_1\big(\mathsf{bGNN}^{(t)}\big),$$

and thus the separation power of basic GNNs is bounded by the separation power of color refinement. We thus recover known results by Xu et al. (2019) and Morris et al. (2019).

Furthermore, if one uses a readout layer in basic GNNs to obtain a graph embedding, one typically applies a function $\mathsf{ro} : \mathbb{R}^{d_t} \to \mathbb{R}^{d_t}$ in the form of $\mathsf{ro}\big(\sum_{v \in V_G} \boldsymbol{F}_v^{(t)}\big)$, in which aggregation takes places over all vertices of the graph. This corresponds to an expression in $\mathsf{TL}_2^{(t+1)}(\sigma, \mathsf{ro})$: $\varphi_j := \mathsf{ro}_j\big(\sum_{x_1} \varphi_j^{(t-1)}(x_1)\big)$, where $\mathsf{ro}_j$ is the projection of the readout function on the $j$the coordinate. We note that this is indeed not a guarded expression anymore, and thus our results tell that

$$\rho_0\big(\mathsf{gcr}^{(t)}\big) \subseteq \rho_0\big(\mathsf{TL}_2^{(t+1)}(\Omega)\big) \subseteq \rho_0\big(\mathsf{bGNN}^{(t)} + \mathsf{readout}\big).$$

More generally, GraphSage allows for the use of general aggregation functions $F$ on the multiset of features of neighboring vertices. To cast the corresponding layers in $\mathsf{TL}(\Omega)$, we need to consider the extension $\mathsf{TL}(\Omega, \Theta)$ with an appropriate set $\Theta$ of aggregation functions, as described in Section C.5. In this way, we can represent layer $t$ by means of the following expressions $\varphi_j^{(t)}(x_1)$, for $j \in [d_t]$.

$$\sigma \left( \left( \sum_{i=1}^{d_{t-1}} V_{ij}^{(t)} \cdot \varphi_i^{(t-1)}(x_1) \right) + \sum_{i=1}^{d_{t-1}} W_{ij}^{(t)} \cdot \mathsf{aggr}_{x_2}^F \left( \varphi_i^{(t-1)}(x_2) \mid E(x_1, x_2) \right) + b_j^{(t)} \cdot \mathbf{1}_{x_1 = x_1} \right),$$

which is now an expression in $\mathsf{GTL}^{(t)}(\{\sigma\}, \Theta)$ and hence the bound in terms of $t$ iterations of color refinement carries over by Proposition C.6. Here, $\Theta$ simply consists of the aggregation functions used in the layers in GraphSage.

**GCNs.** *Graph Convolution Networks* (GCNs) (Kipf & Welling, 2017) operate alike basic GNNs except that a normalized Laplacian $\boldsymbol{D}^{-1/2}(\boldsymbol{I} + \boldsymbol{A})\boldsymbol{D}^{-1/2}$ is used to aggregate features, instead of the adjacency matrix $\boldsymbol{A}$. Here, $\boldsymbol{D}^{-1/2}$ is the diagonal matrix consisting of reciprocal of the square root of the vertex degrees in $G$ plus 1. The initial embedding $\boldsymbol{F}^{(0)}$ is just as before. We

use again $d_t$ to denote the number of features in layer $t$. In layer $t > 0$, a GCN computes $\boldsymbol{F}^{(t)} := \sigma(\boldsymbol{D}^{-1/2}(\boldsymbol{I} + \boldsymbol{A})\boldsymbol{D}^{-1/2} \cdot \boldsymbol{F}^{(t-1)}\boldsymbol{W}^{(t)} + \boldsymbol{B}^{(t)})$. If, in addition to the activation function $\sigma$ we add the function $\frac{1}{\sqrt{x+1}} : \mathbb{R} \to \mathbb{R} : x \mapsto \frac{1}{\sqrt{x+1}}$ to $\Omega$, we can represent the GCN layer, as follows. For $j \in [d_t]$, we define the $\mathsf{GTL}^{(t+1)}(\sigma, \frac{1}{\sqrt{x+1}})$ expressions

$$\varphi_j^{(t)}(x_1) := \sigma\left(f_{1/\sqrt{x+1}}\left(\sum_{x_2} E(x_1, x_2)\right) \cdot \left(\sum_{i=1}^{d_{t-1}} W_{ij}^{(t)} \cdot \varphi_i^{(t-1)}(x_1)\right) \cdot f_{1/\sqrt{x+1}}\left(\sum_{x_2} E(x_1, x_2)\right)\right.$$

$$\left. + f_{1/\sqrt{x+1}}\left(\sum_{x_2} E(x_1, x_2)\right) \cdot \left(\sum_{x_2} E(x_1, x_2) \cdot f_{1/\sqrt{x+1}}\left(\sum_{x_1} E(x_2, x_1)\right) \cdot \left(\sum_{i=1}^{d_{t-1}} W_{ij}^{(t)} \cdot \varphi_i^{(t-1)}(x_2)\right)\right)\right),$$

where we omitted the bias vector for simplicity. We again observe that only guarded summations are needed. However, we remark that in every layer we now add two the overall summation depth, since we need an extra summation to compute the degrees. In other words, a $t$-layered GCN correspond to expressions in $\mathsf{GTL}^{(2t)}(\sigma, \frac{1}{\sqrt{x+1}})$. If we denote by $\mathsf{GCN}^{(t)}$ the class of $t$-layered GCNs, then our results imply

$$\rho_1\big(\mathsf{cr}^{(2t)}\big) \subseteq \rho_1\big(\mathsf{GTL}^{(2t)}(\Omega)\big) \subseteq \rho_1\big(\mathsf{GCN}^{(t)}\big).$$

We remark that another representation can be provided, in which the degree computation is factored out (Geerts et al., 2021a), resulting in a better upper bound $\rho_1\big(\mathsf{cr}^{(t+1)}\big) \subseteq \rho_1\big(\mathsf{GCN}^{(t)}\big)$. In a similar way as for basic GNNs, we also have $\rho_0\big(\mathsf{gcr}^{(t+1)}\big) \subseteq \rho_0\big(\mathsf{GCN}^{(t)} + \mathsf{readout}\big)$.

**SGCs.** As an other example, we consider a variation of *Simple Graph Convolutions* (SGCs) (Wu et al., 2019), which use powers the adjacency matrix and only apply a non-linear activation function at the end. That is, $\boldsymbol{F} := \sigma(\boldsymbol{A}^p \cdot \boldsymbol{F}^{(0)} \cdot \boldsymbol{W})$ for some $p \in \mathbb{N}$ and $\boldsymbol{W} \in \mathbb{R}^{d_0 \times d_1}$. We remark that SGCs actually use powers of the normalized Laplacian, that is, $\boldsymbol{F} := \sigma\big((\boldsymbol{D}^{-1/2}(\boldsymbol{I} + \boldsymbol{A}_G)\boldsymbol{D}^{-1/2}))^p \cdot \boldsymbol{F}^{(0)} \cdot \boldsymbol{W}\big)$ but this only incurs an additional summation depth as for GCNs. We focus here on our simpler version. It should be clear that we can represent the architecture in $\mathsf{TL}_{p+1}^{(p)}(\Omega)$ by means of the expressions:

$$\varphi_j^{(t)}(x_1) := \sigma\left(\sum_{x_2} \cdots \sum_{x_{p+1}} \prod_{k=1}^{p} E(x_k, x_{k+1}) \cdot \left(\sum_{i=1}^{d_0} W_{ij} \cdot \varphi_i^{(0)}(x_{p+1})\right)\right),$$

for $j \in [d_1]$. A naive application of our results would imply an upper bound on their separation power by $p$-WL. We can, however, use Proposition 4.5. Indeed, it is readily verified that these expressions have a treewidth of one, because the variables form a path. And indeed, when for example, $p = 3$, we can equivalently write $\varphi_j^{(t)}(x_1)$ as

$$\sigma\left(\sum_{x_2} E(x_1, x_2) \cdot \left(\sum_{x_1} E(x_2, x_1) \cdot \left(\sum_{x_2} E(x_1, x_2) \cdot \left(\sum_{i=1}^{d_0} W_{ij} \cdot \varphi_i^{(0)}(x_2)\right)\right)\right)\right),$$

by reordering the summations and reusing index variables. This holds for arbitrary $p$. We thus obtain guarded expressions in $\mathsf{GTL}^{(p)}(\sigma)$ and our results tell that $t$-layered SGCs are bounded by $\mathsf{cr}^{(p)}$ for vertex embeddings, and by $\mathsf{gcr}^{(p)}$ for SGCs + readout.

**Principal Neighbourhood Aggregation.** Our next example is a GNN in which different aggregation functions are used: *Principal Neighborhood Aggregation* (PNA) is an architecture proposed by Corso et al. (2020) in which aggregation over neighboring vertices is done by means of mean, stdv, max and min, and this in parallel. In addition, after aggregation, three different *scalers* are applied. Scalers are diagonal matrices whose diagonal entries are a function of the vertex degrees. Given the features for each vertex $v$ computed in layer $t - 1$, that is, $\boldsymbol{F}_{v:}^{(t-1)} \in \mathbb{R}^{1 \times \ell}$, a PNA computes $v$'s new features in layer $t$ in the following way (see layer definition (8) in (Corso et al., 2020)). First, vectors $\boldsymbol{G}_{v:}^{(t)} \in \mathbb{R}^{1 \times 4\ell}$ are computed such that

$$G_{vj}^{(t)} = \begin{cases} \mathsf{mean}\left(\{\!\{\mathsf{mlp}_j(F_{w:}^{(t-1)}) \mid w \in N_G(v)\}\!\}\right) & \text{for } 1 \leq j \leq \ell \\ \mathsf{stdv}\left(\{\!\{\mathsf{mlp}_j(F_{w:}^{(t-1)}) \mid w \in N_G(v)\}\!\}\right) & \text{for } \ell + 1 \leq j \leq 2\ell \\ \mathsf{max}\left(\{\!\{\mathsf{mlp}_j(F_{w:}^{(t-1)}) \mid w \in N_G(v)\}\!\}\right) & \text{for } 2\ell + 1 \leq j \leq 3\ell \\ \mathsf{min}\left(\{\!\{\mathsf{mlp}_j(F_{w:}^{(t-1)}) \mid w \in N_G(v)\}\!\}\right) & \text{for } 3\ell + 1 \leq j \leq 4\ell, \end{cases}$$

where $\mathsf{mlp}_j : \mathbb{R}^\ell \to \mathbb{R}$ is the projection of an MLP $\mathsf{mlp} : \mathbb{R}^\ell \to \mathbb{R}^\ell$ on the $j$th coordinate. Then, three different scalers are applied. The first scaler is simply the identity, the second two scalers $s_1$ and $s_2$ depend on the vertex degrees. As such, vectors $\boldsymbol{H}^{(t)}_{v:} \in \mathbb{R}^{12\ell}$ are constructed as follows:

$$H^{(t)}_{vj} = \begin{cases} H^{(t)}_{vj} & \text{for } 1 \le j \le 4\ell \\ s_1(\deg_G(v)) \cdot H^{(t)}_{vj} & \text{for } 4\ell + 1 \le j \le 8\ell \\ s_2(\deg_G(v)) \cdot H^{(t)}_{vj} & \text{for } 8\ell + 1 \le j \le 12\ell, \end{cases}$$

where $s_1$ and $s_2$ are functions from $\mathbb{R} \to \mathbb{R}$ (see (Corso et al., 2020) for details). Finally, the new vertex embedding is obtained as

$$\boldsymbol{F}^{(t)}_{v:} = \mathsf{mlp}'(\boldsymbol{H}^{(t)}_{v:})$$

for some MLP $\mathsf{mlp}' : \mathbb{R}^{12\ell} \to \mathbb{R}^\ell$. The above layer definition translates naturally into expressions in $\mathsf{TL}(\Omega, \Theta)$, the extension of $\mathsf{TL}(\Omega)$ with aggregate functions (Section C.5). Indeed, suppose that for each $j \in [\ell]$ we have $\mathsf{TL}(\Omega, \Theta)$ expressions $\varphi^{(t-1)}_j(x_1)$ such that $[\![\varphi^{(t-1)}_j, v]\!]_G = F^{(t-1)}_{vj}$ for any vertex $v$. Then, $G^{(t)}_{vj}$ simply corresponds to the guarded expressions

$$\psi^{(t)}_j(x_1) := \mathsf{aggr}^{\mathsf{mean}}_{x_2}(\mathsf{mlp}_j(\varphi^{(t-1)}_1(x_2), \ldots, \varphi^{(t-1)}_\ell(x_2)) \mid E(x_1, x_2)),$$

for $1 \le j \le \ell$, and similarly for the other components of $G^{(t)}_{v:}$ using the respective aggregation functions, $\mathsf{stdv}$, $\mathsf{max}$ and $\mathsf{min}$. Then, $H^{(t)}_{vj}$ corresponds to

$$\xi^{(t)}_j(x_1) = \begin{cases} \psi^{(t)}_j(x_1) & \text{for } 1 \le j \le 4\ell \\ s_1(\mathsf{aggr}^{\mathsf{sum}}_{x_2}(\mathbf{1}_{x_2=x_2} \mid E(x_1, x_2))) \cdot \psi^{(t)}_j(x_1) & \text{for } 4\ell + 1 \le j \le 8\ell \\ s_2(\mathsf{aggr}^{\mathsf{sum}}_{x_2}(\mathbf{1}_{x_2=x_2} \mid E(x_1, x_2))) \cdot \psi^{(t)}_j(x_1) & \text{for } 8\ell + 1 \le j \le 12\ell, \end{cases}$$

where we use summation aggregation to compute the degree information used in the functions in the scalers $s_1$ and $s_2$. And finally,

$$\varphi^{(t)}_j := \mathsf{mlp}'_j(\xi^{(t)}_1(x_1), \ldots, \xi^{(t)}_{12\ell}(x_1))$$

represents $F^{(t)}_{vj}$. We see that all expressions only use two index variables and aggregation is applied in a guarded way. Furthermore, in each layer, the aggregation depth increases with one. As such, a $t$-layered PNA can be represented in $\mathsf{GTL}^{(t)}(\Omega, \Theta)$, where $\Omega$ consists of the MLPs and functions used in scalers, and $\Theta$ consists of $\mathsf{sum}$ (for computing vertex degrees), and $\mathsf{mean}$, $\mathsf{stdv}$, $\mathsf{max}$ and $\mathsf{min}$. Proposition C.6 then implies a bound on the separation power by $\mathsf{cr}^{(t)}$.

**Other example.** In the same way, one can also easily analyze GATs (Velickovic et al., 2018) and show that these can be represented in $\mathsf{GTL}(\Omega)$ as well, and thus bounds by color refinement can be obtained.

### D.2 $k$-DIMENSIONAL WEISFEILER-LEMAN TESTS

We next discuss architectures related to the $k$-dimensional Weisfeiler-Leman algorithms. For $k = 1$, we discussed the extended GINs in the main paper. We here focus on arbitrary $k \ge 2$.

**Folklore GNNs.** We first consider the "Folklore" GNNs or $k$-FGNNs for short (Maron et al., 2019b). For $k \ge 2$, $k$-FGNNs computes a tensors. In particular, the initial tensor $\mathbf{F}^{(0)}$ encodes $\mathsf{atp}_k(G, \boldsymbol{v})$ for each $\boldsymbol{v} \in V^k_G$. We can represent this tensor by the following $k^2(\ell + 2)$ expressions in $\mathsf{TL}^{(0)}_k$:

$$\varphi^{(0)}_{r,s,j}(x_1, \ldots, x_k) := \begin{cases} \mathbf{1}_{x_r=x_s} \cdot P_j(x_r) & \text{for } j \in [\ell] \\ E(x_r, x_s) & \text{for } j = \ell + 1 \\ \mathbf{1}_{x_r=x_s} & \text{for } j = \ell + 2 \end{cases},$$

for $r, s \in [k]$ and $j \in [\ell + 2]$. We note: $[\![\varphi^{(0)}_{r,s,j}, (v_1, \ldots, v_k)]\!]_G = F^{(0)}_{v_1,\ldots,v_k,r,s,j}$ for all $(r, s, j) \in [k]^2 \times [\ell + 2]$, as desired. We let $\tau_0 := [k]^2 \times [\ell + 2]$ and set $d_0 = k^2 \times (\ell + 2)$.

Then, in layer $t$, a $k$-FGNN computes a tensor

$$\mathbf{F}^{(t)}_{v_1,\ldots,v_k,\bullet} := \mathsf{mlp}^{(t)}_0\Big(\mathbf{F}^{(t-1)}_{v_1,\ldots,v_k,\bullet}, \sum_{w \in V_G} \prod_{s=1}^{k} \mathsf{mlp}^{(t)}_s(\mathbf{F}^{(t-1)}_{v_1,\ldots,v_{s-1},w,v_{s+1},\ldots,v_k,\bullet})\Big),$$

where $\mathsf{mlp}_s^{(t)} : \mathbb{R}^{d_{t-1}} \to \mathbb{R}^{d_t'}$, for $s \in [k]$, and and $\mathsf{mlp}_0^{(t)} : \mathbb{R}^{d_{t-1} \times d_t'} \to \mathbb{R}^{d_t}$ are MLPs. We here use $\bullet$ to denote combinations of indices in $\tau_d$ for $\mathbf{F}^{(t)}$ and in $\tau_{d-1}$ for $\mathbf{F}^{(t-1)}$.

Let $\mathbf{F}^{(t-1)} \in \mathbb{R}^{n^k \times d_{t-1}}$ be the tensor computed by an $k$-FGNN in layer $t-1$. Assume that for each tuple of elements $\boldsymbol{j}$ in $\tau_{d_{t-1}}$ we have an expression $\varphi_{\boldsymbol{j}}^{(t-1)}(x_1, \ldots, x_k)$ satisfying $[\![\varphi_{\boldsymbol{j}}^{(t-1)}, (v_1, \ldots, v_k)]\!]_G = F_{v_1, \ldots, v_k, \boldsymbol{j}}^{(t-1)}$ and such that it is an expression in $\mathsf{TL}_{k+1}^{(t-1)}(\Omega)$. That is, we need $k+1$ index variables and a summation depth of $t-1$ to represent layer $t-1$.

Then, for layer $t$, for each $\boldsymbol{j} \in \tau_{d_t}$, it suffices to consider the expression

$$\varphi_{\boldsymbol{j}}^{(t)}(x_1, \ldots, x_k) := \mathsf{mlp}_{0,\boldsymbol{j}}^{(t)}\Big(\big(\varphi_{\boldsymbol{i}}^{(t-1)}(x_1, \ldots, x_k)\big)_{\boldsymbol{i} \in \tau_{d_{t-1}}},$$

$$\sum_{x_{k+1}} \prod_{s=1}^{k} \mathsf{mlp}_{s,\boldsymbol{j}}^{(t)}\big((\varphi_{\boldsymbol{i}}^{(t-1)}(x_1, \ldots, x_{s-1}, x_{k+1}, x_{s+1}, \ldots, x_k))_{\boldsymbol{i} \in \tau_{d_{t-1}}}\big)\Big),$$

where $\mathsf{mlp}_{o,\boldsymbol{j}}^{(t)}$ and $\mathsf{mlp}_{s,\boldsymbol{j}}^{(t)}$ are the projections of the MLPs on the $\boldsymbol{j}$-coordinates. We remark that we need $k+1$ index variables, and one extra summation is needed. We thus obtain expressions in $\mathsf{TL}_{k+1}^{(t)}(\Omega)$ for the $t$th layer, as desired. We remark that the expressions are simple translations of the defining layer definitions. Also, in this case, $\Omega$ consists of all MLPs. When a $k$-FGNN is used for vertex embeddings, we now simply add to each expression a factor $\prod_{s=1}^{k} \mathbf{1}_{x_1 = x_s}$. As an immediate consequence of our results, if we denote by $k$-FGNN$^{(t)}$ the class of $t$-layered $k$-FGNNs, then for vertex embeddings:

$$\rho_1\big(\mathsf{vwl}_k^{(t)}\big) \subseteq \rho_1\big(\mathsf{TL}_{k+1}^{(t)}(\Omega)\big) \subseteq \rho_1\big(k\text{-FGNN}^{(t)}\big)$$

in accordance with the known results from Azizian & Lelarge (2021). When used for graph embeddings, an aggregation layer over all $k$-tuples of vertices is added, followed by the application of an MLP. This results in expressions with no free index variables, and of summation depth $t+k$, where the increase with $k$ stems from the aggregation process over all $k$-tuples. In view of our results, for graph embeddings:

$$\rho_0\big(\mathsf{gwl}_k^{(\infty)}\big) \subseteq \rho_0\big(\mathsf{TL}_{k+1}(\Omega)\big) \subseteq \rho_0\big(k\text{-FGNN}\big)$$

in accordance again with Azizian & Lelarge (2021). We here emphasize that the upper bounds in terms of $k$-WL are obtained without the need to know how $k$-WL works. Indeed, one can really just focus on casting layers in the right tensor language!

We remark that Azizian & Lelarge (2021) define vertex embedding $k$-FGNNs in a different way. Indeed, for a vertex $v$, its embedding is obtained by aggregating of all $(k-1)$ tuples in the remaining coordinates of the tensors. They define $\mathsf{vwl}_k$ accordingly. From the tensor language point of view, this corresponds to the addition of $k-1$ to the summation depth. Our results indicate that we loose the connection between rounds and layers, as in Azizian & Lelarge (2021). This is the reason why we defined vertex embedding $k$-FGNNs in a different way and can ensure a correspondence between rounds and layers for vertex embeddings.

**Other higher-order examples.** It is readily verified that $t$-layered $k$-GNNs (Morris et al., 2019) can be represented in $\mathsf{TL}_{k+1}^{(t)}(\Omega)$, recovering the known upper bound by $\mathsf{vwl}_k^{(t)}$ (Morris et al., 2019). It is an equally easy exercise to show that 2-WL-convolutions (Damke et al., 2020) and Ring-GNNs (Chen et al., 2019) are bounded by 2-WL, by simply writing their layers in $\mathsf{TL}_3(\Omega)$. The invariant graph networks ($k$-IGNs) (Maron et al., 2019b) will be treated in Section E, as their representation in $\mathsf{TL}_{k+1}(\Omega)$ requires some work.

### D.3 AUGMENTED GNNS

Higher-order GNN architectures such as $k$-GNNs, $k$-FGNNs and $k$-IGNs, incur a substantial cost in terms of memory and computation (Morris et al., 2020). Some recent proposals infuse more efficient GNNs with higher-order information by means of some pre-processing step. We next show that the tensor language approach also enables to obtain upper bounds on the separation power of such "augmented" GNNs.

We first consider $\mathcal{F}$-MPNNs (Barceló et al., 2021) in which the initial vertex features are augmented with *homomorphism counts* of rooted graph patterns. More precisely, let $P^r$ be a connected rooted

graph (with root vertex $r$), and consider a graph $G = (V_G, E_G, \mathsf{col}_G)$ and vertex $v \in V_G$. Then, $\mathsf{hom}(P^r, G^v)$ denotes the number of homomorphism from $P$ to $G$, mapping $r$ to $v$. We recall that a homomorphism is an edge-preserving mapping between vertex sets. Given a collection $\mathcal{F} = \{P_1^r, \ldots, P_\ell^r\}$ of rooted patterns, an $\mathcal{F}$-MPNN runs an MPNN on the augmented initial vertex features:

$$\tilde{\boldsymbol{F}}_{v:}^{(0)} := (\boldsymbol{F}_{v:}^{(0)}, \mathsf{hom}(P_1^r, G^v), \ldots, \mathsf{hom}(P_\ell^r, G^v)).$$

Now, take any GNN architecture that can be cast in $\mathsf{GTL}(\Omega)$ or $\mathsf{TL}_2(\Omega)$ and assume, for simplicity of exposition, that a $t$-layer GNN corresponds to expressions in $\mathsf{GTL}^{(t)}(\Omega)$ or $\mathsf{TL}_2^{(t)}(\Omega)$. In order to analyze the impact of the augmented features, one only needs to revise the expressions $\varphi_j^{(0)}(x_1)$ that represent the initial features. In the absence of graph patterns, $\varphi_j^{(0)}(x_1) := P_j(x_1)$, as we have seen before. By contrast, to represent $\tilde{\boldsymbol{F}}_{vj}^{(0)}$ we need to cast the computation of $\mathsf{hom}(P_i^r, G^v)$ in $\mathsf{TL}$. Assume that the graph pattern $P_i$ consists of $p$ vertices and let us identify the vertex set with $[p]$. Furthermore, without of loss generality, we assume that vertex "1" is the root vertex in $P_i$. To obtain $\mathsf{hom}(P_i^r, G^v)$ we need to create an indicator function for the graph pattern $P_i$ and then count how many times this indicator value is equal to one in $G$. The indicator function for $P_i$ is simply given by the expression $\prod_{uv \in E_{P_i}} E(x_u, x_v)$. Then, counting just pours down to summation over all index variables except the one for the root vertex. More precisely, if we define

$$\varphi_{P_i}(x_1) := \sum_{x_2} \cdots \sum_{x_p} \prod_{uv \in E_{P_i}} E(x_u, x_v),$$

then $\llbracket \varphi_{P_i}, v \rrbracket_G = \mathsf{hom}(P_i^r, G^v)$. This encoding results in an expression in $\mathsf{TL}_p$. However, it is well-known that we can equivalently write $\varphi_{P_i}(x_1)$ as an expression $\tilde{\varphi}_{P_i}(x_1)$ in $\mathsf{TL}_{k+1}$ where $k$ is the treewidth of the graph $P_i$. As such, our results imply that $\mathcal{F}$-MPNNs are bounded in separation power by $k$-WL where $k$ is the maximal treewidth of graphs in $\mathcal{F}$. We thus recover the known upper bound as given in Barceló et al. (2021) using our tensor language approach.

Another example of augmented GNN architectures are the *Graph Substructure Networks* (GSNs) (Bouritsas et al., 2020). By contrast to $\mathcal{F}$-MPNNs, subgraph isomorphism counts rather than homomorphism counts are used to augment the initial features. At the core of a GSN thus lies the computation of $\mathsf{sub}(P^r, G^v)$, the number of subgraphs $H$ in $G$ isomorphic to $P$ (and such that the isomorphisms map $r$ to $v$). In a similar way as for homomorphisms counts, we can either directly cast the computation of $\mathsf{sub}(P^r, G^v)$ in $\mathsf{TL}$ resulting again in the use of $p$ index variables. A possible reduction in terms of index variables, however, can be obtained by relying on the result (Theorem 1.1.) by Curticapean et al. (2017) in which it shown that $\mathsf{sub}(P^r, G^v)$ can be computed in terms of homomorphism counts of graph patterns derived from $P^r$. More precisely, Curticapean et al. (2017) define $\mathsf{spasm}(P^r)$ as the set of graphs consisting of all possible homomorphic images of $P^r$. It is then readily verified that if the maximal treewidth of the graphs in $\mathsf{spasm}(P^r)$ is $k$, then $\mathsf{sub}(P^r, G^v)$ can be cast as an expression in $\mathsf{TL}_{k+1}$. Hence, GSNs using a pattern collection $\mathcal{F}$ can be represented in $\mathsf{TL}_{k+1}$, where $k$ is the maximal treewidth of graphs in any of the spams of patterns in $\mathcal{F}$, and thus are bounded in separation power $k$-WL in accordance to the results by Barceló et al. (2021).

As a final example, we consider the recently introduced *Message Passing Simplicial Networks* (MPSNs) (Bodnar et al., 2021). In a nutshell, MPSNs are run on simplicial complexes of graphs instead of on the original graphs. We sketch how our tensor language approach can be used to assess the separation power of MPSNs on *clique complexes*. We use the simplified version of MPSNs which have the same expressive power as the full version of MPSNs (Theorem 6 in Bodnar et al. (2021)).

We recall some definitions. Let $\mathsf{Cliques}(G)$ denote the set of all cliques in $G$. Given two cliques $c$ and $c'$ in $\mathsf{Cliques}(G)$, define $c \prec c'$ if $c \subset c'$ and there exists no $c''$ in $\mathsf{Cliques}(G)$, such that $c \subset c'' \subset c'$. We define $\mathsf{Boundary}(c, G) := \{c' \in \mathsf{Cliques}(G) \mid c' \prec c\}$ and $\mathsf{Upper}(c, G) := \{c' \in \mathsf{Cliques}(G) \mid \exists c'' \in \mathsf{Cliques}(G), c' \prec c'' \text{ and } c \prec c''\}$.

For each $c$ in $\mathsf{Cliques}(G)$ we have an initial feature vector $\boldsymbol{F}_{c:}^{(0)} \in \mathbb{R}^{1 \times \ell}$. Bodnar et al. (2021) initialize all initial features with the same value. Then, in layer $t$, for each $c \in \mathsf{Cliques}(G)$, features are updated as follows:

$$\boldsymbol{G}_{c:}^{(t)} = F_B(\{\{\mathsf{mlp}_B(\boldsymbol{F}_{c:}^{(t-1)}, \boldsymbol{F}_{c':}^{(t-1)}) \mid c' \in \mathsf{Boundary}(G, c)\}\})$$

$$\boldsymbol{H}_{c:}^{(t)} = F_U(\{\{\mathsf{mlp}_U(\boldsymbol{F}_{c:}^{(t-1)}, \boldsymbol{F}_{c':}^{(t-1)}, \boldsymbol{F}_{c \cup c':}^{(t-1)}) \mid c' \in \mathsf{Upper}(G, c)\}\})$$
$$\boldsymbol{F}_{c:}^{(t)} = \mathsf{mlp}(\boldsymbol{F}_{c:}^{(t-1)}, \boldsymbol{G}_{c:}^{(t)}, \boldsymbol{H}_{c:}^{(t)}),$$

where $F_B$ and $F_U$ are aggregation functions and $\mathsf{mlp}_B$, $\mathsf{mlp}_U$ and $\mathsf{mlp}$ are MLPs. With some effort, one can represent these computations by expressions in $\mathsf{TL}_p(\Omega, \Theta)$ where $p$ is largest clique in $G$. As such, the separation power of clique-complex MPSNs on graphs of clique size at most $p$ is bounded by $p - 1$-WL. And indeed, Bodnar et al. (2021) consider Rook's $4 \times 4$ graph, which contains a 4-clique, and the Shirkhande graph, which does not contain a 4-clique. As such, the analysis above implies that clique-complex MPSNs are bounded by 2-WL on the Shrikhande graph, and by 3-WL on Rook's graph, consistent with the observation in Bodnar et al. (2021). A more detailed analysis of MPSNs in terms of summation depth and for other simplicial complexes is left as future work.

This illustrates again that our approach can be used to assess the separation power of a variety of GNN architectures in terms of $k$-WL, by simply writing them as tensor language expressions. Furthermore, bounds in terms of $k$-WL can be used for augmented GNNs which form a more efficient way of incorporating higher-order graph structural information than higher-order GNNs.

## D.4 SPECTRAL GNNS

In general, spectral GNNs are defined in terms of eigenvectors and eigenvalues of the (normalized) graph Laplacian (Bruna et al., 2014; Defferrard et al., 2016; Levie et al., 2019; Balcilar et al., 2021b)). The diagonalization of the graph Laplacian is, however, avoided in practice, due to its excessive cost. Instead, by relying on approximation results in spectral graph analysis (Hammond et al., 2011), the layers of practical spectral GNNs are defined in term propagation matrices consisting of functions, which operate directly on the graph Laplacian. This viewpoint allows for a spectral analysis of spectral *and* "spatial" GNNs in a uniform way, as shown by Balcilar et al. (2021b). In this section, we consider two specific instances of spectral GNNs: ChebNet (Defferrard et al., 2016) and CayleyNet (Levie et al., 2019), and assess their separation power in terms of tensor logic. Our general results then provide bounds on their separation power in terms color refinement and 2-WL, respectively.

**Chebnet.** The separation power of ChebNet (Defferrard et al., 2016) was already analyzed in Balcilar et al. (2021a) by representing them in the MATLANG matrix query language (Brijder et al., 2019). It was shown (Theorem 2 (Balcilar et al., 2021a)) that it is only the maximal eigenvalue $\lambda_{\max}$ of the graph Laplacian used in the layers of ChebNet that may result in the separation power of ChebNet to go beyond 1-WL. We here revisit and refine this result by showing that, when ignoring the use of $\lambda_{\max}$, the separation power of Chebnet is bounded already by color refinement (which, as mentioned in Section 2, is weaker than 1-WL for vertex embeddings). In a nutshell, the layers of a ChebNet are defined in terms of Chebyshev polynomials of the normalized Laplacian $\boldsymbol{L}_{norm} = \boldsymbol{I} - \boldsymbol{D}^{-1/2} \cdot \boldsymbol{A} \cdot \boldsymbol{D}^{-1/2}$ and these polynomials can be easily represented in $\mathsf{GTL}(\Omega)$. One can alternatively use the graph Laplacian $\boldsymbol{L} = \boldsymbol{D} - \boldsymbol{A}$ in a ChebNet, which allows for a similar analysis. The distinction between the choice of $\boldsymbol{L}_{norm}$ and $\boldsymbol{L}$ only shows in the needed summation depth (in as in similar way as for the GCNs described earlier). We only consider the normalized Laplacian here.

More precisely, following Balcilar et al. (2021a;b), in layer $t$, vertex embeddings are updated in a ChebNet according to:

$$\boldsymbol{F}^{(t)} := \sigma\left(\sum_{s=1}^{p} \boldsymbol{C}^{(s)} \cdot \boldsymbol{F}^{(t-1)} \cdot \boldsymbol{W}^{(t-1,s)}\right),$$

with

$$\boldsymbol{C}^{(1)} := \boldsymbol{I}, \boldsymbol{C}^{(2)} = \frac{2}{\lambda_{\max}} \boldsymbol{L}_{norm} - \boldsymbol{I}, \boldsymbol{C}^{(s)} = 2\boldsymbol{C}^{(2)} \cdot \boldsymbol{C}^{(s-1)} - \boldsymbol{C}^{(s-2)}, \text{ for } s \geq 3,$$

and where $\lambda_{\max}$ denotes the maximum eigenvalue of $\boldsymbol{L}_{norm}$. We next use a similar analysis as in Balcilar et al. (2021a). That is, we ignore for the moment the maximal eigenvalue $\lambda_{\max}$ and redefine $\boldsymbol{C}^{(2)}$ as $c\boldsymbol{L}_{norm} - \boldsymbol{I}$ for some constant $c$. We thus see that each $\boldsymbol{C}^{(s)}$ is a polynomial of the form $p_s(c, \boldsymbol{L}_{norm}) := \sum_{i=0}^{q_s} a_i^{(s)}(c) \cdot (\boldsymbol{L}_{norm})^i$ with scalar functions $a_i^{(s)} : \mathbb{R} \to \mathbb{R}$ and where we interpret $(\boldsymbol{L}_{norm})^0 = \boldsymbol{I}$. To upper bound the separation power using our tensor language approach,

we can thus shift our attention entirely to representing $(\boldsymbol{L}_{norm})^i \cdot \boldsymbol{F}^{(t-1)} \cdot \boldsymbol{W}^{(t-1,s)}$ for powers $i \in \mathbb{N}$. Furthermore, since $(\boldsymbol{L}_{norm})^i$ is again a polynomial of the form $q_i(\boldsymbol{D}^{-1/2} \cdot \boldsymbol{A} \cdot \boldsymbol{D}^{-1/2}) := \sum_{j=0}^{r_i} b_{ij} \cdot (\boldsymbol{D}^{-1/2} \cdot \boldsymbol{A} \cdot \boldsymbol{D}^{-1/2})^j$, we can further narrow down the problem to represent

$$(\boldsymbol{D}^{-1/2} \cdot \boldsymbol{A} \cdot \boldsymbol{D}^{-1/2})^j \cdot \boldsymbol{F}^{(t-1)} \cdot \boldsymbol{W}^{(t-1,s)}$$

in $\mathsf{GTL}(\Omega)$, for powers $j \in \mathbb{N}$. And indeed, combining our analysis for GCNs and SGCs results in expressions in $\mathsf{GTL}(\Omega)$. As an example let us consider $(\boldsymbol{D}^{-1/2} \cdot \boldsymbol{A} \cdot \boldsymbol{D}^{-1/2})^2 \cdot \boldsymbol{F}^{(t-1)} \cdot \boldsymbol{W}^{(t-1)}$, that is we use a power of two. It then suffices to define, for each output dimension $j$, the expressions:

$$\psi_j^2(x_1) = f_{1/\sqrt{x}}\big(\textstyle\sum_{x_2} E(x_1, x_2)\big) \cdot \sum_{x_2} \Bigg( E(x_1, x_2) \cdot f_{1/x}\big(\textstyle\sum_{x_1}(E(x_2, x_1))\cdot$$

$$\sum_{x_1} \Big( E(x_2, x_1) \cdot f_{1/\sqrt{x}}\big(\textstyle\sum_{x_2} E(x_1, x_2)\big) \cdot \big(\textstyle\sum_{i=1}^{d_{t-1}} W_{ij}^{(t-1)} \varphi_i^{(t-1)}(x_1)\big)\Big) \Bigg),$$

where the $\varphi_i^{(t-1)}(x_1)$ are expressions representing layer $t-1$. It is then readily verified that we can use $\psi_j^2(x_1)$ to cast layer $t$ of a ChebNet in $\mathsf{GTL}(\Omega)$ with $\Omega$ consisting of $f_{1/\sqrt{x}} : \mathbb{R} \to \mathbb{R} : x \mapsto \frac{1}{\sqrt{x}}$, $f_{1/x} : \mathbb{R} \to \mathbb{R} : x \mapsto \frac{1}{x}$, and the used activation function $\sigma$. We thus recover (and slightly refine) Theorem 2 in Balcilar et al. (2021a):

**Corollary D.1.** *On graphs sharing the same $\lambda_{\max}$ values, the separation power of* ChebNet *is bounded by color refinement, both for graph and vertex embeddings.*

A more fine-grained analysis of the expressions is needed when interested in bounding the summation depth and thus of the number of rounds needed for color refinement. Moreover, as shown by Balcilar et al. (2021a), when graphs have non-regular components with different $\lambda_{\max}$ values, ChebNet can distinguish them, whilst 1-WL cannot. To our knowledge, $\lambda_{\max}$ cannot be computed in $\mathsf{TL}_k(\Omega)$ for any $k$. This implies that it not clear whether an upper bound on the separation power can be obtained for ChebNet taking $\lambda_{\max}$ into account. It is an interesting open question whether there are two graphs $G$ and $H$ which cannot be distinguished by $k$-WL but can be distinguished based on $\lambda_{\max}$. A positive answer would imply that the computation of $\lambda_{\max}$ is beyond reach for $\mathsf{TL}(\Omega)$ and other techniques are needed.

**CayleyNet.** We next show how the separation power of CayleyNet (Levie et al., 2019) can be analyzed. To our knowledge, this analysis is new. We show that the separation power of CayleyNet is bounded by 2-WL. Following Levie et al. (2019) and Balcilar et al. (2021b), in each layer $t$, a CayleyNet updates features as follows:

$$\boldsymbol{F}^{(t)} := \sigma \left( \sum_{s=1}^{p} \boldsymbol{C}^{(s)} \cdot \boldsymbol{F}^{(t-1)} \boldsymbol{W}^{(t-1,s)} \right),$$

with

$$\boldsymbol{C}^{(1)} := \boldsymbol{I}, \boldsymbol{C}^{(2s)} := \mathsf{Re}\left( \left(\frac{h\boldsymbol{L} - \imath\boldsymbol{I}}{h\boldsymbol{L} + \imath\boldsymbol{I}}\right)^s \right), \boldsymbol{C}^{(2s+1)} := \mathsf{Re}\left( \imath\left(\frac{h\boldsymbol{L} - \imath\boldsymbol{I}}{h\boldsymbol{L} + \imath\boldsymbol{I}}\right)^s \right),$$

where $h$ is a constant, $\imath$ is the imaginary unit, and $\mathsf{Re} : \mathbb{C} \to \mathbb{C}$ maps a complex number to its real part. We immediately observe that a CayleyNet requires the use of complex numbers and matrix inversion. So far, we considered real numbers only, but when our separation results are concerned, the choice between real or complex numbers is insignificant. In fact, only the proof of Proposition C.3 requires a minor modification when working on complex numbers: the infinite disjunctions used in the proof now need to range over complex numbers. For matrix inversion, when dealing with separation power, one can use different expressions in $\mathsf{TL}(\Omega)$ for computing the matrix inverse, depending on the input size. And indeed, it is well-known (see e.g., Csanky (1976)) that based on the characteristic polynomial of $\boldsymbol{A}$, $\boldsymbol{A}^{-1}$ for any matrix $\boldsymbol{A} \in \mathbb{R}^{n \times n}$ can be computed as a polynomial $\frac{-1}{c_n} \sum_{i=1}^{n-1} c_i \boldsymbol{A}^{n-1-i}$ if $c_n \neq 0$ and where each coefficient $c_i$ is a polynomial in $\mathsf{tr}(\boldsymbol{A}^j)$, for various $j$. Here, $\mathsf{tr}(\cdot)$ is the *trace* of a matrix. As a consequence, layers in CayleyNet can be viewed as polynomials in $h\boldsymbol{L} - \imath\boldsymbol{I}$ with coefficients polynomials in $\mathsf{tr}((h\boldsymbol{L} - \imath\boldsymbol{I})^j)$. One now needs three index variables to represent the trace computations $\mathsf{tr}((h\boldsymbol{L} - \imath\boldsymbol{I})^j)$. Indeed, let $\varphi_0(x_1, x_2)$ be

the $\mathsf{TL}_2$ expression representing $h\boldsymbol{L} - \imath\boldsymbol{I}$. Then, for example, $(h\boldsymbol{L} - \imath\boldsymbol{I})^j$ can be computed in $\mathsf{TL}_3$ using

$$\varphi_j(x_1, x_2) := \sum_{x_3} \varphi_0(x_1, x_3) \cdot \varphi_{j-1}(x_3, x_2)$$

and hence $\mathsf{tr}((h\boldsymbol{L}-\imath\boldsymbol{I})^j)$ is represented by $\sum_{x_1}\sum_{x_2}\varphi_j(x_1,x_2)\cdot\mathbf{1}_{x_1=x_2}$.. In other words, we obtain expressions in $\mathsf{TL}_3$. The polynomials in $h\boldsymbol{L} - \imath\boldsymbol{I}$ can be represented in $\mathsf{TL}_2$ just as for ChebNet. This implies that each layer in CayleyNet can be represented, on graphs of fixed size, by $\mathsf{TL}_3(\Omega)$ expressions, where $\Omega$ includes the activation function $\sigma$ and the function Re. This suffices to use our general results and conclude that CayleyNets are bounded in separation power by 2-WL. An interesting question is to find graphs that can be separated by a CayleyNet but not by 1-WL. We leave this as an open problem.

## E   PROOF OF THEOREM 5.1

We here consider another higher-order GNN proposal: the *invariant graph networks* or $k$-IGNs of Maron et al. (2019b). By contrast to $k$-FGNNs, $k$-IGNs are linear architectures. If we denote by $k$-IGN$^{(t)}$ the class of $t$ layered $k$-IGNs, then following inclusions are known (Maron et al., 2019b)

$$\rho_1\big(k\text{-IGN}^{(t)}\big) \subseteq \rho_1\big(\mathsf{vwl}_{k-1}^{(t)}\big) \text{ and } \rho_0\big(k\text{-IGN}\big) \subseteq \rho_0\big(\mathsf{gwl}_{k-1}^{(\infty)}\big).$$

The reverse inclusions were posed as open problems in Maron et al. (2019a) and were shown to hold by Chen et al. (2020) for $k = 2$, by means of an extensive case analysis and by relying on properties of 1-WL. In this section, we show that the separation power of $k$-IGNs is bounded by that of $(k-1)$-WL, for arbitrary $k \geq 2$. Theorem 4.2 tells that we can entirely shift our attention to showing that the layers of $k$-IGNs can be represented in $\mathsf{TL}_k(\Omega)$. In other words, we only need to show that $k$ index variables are needed for the layers. As we will see below, this requires a bit of work since a naive representation of the layers of $k$-IGNs use $2k$ index variables. Nevertheless, we show that this can be reduced to $k$ index variables only.

By inspecting the expressions needed to represent the layers of $k$-IGNs in $\mathsf{TL}_k(\Omega)$, we obtain that a $t$ layer $k$-IGN$^{(t)}$ require expressions of summation depth of $tk$. In other words, the correspondence between layers and summation depth is precisely in sync. This implies, by Theorem 4.2:

$$\rho_1\big(k\text{-IGN}\big) = \rho_1\big(\mathsf{vwl}_{k-1}^{(\infty)}\big),$$

where we ignore the number of layers. We similarly obtain that $\rho_0\big(k\text{-IGN}\big) = \rho_0\big(\mathsf{gwl}_{k-1}^{(\infty)}\big)$, hereby answering the open problem posed in Maron et al. (2019a). Finally, we observe that the $k$-IGNs used in Maron et al. (2019b) to show the inclusion $\rho_1\big(k\text{-IGN}^{(t)}\big) \subseteq \rho_1\big(\mathsf{vwl}_{k-1}^{(t)}\big)$ are of very simple form. By defining a simple class of $k$-IGNs, denoted by $k$-GINs, we obtain

$$\rho_1\big(k\text{-GIN}^{(t)}\big) = \rho_1\big(\mathsf{vwl}_{k-1}^{(t)}\big),$$

hereby recovering the layer/round connections.

We start with the following lemma:

**Lemma E.1.** *For any $k \geq 2$, a $t$ layer $k$-IGNs can be represented in $\mathsf{TL}_k^{(tk)}(\Omega)$.*

Before proving this lemma, we recall $k$-IGNs. These are architectures that consist of linear equivariant layers. Such linear layers allow for an explicit description. Indeed, following Maron et al. (2019c), let $\sim_\ell$ be the equality pattern equivalence relation on $[n]^\ell$ such that for $\boldsymbol{a}, \boldsymbol{b} \in [n]^\ell$, $\boldsymbol{a} \sim_\ell \boldsymbol{b}$ if and only if $a_i = a_j \Leftrightarrow b_i = b_j$ for all $j \in [\ell]$. We denote by $[n]^\ell/_{\sim_\ell}$ the equivalence classes induced by $\sim_\ell$. Let us denote by $\mathbf{F}^{(t-1)} \in \mathbb{R}^{n^k \times d_{t-1}}$ the tensor computed by an $k$-IGN in layer $t-1$. Then, in layer $t$, a new tensor in $\mathbb{R}^{n^k \times d_t}$ is computed, as follows. For $j \in [d_t]$ and $v_1, \ldots, v_k \in [n]^k$:

$$F_{v_1,\ldots,v_k,j}^{(t)} := \sigma\left( \sum_{\gamma \in [n]^{2k}/_{\sim_{2k}}} \sum_{\boldsymbol{w} \in [n]^k} \mathbf{1}_{(\boldsymbol{v},\boldsymbol{w}) \in \gamma} \sum_{i \in [d_{t-1}]} c_{\gamma,i,j} F_{w_1,\ldots,w_k,i}^{(t-1)} + \sum_{\mu \in [n]^k/_{\sim_k}} \mathbf{1}_{\boldsymbol{v} \in \mu} b_{\mu,j} \right) \quad (1)$$

for activation function $\sigma$, constants $c_{\gamma,i,j}$ and $b_{\mu,j}$ in $\mathbb{R}$ and where $\mathbf{1}_{(\boldsymbol{v},\boldsymbol{w}) \in \gamma}$ and $\mathbf{1}_{\boldsymbol{v} \in \mu}$ are indicator functions for the $2k$-tuple $(\boldsymbol{v}, \boldsymbol{w})$ to be in the equivalence class $\gamma \in [n]^{2k}/_{\sim_{2k}}$ and the $k$-tuple $\boldsymbol{v}$ to

be in class $\mu \in [n]^k/_{\sim_k}$. As initial tensor $\mathbf{F}^{(0)}$ one defines $F^{(0)}_{v_1,\ldots,v_k,j} := \mathsf{atp}_k(G, \boldsymbol{v}) \in \mathbb{R}^{d_0}$, with $d_0 = 2\binom{k}{2} + k\ell$ where $\ell$ is the number of initial vertex labels, just as for $k$-FGNNs.

We remark that the need for having a summation depth of $tk$ in the expressions in $\mathsf{TL}_k(\Omega)$, or equivalently for requiring $tk$ rounds of $(k-1)$-WL, can intuitively be explained that each layer of a $k$-IGN aggregates more information from "neighbouring" $k$-tuples than $(k-1)$-WL does. Indeed, in each layer, an $k$-IGN can use previous tuple embeddings of all possible $k$-tuples. In a single round of $(k-1)$-WL only previous tuple embeddings from specific sets of $k$-tuples are used. It is only after an additional $k-1$ rounds, that $k$-WL gets to the information about arbitrary $k$-tuples, whereas this information is available in a $k$-IGN in one layer directly.

*Proof of Lemma E.1.* We have seen how $\mathbf{F}^{(0)}$ can be represented in $\mathsf{TL}_k(\Omega)$ when dealing with $k$-FGNNs. We assume now that also the $t-1$th layer $\mathbf{F}^{(t-1)}$ can be represented by $d_{t-1}$ expressions in $\mathsf{TL}_k^{((t-1)k)}(\Omega)$ and show that the same holds for the $t$th layer.

We first represent $\mathbf{F}^{(t)}$ in $\mathsf{TL}_{2k}(\Omega)$, based on the explicit description given earlier. The expressions use index variables $x_1, \ldots, x_k$ and $y_1, \ldots, y_k$. More specifically, for $j \in [d_t]$ we consider the expressions:

$$
\varphi_j^{(t)}(x_1, \ldots, x_k) = \sigma \Bigg( \sum_{\gamma \in [n]^{2k}/_{\sim_{2k}}} \sum_{i=1}^{d_{t-1}} c_{\gamma,i,j}
$$
$$
\sum_{y_1} \cdots \sum_{y_k} \psi_\gamma(x_1, \ldots, x_k, y_1, \ldots, y_k) \cdot \varphi_i^{(t-1)}(y_1, \ldots, y_k)
$$
$$
+ \sum_{\mu \in [n]^k/_{\sim_k}} b_{\mu,j} \cdot \psi_\mu(x_1, \ldots, x_k) \Bigg), \quad (2)
$$

where $\psi_\mu(x_1, \ldots, x_k)$ is a product of expressions of the form $\mathbf{1}_{x_i \, \mathsf{op} \, x_j}$ encoding the equality pattern $\mu$, and similarly, $\psi_\gamma(x_1, \ldots, x_k, y_1, \ldots, y_k)$ is a product of expressions of the form $\mathbf{1}_{x_i \, \mathsf{op} \, x_j}$, $\mathbf{1}_{y_i \, \mathsf{op} \, y_j}$ and $\mathbf{1}_{x_i \, \mathsf{op} \, y_j}$ encoding the equality pattern $\gamma$. These expressions are indicator functions for the their corresponding equality patterns. That is,

$$
[\![\psi_\gamma, (\boldsymbol{v}, \boldsymbol{w})]\!]_G = \begin{cases} 1 & \text{if } (\boldsymbol{v}, \boldsymbol{w}) \in \gamma \\ 0 & \text{otherwise} \end{cases} \qquad [\![\psi_\mu, \boldsymbol{v}]\!]_G = \begin{cases} 1 & \text{if } \boldsymbol{v} \in \mu \\ 0 & \text{otherwise} \end{cases}
$$

We remark that in the expressions $\varphi_j^{(t)}$ we have two kinds of summations: those ranging over a fixed number of elements (over equality patterns, feature dimension), and those ranging over the index variables $y_1, \ldots, y_k$. The latter are the only ones contributing the summation depth. The former are just concise representations of a long summation over a fixed number of expressions.

We now only need to show that we can equivalently write $\varphi_j^{(t)}(x_1, \ldots, x_k)$ as expressions in $\mathsf{TL}_k(\Omega)$, that is, using only indices $x_1, \ldots, x_k$. As such, we can already ignore the term $\sum_{\mu \in [n]^k/_{\sim_k}} b_{\mu,j} \cdot \psi_\mu(x_1, \ldots, x_k)$ since this is already in $\mathsf{TL}_k(\Omega)$. Furthermore, this expressions does not affect the summation depth.

Furthermore, as just mentioned, we can expand expression $\varphi_j^{(t)}$ into linear combinations of other simpler expressions. As such, it suffices to show that $k$ index variables suffice for each expression of the form:

$$
\sum_{y_1} \cdots \sum_{y_k} \psi_\gamma(x_1, \ldots, x_k, y_1, \ldots, y_k) \cdot \varphi_i^{(t)}(y_1, \ldots, y_k), \quad (3)
$$

obtained by fixing $\mu$ and $i$ in expression (2). To reduce the number of variables, as a first step we eliminate any disequality using the inclusion-exclusion principle. More precisely, we observe that $\psi_\gamma(\boldsymbol{x}, \boldsymbol{y})$ can be written as:

$$
\prod_{(i,j) \in I} \mathbf{1}_{x_i = x_j} \cdot \prod_{(i,j) \in \bar{I}} \mathbf{1}_{x_i \neq x_j} \cdot \prod_{(i,j) \in J} \mathbf{1}_{y_i = y_j} \cdot \prod_{(i,j) \in \bar{J}} \mathbf{1}_{y_i \neq y_j} \prod_{(i,j) \in K} \mathbf{1}_{x_i = y_j} \cdot \prod_{(i,j) \in \bar{K}} \mathbf{1}_{x_i \neq y_j}
$$

$$= \sum_{A \subseteq \bar{I}} \sum_{B \subseteq \bar{J}} \sum_{C \subseteq \bar{K}} (-1)^{|A|+|B|+|C|} \prod_{(i,j) \in I \cup A} \mathbf{1}_{x_i=x_j} \prod_{(i,j) \in J \cup B} \mathbf{1}_{y_i=y_j} \cdot \prod_{(i,j) \in J \cup C} \mathbf{1}_{x_i=y_j}, \quad (4)$$

for some sets $I$, $J$ and $K$ of pairs of indices in $[k]^2$, and where $\bar{I} = [k]^2 \setminus I$, $\bar{J} = [k]^2 \setminus J$ and $\bar{K} = [k]^2 \setminus K$. Here we use that $\mathbf{1}_{x_i \neq x_j} = 1 - \mathbf{1}_{x_i=x_j}$, $\mathbf{1}_{y_i \neq y_j} = 1 - \mathbf{1}_{y_i=y_j}$ and $\mathbf{1}_{x_i \neq y_j} = 1 - \mathbf{1}_{y_i=y_j}$ and use the inclusion-exclusion principle to obtain a polynomial in equality conditions only.

In view of expression (4), we can push the summations over $y_1, \ldots, y_k$ in expression (3) to the subexpressions that actually use $y_1, \ldots, y_k$. That is, we can rewrite expression (3) into the equivalent expression:

$$\sum_{A \subseteq \bar{I}} \sum_{B \subseteq \bar{J}} \sum_{C \subseteq \bar{K}} (-1)^{|A|+|B|+|C|} \cdot \prod_{(i,j) \in I \cup A} \mathbf{1}_{x_i=x_j}$$

$$\cdot \left( \sum_{y_1} \cdots \sum_{y_k} \prod_{(i,j) \in J \cup B} \mathbf{1}_{y_i=y_j} \cdot \prod_{(i,j) \in K \cup C} \mathbf{1}_{x_i=y_j} \cdot \varphi_i^{(t-1)}(y_1, \ldots, y_k) \right). \quad (5)$$

By fixing $A$, $B$ and $C$, it now suffices to argue that

$$\prod_{(i,j) \in I \cup A} \mathbf{1}_{x_i=x_j} \cdot \left( \sum_{y_1} \cdots \sum_{y_k} \prod_{(i,j) \in J \cup B} \mathbf{1}_{y_i=y_j} \cdot \prod_{(i,j) \in K \cup C} \mathbf{1}_{x_i=y_j} \cdot \varphi_i^{(t-1)}(y_1, \ldots, y_k) \right), \quad (6)$$

can be equivalently expressed in $\mathsf{TL}_k(\Omega)$.

Since our aim is to reduced the number of index variables from $2k$ to $k$, it is important to known which variables are the same. In expression (6), some equalities that hold between the variables may not be explicitly mentioned. For this reason, we expand $I \cup A$, $J \cup B$ and $K \cup C$ with their implied equalities. That is, $\mathbf{1}_{x_i=x_j}$ is added to $I \cup A$, if for any $(\boldsymbol{v}, \boldsymbol{w})$ such that

$$\llbracket \prod_{(i,j) \in I \cup A} \mathbf{1}_{x_i=x_j} \cdot \prod_{(i,j) \in J \cup B} \mathbf{1}_{y_i=y_j} \cdot \prod_{(i,j) \in K \cup C} \mathbf{1}_{x_i=y_j}, (\boldsymbol{v}, \boldsymbol{w}) \rrbracket_G = 1 \Rightarrow \llbracket \mathbf{1}_{x_i=x_j}, \boldsymbol{v} \rrbracket_G = 1$$

holds. Similar implied equalities $\mathbf{1}_{y_i=y_j}$ and $\mathbf{1}_{x_i=y_j}$ are added to $J \cup B$ and $K \cup C$, respectively. let us denoted by $I'$, $J'$ and $K'$. It should be clear that we can add these implied equalities to expression (6) without changing its semantics. In other words, expression (6) can be equivalently represented by

$$\prod_{(i,j) \in I'} \mathbf{1}_{x_i=x_j} \cdot \left( \sum_{y_1} \cdots \sum_{y_k} \prod_{(i,j) \in J'} \mathbf{1}_{y_i=y_j} \cdot \prod_{(i,j) \in K'} \mathbf{1}_{x_i=y_j} \cdot \varphi_i^{((t-1)}(y_1, \ldots, y_k) \right), \quad (7)$$

There now two types of index variables among the $y_1, \ldots, y_k$: those that are equal to some $x_i$, and those that are not. Now suppose that $(j, j') \in J'$, and thus $y_j = y_{j'}$, and that also $(i, j) \in K'$, and thus $x_i = y_j$. Since we included the implied equalities, we also have $(i, j') \in K'$, and thus $x_i = y_{j'}$. There is no reason to keep $(j, j') \in J'$ as it is implied by $(i, j)$ and $(i, j') \in K'$. We can thus safely remove all pairs $(j, j')$ from $J'$ such that $(i, j) \in K'$ (and thus also $(i, j') \in K'$). We denote by $J''$ be the reduced set of pairs of indices obtained from $J'$ in this way. We have that expression (7) can be equivalently written as

$$\prod_{(i,j) \in I'} \mathbf{1}_{x_i=x_j} \cdot \left( \sum_{y_1} \cdots \sum_{y_k} \prod_{(i,j) \in K'} \mathbf{1}_{x_i=y_j} \cdot \prod_{(i,j) \in J''} \mathbf{1}_{y_i=y_j} \cdot \varphi_i^{(t-1)}(y_1, \ldots, y_k) \right), \quad (8)$$

where we also switched the order of equalities in $J''$ and $K'$. Our construction of $J''$ and $K'$ ensures that none of the variables $y_j$ with $j$ belonging to a pair in $J''$ is equal to some $x_i$.

By contrast, the variable $y_j$ occurring in $(i, j) \in K'$ are equal to $x_i$. We observe, however, that also certain equalities among the variables $\{x_1, \ldots, x_k\}$ hold, as represented by the pairs in $I'$. let $I'(i) := \{i' \mid (i, i') \in I'\}$ and define $\hat{\imath}$ as a unique representative element in $I'(i)$. For example, one can take $\hat{\imath}$ to be smallest index in $I'(i)$. We use this representative index (and corresponding $x$-variable) to simplify $K'$. More precisely, we replace each pair $(i, j) \in K'$ with the pair $(\hat{\imath}, j)$. In

terms of variables, we replace $x_i = y_j$ with $x_{\hat{i}} = y_j$. Let $K''$ be the set $K''$ modified in that way. Expression (8) can thus be equivalently written as

$$\prod_{(i,j)\in I'} \mathbf{1}_{x_i=x_j} \cdot \left( \sum_{y_1} \cdots \sum_{y_k} \prod_{(\hat{i},j)\in K''} \mathbf{1}_{x_{\hat{i}}=y_j} \cdot \prod_{(i,j)\in J''} \mathbf{1}_{y_i=y_j} \cdot \varphi_i^{(t-1)}(y_1,\ldots,y_k) \right), \quad (9)$$

where the free index variables of the subexpression

$$\sum_{y_1} \cdots \sum_{y_k} \prod_{(\hat{i},j)\in K''} \mathbf{1}_{x_{\hat{i}}=y_j} \cdot \prod_{(i,j)\in J''} \mathbf{1}_{y_i=y_j} \cdot \varphi_i^{(t-1)}(y_1,\ldots,y_k) \quad (10)$$

are precisely the index variables $x_{\hat{i}}$ for $(\hat{i},j) \in K''$. Recall that our aim is to reduce the variables from $2k$ to $k$. We are now finally ready to do this. More specifically, we consider a bijection $\beta : \{y_1,\ldots,y_k\} \to \{x_1,\ldots,x_k\}$ in which ensure that for each $\hat{i}$ there is a $j$ such that $(\hat{i},j) \in K''$ and $\beta(y_j) = x_{\hat{i}}$. Furthermore, among the summations $\sum_{y_1} \cdots \sum_{y_k}$ we can ignore those for which $\beta(y_j) = x_{\hat{i}}$ holds. After all, they only contribute for a given $x_{\hat{i}}$ value. Let $Y$ be those indices in $[k]$ such that $\beta(y_j) \neq x_{\hat{i}}$ for some $\hat{i}$. Then, we can equivalently write expression (9) as

$$\prod_{(i,j)\in I'} \mathbf{1}_{x_i=x_j} \cdot \left( \sum_{\beta(y_i),i\in Y} \prod_{(\hat{i},j)\in K'} \mathbf{1}_{x_{\hat{i}}=\beta(y_j)} \cdot \prod_{(i,j)\in J''} \mathbf{1}_{\beta(y_i)=\beta(y_j)} \right.$$
$$\left. \cdot \beta(\varphi_i^{(t-1)}(y_1,\ldots,y_k)) \right), \quad (11)$$

where $\beta(\varphi_i^{(t-1)}(y_1,\ldots,y_k))$ denotes the expression obtained by renaming of variables $y_1,\ldots,y_j$ in $\varphi_i^{(t-1)}(y_1,\ldots,y_k)$ into $x$-variables according to $\beta$. This is our desired expression in $\mathsf{TL}_k(\Omega)$. If we analyze the summation depth of this expression, we have by induction that the summation depth of $\varphi_i^{(t-1)}$ is at most $(t-1)k$. In the above expression, we are increasing the summation depth with at most $|Y|$. The largest size of $Y$ is $k$, which occurs when none of the $y$-variables are equal to any of the $x$-variables. As a consequence, we obtained an expression of summation depth at most $tk$, as desired. □

As a consequence, when using $k$-IGNs$^{(t)}$ for vertex embeddings, using $(G,v) \to \mathbf{F}_{v,\ldots,v,:}^{(t)}$ one simply pads the layer expression with $\prod_{i\in[k]} \mathbf{1}_{x_1=x_i}$ which does not affect the number of variables or summation depth. When using $k$-IGNs$^{(t)}$ of graph embeddings, an additional invariant layer is added to obtain an embedding from $G \to \mathbb{R}^{d_t}$. Such invariant layers have a similar (simpler) representation as given in equation 1 (Maron et al., 2019c), and allow for a similar analysis. One can verify that expressions in $\mathsf{TL}_k^{((t+1)k)}(\Omega)$ are needed when such an invariant layer is added to previous $t$ layers. Based on this, Theorem 4.2, Lemma E.1 and Theorem 1 in Maron et al. (2019b), imply that $\rho_1(k\text{-IGN}) = \rho_1(\mathsf{vwl}_{k-1}^{(\infty)})$ and $\rho_0(k\text{-IGN}) = \rho_0(\mathsf{gwl}_{k-1}^{(\infty)})$ hold.

$k$**-dimensional GINs.** We can recover a layer-based characterization for $k$-IGNs that compute vertex embeddings by considering a special subset of $k$-IGNs. Indeed, the $k$-IGNs used in Maron et al. (2019b) to show $\rho_1(\mathsf{wl}_{k-1}^{(t)}) \subseteq \rho_1(k\text{-IGN}^{(t)})$ are of a very special form. We extract the essence of these special $k$-IGNs in the form of $k$-dimensional GINs. That is, we define the class $k$-GINs to consist of layers defined as follows. The initial layers are just as for $k$-IGNs. Then, for $t \geq 1$:

$$\mathbf{F}_{v_1,\ldots,v_k,:}^{(t)} := \mathsf{mlp}_0^{(t)}\Big(\mathbf{F}_{v_1,\ldots,v_k,:}^{(t-1)}, \sum_{u\in V_G} \mathsf{mlp}_1^{(t)}(\mathbf{F}_{u,v_2,\ldots,v_k,:}^{(t-1)}), \sum_{u\in V_G} \mathsf{mlp}_1^{(t)}(\mathbf{F}_{v_1,u,\ldots,v_k,:}^{(t-1)})$$
$$,\ldots, \sum_{u\in V_G} \mathsf{mlp}_1^{(t)}(\mathbf{F}_{v_1,v_2,\ldots,v_{k-1},w,:}^{(t-1)}))\Big),$$

where $F_{v_1,v_2,\ldots,v_k,:}^{(t-1)} \in \mathbb{R}^{d_{t-1}}$, $\mathsf{mlp}_1^{(t)} : \mathbb{R}^{d_{t-1}} \to \mathbb{R}^{b_t}$ and $\mathsf{mlp}_1^{(t)} : \mathbb{R}^{d_{t-1}+kb_t} \to \mathbb{R}^{d_t}$ are MLPs. It is now an easy exercise to show that $k$-GIN$^{(t)}$ can be represented in $\mathsf{TL}_k^{(t)}(\Omega)$ (remark that the summations used increase the summation depth with one only in each layer). Combined with Theorem 4.2 and by inspecting the proof of Theorem 1 in Maron et al. (2019b), we obtain:

**Proposition E.2.** *For any $k \geq 2$ and any $t \geq 0$: $\rho_1(k\text{-}\mathsf{GIN}^{(t)}) = \rho_1(\mathsf{vwl}_{k-1}^{(t)})$.*

We can define the invariant version of $k$-IGNs by adding a simple readout layer of the form

$$\sum_{v_1,\ldots,v_k \in V_G} \mathsf{mlp}(\mathbf{F}_{v_1,\ldots,v_k,:}^{(t)}),$$

as is used in Maron et al. (2019b). We obtain, $\rho_0(k\text{-}\mathsf{GIN}) = \rho_0(\mathsf{gwl}_{k-1}^{(\infty)})$, by simply rephrasing the readout layer in $\mathsf{TL}_k(\Omega)$.

## F  DETAILS OF SECTION 6

Let $\mathcal{C}(\mathcal{G}_s, \mathbb{R}^\ell)$ be the class of all continuous functions from $\mathcal{G}_s$ to $\mathbb{R}^\ell$. We always assume that $\mathcal{G}_s$ forms a compact space. For example, when vertices are labeled with values in $\{0,1\}^{\ell_0}$, $\mathcal{G}_s$ is a finite set which we equip with the discrete topology. When vertices carry labels in $\mathbb{R}^{\ell_0}$ we assume that these labels come from a compact set $K \subset \mathbb{R}^{\ell_0}$. In this case, one can represent graphs in $\mathcal{G}_s$ by elements in $(\mathbb{R}^{\ell_0})^2$ and the topology used is the one induced by some metric $\|.\|$ on the reals. Similarly, we equip $\mathbb{R}^\ell$ with the topology induced by some metric $\|.\|$.

Consider $\mathcal{F} \subseteq \mathcal{C}(\mathcal{G}_s, \mathbb{R}^\ell)$ and define $\overline{\mathcal{F}}$ as the *closure of $\mathcal{F}$* in $\mathcal{C}(\mathcal{G}_s, \mathbb{R}^\ell)$ under the usual topology induced by $f \mapsto \sup_{G,\boldsymbol{v}}\|f(G,\boldsymbol{v})\|$. In other words, a continuous function $h : \mathcal{G}_s \to \mathbb{R}^\ell$ is in $\overline{\mathcal{F}}$ if there exists a sequence of functions $f_1, f_2, \ldots \in \mathcal{F}$ such that $\lim_{i\to\infty} \sup_{G,\boldsymbol{v}}\|f_i(G,\boldsymbol{v}) - h(G,\boldsymbol{v})\| = 0$. The following theorem provides a characterization of the closure of a set of functions. We state it here modified to our setting.

**Theorem F.1** ((Timofte, 2005)). *Let $\mathcal{F} \subseteq \mathcal{C}(\mathcal{G}_s, \mathbb{R}^\ell)$ such that there exists a set $\mathcal{S} \subseteq \mathcal{C}(\mathcal{G}_s, \mathbb{R})$ satisfying $\mathcal{S} \cdot \mathcal{F} \subseteq \mathcal{F}$ and $\rho(\mathcal{S}) \subseteq \rho(\mathcal{F})$. Then,*

$$\overline{\mathcal{F}} := \left\{ f \in \mathcal{C}(\mathcal{G}_s, \mathbb{R}^\ell) \mid \rho(F) \subseteq \rho(f), \forall (G,\boldsymbol{v}) \in \mathcal{G}_s, f(G,\boldsymbol{v}) \in \overline{\mathcal{F}(G,\boldsymbol{v})} \right\},$$

*where $\mathcal{F}(G,\boldsymbol{v}) := \{h(G,\boldsymbol{v}) \mid h \in \mathcal{F}\} \subseteq \mathbb{R}^\ell$. We can equivalently replace $\rho(\mathcal{F})$ by $\rho(\mathcal{S})$ in the expression for $\overline{\mathcal{F}}$.* $\square$

We will use this theorem to show Theorem 6.1 in the setting that $\mathcal{F}$ consists of functions that can be represented in $\mathsf{TL}(\Omega)$, and more generally, sets of functions that satisfy two conditions, stated below. We more generally allow $\mathcal{F}$ to consist of functions $f : \mathcal{G}_s \to \mathbb{R}^{\ell_f}$, where the $\ell_f \in \mathbb{N}$ may depend on $f$. We will require $\mathcal{F}$ to satisfy the following two conditions:

**concatenation-closed:** If $f_1 : \mathcal{G}_s \to \mathbb{R}^p$ and $f_2 : \mathcal{G}_s \to \mathbb{R}^q$ are in $\mathcal{F}$, then $g := (f_1, f_2) : \mathcal{G}_s \to \mathbb{R}^{p+q} : (G,\boldsymbol{v}) \mapsto (f_1(G,\boldsymbol{v}), f_2(G,\boldsymbol{v}))$ is also in $\mathcal{F}$.

**function-closed:** For a fixed $\ell \in \mathbb{N}$, for any $f \in \mathcal{F}$ such that $f : \mathcal{G}_s \to \mathbb{R}^p$, also $h \circ f : \mathcal{G}_s \to \mathbb{R}^\ell$ is in $\mathcal{F}$ for any continuous function $h \in \mathcal{C}(\mathbb{R}^p, \mathbb{R}^\ell)$.

We denote by $\mathcal{F}_\ell$ be the subset of $\mathcal{F}$ of functions from $\mathcal{G}_s$ to $\mathbb{R}^\ell$.

**Theorem 6.1.** *For any $\ell$, and any set $\mathcal{F}$ of functions, concatenation and function closed for $\ell$, we have: $\overline{\mathcal{F}_\ell} = \{f : \mathcal{G}_s \to \mathbb{R}^\ell \mid \rho_s(\mathcal{F}) \subseteq \rho_s(f)\}$.*

*Proof.* The proof consist of (i) verifying the existence of a set $\mathcal{S}$ as mentioned Theorem F.1; and of (ii) eliminating the pointwise convergence condition "$\forall (G,\boldsymbol{v}) \in \mathcal{G}_s, f(G,\boldsymbol{v}) \in \overline{\mathcal{F}_\ell(G,\boldsymbol{v})}$" in the closure characterization in Theorem F.1.

For showing (ii) we argue that $\overline{\mathcal{F}_\ell(G,\boldsymbol{v})} = \mathbb{R}^\ell$ such that the conditions $f(G,\boldsymbol{v}) \in \overline{\mathcal{F}_\ell(G,\boldsymbol{v})}$ is automatically satisfied for any $f \in \mathcal{C}(\mathcal{G}_s, \mathbb{R}^\ell)$. Indeed, take an arbitrary $f \in \mathcal{G}_k \to \mathbb{R}^\ell$ and consider the constant functions $b_i : \mathbb{R}^\ell \to \mathbb{R}^\ell : \boldsymbol{x} \mapsto \boldsymbol{b}_i$ with $\boldsymbol{b}_i \in \mathbb{R}^\ell$ the $i$th basis vector. Since $\mathcal{F}$ is function-closed for $\ell$, so is $\mathcal{F}_\ell$. Hence, $b_i := g_i \circ f \in \mathcal{F}_\ell$ as well. Furthermore, if $s_a : \mathbb{R}^\ell \to \mathbb{R}^\ell : \boldsymbol{x} \mapsto a \times \boldsymbol{x}$, for $a \in \mathbb{R}$, then $s_a \circ f \in \mathcal{F}_\ell$ and thus $\mathcal{F}_\ell$ is closed under scalar multiplication. Finally, consider $+ : \mathbb{R}^{2\ell} \to \mathbb{R}^\ell : (\boldsymbol{x}, \boldsymbol{y}) \mapsto \boldsymbol{x} + \boldsymbol{y}$. For $f$ and $g$ in $\mathcal{F}_\ell$, $h = (f,g) \in \mathcal{F}$ since $\mathcal{F}$ is concatenation-closed. As a consequence, the function $+ \circ h : \mathcal{G}_s \to \mathbb{R}^\ell$ is in $\mathcal{F}_\ell$, showing that $\mathcal{F}_\ell$ is also closed under addition. All combined, this shows that $\mathcal{F}_\ell$ is closed under taking linear combinations and since the basis vectors of $\mathbb{R}^\ell$ can be attained, $\overline{\mathcal{F}_\ell(G,\boldsymbol{v})} := \mathbb{R}^\ell$, as desired.

For (i), we show the existence of a set $\mathcal{S} \subseteq \mathcal{C}(\mathcal{G}_s, \mathbb{R})$ such that $\mathcal{S} \cdot \mathcal{F}_\ell \subseteq \mathcal{F}_\ell$ and $\rho_s(\mathcal{S}) \subseteq \rho_s(\mathcal{F}_\ell)$ hold. Similarly as in Azizian & Lelarge (2021), we define

$$\mathcal{S} := \big\{ f \in \mathcal{C}(\mathcal{G}_s, \mathbb{R}) \mid \underbrace{(f, f, \ldots, f)}_{\ell \text{ times}} \in \mathcal{F}_\ell \big\}.$$

We remark that for $s \in \mathcal{S}$ and $f \in \mathcal{F}_\ell$, $s \cdot f : \mathcal{G}_s \to \mathbb{R}^\ell : (G, \boldsymbol{v}) \mapsto s(G, \boldsymbol{v}) \odot f(G, \boldsymbol{v})$, with $\odot$ being pointwise multiplication, is also in $\mathcal{F}_\ell$. Indeed, $s \cdot f = \odot \circ (s, f)$ with $(s, f)$ the concatenation of $s$ and $f$ and $\odot : \mathbb{R}^{2\ell} \to \mathbb{R}^\ell : (\boldsymbol{x}, \boldsymbol{y}) \to \boldsymbol{x} \odot \boldsymbol{y}$ being pointwise multiplication.

It remains to verify $\rho_s(\mathcal{S}) \subseteq \rho_s(\mathcal{F}_\ell)$. Assume that $(G, \boldsymbol{v})$ and $(H, \boldsymbol{w})$ are not in $\rho_s(\mathcal{F}_\ell)$. By definition, this implies the existence of a function $\hat{f} \in \mathcal{F}_\ell$ such that $\hat{f}(G, \boldsymbol{v}) = \boldsymbol{a} \neq \boldsymbol{b} = \hat{f}(H, \boldsymbol{w})$ with $\boldsymbol{a}, \boldsymbol{b} \in \mathbb{R}^\ell$. We argue that $(G, \boldsymbol{v})$ and $(H, \boldsymbol{w})$ are also not in $\rho_s(\mathcal{S})$ either. Indeed, Proposition 1 in Maron et al. (2019b) implies that there exists natural numbers $\boldsymbol{\alpha} = (\alpha_1, \ldots, \alpha_\ell) \in \mathbb{N}^\ell$ such that the mapping $h_{\boldsymbol{\alpha}} : \mathbb{R}^\ell \to \mathbb{R} : \boldsymbol{x} \to \prod_{i=1}^\ell x_i^{\alpha_i}$ satisfies $h_{\boldsymbol{\alpha}}(\boldsymbol{a}) = a \neq b = h_{\boldsymbol{\alpha}}(\boldsymbol{b})$, with $a, b \in \mathbb{R}$. Since $\mathcal{F}$ (and thus also $\mathcal{F}_\ell$) is function-closed, $h_{\boldsymbol{\alpha}} \circ f \in \mathcal{F}_\ell$ for any $f \in \mathcal{F}_\ell$. In particular, $g := h_{\boldsymbol{\alpha}} \circ \hat{f} \in \mathcal{F}_\ell$ and concatenation-closure implies that $(g, \ldots, g) : \mathcal{G}_s \to \mathbb{R}^\ell$ is in $\mathcal{F}_\ell$ too. Hence, $g \in \mathcal{S}$, by definition. It now suffices to observe that $g(G, \boldsymbol{v}) = h_{\boldsymbol{\alpha}}(\hat{f}(G, \boldsymbol{v})) = a \neq b = h_{\boldsymbol{\alpha}}(\hat{f}(H, \boldsymbol{w})) = g(H, \boldsymbol{w})$, and thus $(G, \boldsymbol{v})$ and $(H, \boldsymbol{w})$ are not in $\rho_s(\mathcal{S})$, as desired. $\qquad\square$

When we know more about $\rho_s(\mathcal{F}_\ell)$ we can say a bit more. In the following, we let $\mathsf{alg} \in \{\mathsf{cr}^{(t)}, \mathsf{gcr}^{(t)}, \mathsf{vwl}_k^{(t)}, \mathsf{gwl}_k^{(\infty)}\}$ and only consider the setting where $s$ is either 0 (invariant graph functions) or $s = 1$ (equivariant graph/vertex functions).

**Corollary 6.2.** *Under the assumptions of Theorem 6.1 and if $\rho(\mathcal{F}_\ell) = \rho(\mathsf{alg})$, then $\overline{\mathcal{F}_\ell} = \{f : \mathcal{G}_s \to \mathbb{R}^\ell \mid \rho(\mathsf{alg}) \subseteq \rho(f)\}$.*

*Proof.* This is just a mere restatement of Theorem 6.1 in which $\rho_s(\mathcal{F}_\ell)$ in the condition $\rho_s(\mathcal{F}_\ell) \subseteq \rho_s(f)$ is replaced by $\rho_s(\mathsf{alg})$, where $s = 1$ for $\mathsf{alg} \in \{\mathsf{cr}^{(t)}, \mathsf{vwl}_k^{(t)}\}$ and $s = 0$ for $\mathsf{alg} \in \{\mathsf{gcr}^{(t)}, \mathsf{gwl}_k^{(\infty)}\}$. $\qquad\square$

To relate all this to functions representable by tensor languages, we make the following observations. First, if we consider $\mathcal{F}$ to be the set of all functions that can be represented in $\mathsf{GTL}^{(t)}(\Omega)$, $\mathsf{TL}_2^{(t+1)}(\Omega)$, $\mathsf{TL}_{k+1}^{(t)}(\Omega)$ or $\mathsf{TL}(\Omega)$, then $\mathcal{F}$ will be automatically concatenation and function-closed, provided that $\Omega$ consists of all functions in $\bigcup_p \mathcal{C}(\mathbb{R}^p, \mathbb{R}^\ell)$. Hence, Theorem 6.1 applies. Furthermore, our results from Section 4 tell us that for all $t \geq 0$, and $k \geq 1$, $\rho_1(\mathsf{cr}^{(t)}) = \rho_1(\mathsf{GTL}^{(t)}(\Omega))$, $\rho_0(\mathsf{gcr}^{(t)}) = \rho_0(\mathsf{TL}_2^{(t+1)}(\Omega)) = \rho_0(\mathsf{gwl}_1^{(t)})$, $\rho_1(\mathsf{ewl}_k^{(t)}) = \rho_1(\mathsf{TL}_{k+1}^{(t)}(\Omega))$, and $\rho_0(\mathsf{TL}_{k+1}(\Omega)) = \rho_0(\mathsf{gwl}_k^{(\infty)})$. As a consequence, Corollary 6.2 applies as well. We thus easily obtain the following characterizations:

**Proposition F.2.** *For any $t \geq 0$ and $k \geq 1$:*

- *If $\mathcal{F}$ consists of all functions representable in $\mathsf{GTL}^{(t)}(\Omega)$, then $\overline{\mathcal{F}_\ell} = \{f : \mathcal{G}_1 \to \mathbb{R}^\ell \mid \rho_1(\mathsf{cr}^{(t)}) \subseteq \rho_1(f)\}$;*

- *If $\mathcal{F}$ consists of all functions representable in $\mathsf{TL}_{k+1}^{(t)}(\Omega)$, then $\overline{\mathcal{F}_\ell} = \{f : \mathcal{G}_1 \to \mathbb{R}^\ell \mid \rho_1(\mathsf{vwl}_k^{(t)}) \subseteq \rho_1(f)\}$;*

- *If $\mathcal{F}$ consists of all functions representable in $\mathsf{TL}_2^{(t+1)}(\Omega)$, then $\overline{\mathcal{F}_\ell} = \{f : \mathcal{G}_0 \to \mathbb{R}^\ell \mid \rho_0(\mathsf{gwl}_1^{(t)}) \subseteq \rho_0(f)\}$; and finally,*

- *If $\mathcal{F}$ consists of all functions representable in $\mathsf{TL}_{k+1}(\Omega)$, then $\overline{\mathcal{F}_\ell} = \{f : \mathcal{G}_0 \to \mathbb{R}^\ell \mid \rho_0(\mathsf{gwl}_k^{(\infty)}) \subseteq \rho_0(f)\}$,*

*provided that $\Omega$ consists of all functions in $\bigcup_p \mathcal{C}(\mathbb{R}^p, \mathbb{R}^\ell)$.*

In fact, Lemma 32 in Azizian & Lelarge (2021) implies that we can equivalently populate $\Omega$ with all MLPs instead of all continuous functions. We can thus use MLPs and continuous functions interchangeably when considering the closure of functions.

At this point, we want to make a comparison with the results and techniques in Azizian & Lelarge (2021). Our proof strategy is very similar and is also based on Theorem F.1. The key distinguishing feature is that we consider functions $f : \mathcal{G}_s \to \mathbb{R}^{\ell_f}$ instead of functions from graphs alone. This has as great advantage that no separate proofs are needed to deal with invariant or equivariant functions. Equivariance incurs quite some complexity in the setting considered in Azizian & Lelarge (2021). A second major difference is that, by considering functions representable in tensor languages, and based on our results from Section 4, we obtain a more fine-grained characterization. Indeed, we obtain characterizations in terms of the number of rounds used in CR and $k$-WL. In Azizian & Lelarge (2021), $t$ is always set to $\infty$, that is, an unbounded number of rounds is considered. Furthermore, when it concerns functions $f : \mathcal{G}_1 \to \mathbb{R}^{\ell_f}$, we recall that CR is different from 1-WL. Only 1-WL is considered in Azizian & Lelarge (2021). Finally, another difference is that we define the equivariant version $\mathsf{vwl}_k$ in a different way than is done in Azizian & Lelarge (2021), because in this way, a tighter connection to logics and tensor languages can be made. In fact, if we were to use the equivariant version of $k$-WL from Azizian & Lelarge (2021), then we necessarily have to consider an unbounded number of rounds (similarly as in our $\mathsf{gwl}_k$ case).

We conclude this section by providing a little more details about the consequences of the above results for GNNs. As we already mentioned in Section 6.2, many common GNN architectures are concatenation and function-closed (using MLPs instead of continuous functions). This holds, for example, for the classes $\mathsf{GIN}_\ell^{(t)}$, $\mathsf{eGIN}_\ell^{(t)}$, $k$-$\mathsf{FGNN}_\ell^{(t)}$ and $k$-$\mathsf{GIN}_\ell^{(t)}$ and $k$-$\mathsf{IGN}^{(t)}$, as described in Section 5 and further detailed in Section E and D. Here, the subscript $\ell$ refers to the dimension of the embedding space.

We now consider a function $f$ that is not more separating than $\mathsf{cr}^{(t)}$ (respectively, $\mathsf{gcr}^{(t)}$, $\mathsf{vwl}_k^{(t)}$ or $\mathsf{gwl}_k^{(\infty)}$, for some $k \geq 1$), and want to know whether $f$ can be approximated by a class of GNNs. Proposition F.2 tells that such $f$ can be approximated by a class of GNNs as long as these are at least as separating as $\mathsf{GTL}^{(t)}$ (respectively, $\mathsf{TL}_2^{(t+1)}$, $\mathsf{TL}_{k+1}^{(t)}$ or $\mathsf{TL}_{k+1}^{(\infty)}$). This, in turn, amounts showing that the GNNs can be represented in the corresponding tensor language fragment, and that they can match the corresponding labeling algorithm in separation power. We illustrate this for the GNN architectures mentioned above.

- In Section 5 we showed that $\mathsf{GIN}_\ell^{(t)}$ can be represented in $\mathsf{GTL}^{(t)}(\Omega)$. Theorem 4.3 then implies that $\rho_1(\mathsf{cr}^{(t)}) \subseteq \rho_1(\mathsf{GIN}_\ell^{(t)})$. Furthermore, Xu et al. (2019) showed that $\rho_1(\mathsf{GIN}_\ell^{(t)}) \subseteq \rho_1(\mathsf{cr}^{(t)})$. As a consequence, $\rho_1(\mathsf{GIN}_\ell^{(t)}) = \rho_1(\mathsf{cr}^{(t)})$. We note that the lower bound for GINs only holds when graphs carry discrete labels. The same restriction is imposed in Azizian & Lelarge (2021).

- In Section 5 we showed that $\mathsf{eGIN}_\ell^{(t)}$ can be represented in $\mathsf{TL}_2^{(t)}(\Omega)$. Theorem 4.2 then implies that $\rho_1(\mathsf{vwl}_1^{(t)}) \subseteq \rho_1(\mathsf{eGIN}_\ell^{(t)})$. Furthermore, Barceló et al. (2020) showed that $\rho_1(\mathsf{eGIN}_\ell^{(t)}) \subseteq \rho_1(\mathsf{vwl}_1^{(t)})$. As a consequence, $\rho_1(\mathsf{eGIN}_\ell^{(t)}) = \rho_1(\mathsf{vwl}_1^{(t)})$. Again, the lower bound is only valid when graphs carry discrete labels.

- In Section 5 we mentioned (see details in Section D) that $k$-$\mathsf{FGNN}_\ell^{(t)}$ can be represented in $\mathsf{TL}_{k+1}^{(t)}(\Omega)$. Theorem 4.2 then implies that $\rho_1(\mathsf{vwl}_k^{(t)}) \subseteq \rho_1(k\text{-}\mathsf{FGNN}_\ell^{(t)})$. Furthermore, Maron et al. (2019b) showed that $\rho_1(k\text{-}\mathsf{FGNN}_\ell^{(t)}) \subseteq \rho_1(\mathsf{vwl}_k^{(t)})$. As a consequence, $\rho_1(k\text{-}\mathsf{FGNN}_\ell^{(t)}) \subseteq \rho_1(\mathsf{vwl}_k^{(t)})$. Similarly, $\rho_1((k+1)\text{-}\mathsf{GIN}_\ell^{(t)}) = \rho_1(\mathsf{vwl}_k^{(t)})$ for the special class of $(k+1)$-IGNs described in Section E. No restrictions are in place for the lower bounds and hence real-valued vertex-labelled graphs can be considered.

- When $\mathsf{GIN}_\ell^{(t)}$ or $\mathsf{eGIN}_\ell^{(t)}$ are extended with a readout layer, we showed in Section 5 that these can be represented in $\mathsf{TL}_2^{(t+1)}(\Omega)$. Theorem 4.4 and the results by Xu et al. (2019) and Barceló et al. (2020) then imply that $\rho_0(\mathsf{vwl}_1^{(t)})$ and $\rho_0(\mathsf{gcr}^{(t)})$ coincide with the separation power of these architectures with a readout layer. Here again, discrete labels need to be considered.

- Similarly, when $k$-FGNN or $(k+1)$-IGNs are used for graph embeddings, we can represent these in $\mathsf{TL}_{k+1}(\Omega)$ resulting again that their separation power coincides with that of $\mathsf{gwl}_k^{(\infty)}$. No restrictions are again in place on the vertex labels.

So for all these architectures, Corollary 6.2 applies and we can characterize the closures of these architectures in terms of functions that not more separating than their corresponding versions of cr or $k$-WL, as described in the main paper. In summary,

**Proposition F.3.** *For any $t \geq 0$:*

$$\overline{\mathsf{GIN}_\ell^{(t)}} = \{f : \mathcal{G}_1 \to \mathbb{R}^\ell \mid \rho_1(\mathsf{cr}^{(t)}) \subseteq \rho_1(f)\} = \overline{\mathsf{GTL}^{(t)}(\Omega)_\ell}$$

$$\overline{\mathsf{eGIN}_\ell^{(t)}} = \{f : \mathcal{G}_1 \to \mathbb{R}^\ell \mid \rho_1(\mathsf{vwl}_1^{(t)}) \subseteq \rho_1(f)\} = \overline{\mathsf{TL}_2^{(t)}(\Omega)_\ell}$$

*and when extended with a readout layer:*

$$\overline{\mathsf{GIN}_\ell^{(t)}} = \overline{\mathsf{eGIN}_\ell^{(t)}} = \{f : \mathcal{G}_0 \to \mathbb{R}^\ell \mid \rho_0(\mathsf{gwl}_1^{(t)}) \subseteq \rho_0(f)\} = \overline{\mathsf{TL}_2^{(t+1)}(\Omega)_\ell}.$$

*Furthermore, for any $k \geq 1$*

$$\overline{k\text{-}\mathsf{FGNN}_\ell^{(t)}} = \overline{k\text{-}\mathsf{GIN}_\ell^{(t)}} = \{f : \mathcal{G}_1 \to \mathbb{R}^\ell \mid \rho_1(\mathsf{vwl}_k^{(t)}) \subseteq \rho_1(f)\} = \overline{\mathsf{TL}_{k+1}^{(t)}(\Omega)_\ell}$$

$$\overline{(k+1)\text{-}\mathsf{IGN}_\ell} = \{f : \mathcal{G}_1 \to \mathbb{R}^\ell \mid \rho_1(\mathsf{vwl}_k^{(\infty)}) \subseteq \rho_1(f)\} = \overline{\mathsf{TL}_{k+1}(\Omega)_\ell}$$

*and when converted into graph embeddings:*

$$\overline{k\text{-}\mathsf{FGNN}_\ell} = \overline{k\text{-}\mathsf{GIN}_\ell} = \overline{(k+1)\text{-}\mathsf{IGN}_\ell} = \{f : \mathcal{G}_0 \to \mathbb{R}^\ell \mid \rho_0(\mathsf{gwl}_k^{(\infty)}) \subseteq \rho_0(f)\} = \overline{\mathsf{TL}_{k+1}(\Omega)_\ell},$$

*where the closures of the tensor languages are interpreted as the closure of the graph or graph/vertex functions that they can represent. For results involving GINs or eGINs, the graphs considered should have discretely labeled vertices.*

As a side note, we remark that in order to simulate CR on graphs with real-valued labels, one can use a GNN architecture of the form $\boldsymbol{F}_{v:}^{(t)} = \big(\boldsymbol{F}_{v:}^{(t-1)}, \sum_{u \in N_G(v)} \mathsf{mlp}(\boldsymbol{F}_{u:}^{(t-1)})\big)$, which translates in $\mathsf{GTL}^{(t)}(\Omega)$ as expressions of the form

$$\varphi_j^{(t)}(x_1) := \begin{cases} \varphi_j^{(t-1)}(x_1) & 1 \leq j \leq d_{t-1} \\ \sum_{x_2} E(x_1, x_2) \cdot \mathsf{mlp}_j\big(\varphi_1^{(t-1)}(x_1), \ldots, \varphi_{d_t}^{(t-1)}(x_1)\big) & d_{t-1} < j \leq d_t. \end{cases}$$

The upper bound in terms of CR follows from our main results. To show that CR can be simulated, it suffices to observe that one can approximate the function used in Proposition 1 in Maron et al. (2019b) to injectively encode multisets of real vectors by means of MLPs. As such, a continuous version of the first bullet in the previous proposition can be obtained.

## G  DETAILS ON TREEWIDTH AND PROPOSITION 4.5

As an extension of our main results in Section 4, we enrich the class of tensor language expressions for which connections to $k$-WL exist. More precisely, instead of requiring expressions to belong to $\mathsf{TL}_{k+1}(\Omega)$, that is to only use $k+1$ index variables, we investigate when expressions in $\mathsf{TL}(\Omega)$ are *semantically equivalent* to an expression using $k+1$ variables. Proposition 4.5 identifies a large class of such expressions, those of treewidth $k$. As a consequence, even when representing GNN architectures may require more than $k+1$ index variables, sometimes this number can be reduced. As a consequence of our results, this implies that their separation power is in fact upper bounded by $\ell$-WL for a smaller $\ell < k$. Stated otherwise, to boost the separation power of GNNs, the treewidth of the expressions representing the layers of the GNNs must have large treewidth.

We next introduce some concepts related to treewidth. We here closely follow the exposition given in Abo Khamis et al. (2016) for introducing treewidth by means variable elimination sequences of hypergraphs.

In this section, we restrict ourselves to summation aggregation.

### G.1  ELIMINATION SEQUENCES

We first define elimination sequences for hypergraphs. Later on, we show how to associate such hypergraphs to expressions in tensor languages, allowing us to define elimination sequences for tensor language expressions.

With a *multi-hypergraph* $\mathcal{H} = (\mathcal{V}, \mathcal{E})$ we simply mean a multiset $\mathcal{E}$ of subsets of vertices $\mathcal{V}$. An *elimination hypergraph sequences* is a vertex ordering $\sigma = v_1, \ldots, v_n$ of the vertices of $\mathcal{H}$. With such a sequence $\sigma$, we can associate for $j = n, n-1, n-2, \ldots, 1$ a sequence of $n$ multi-hypergraphs $\mathcal{H}_n^\sigma, \mathcal{H}_{n-1}^\sigma, \ldots, \mathcal{H}_1^\sigma$ as follows. We define

$$\mathcal{H}_n := (\mathcal{V}_n, \mathcal{E}_n) := \mathcal{H}$$
$$\partial(v_n) := \{F \in \mathcal{E}_n \mid v_n \in F\}$$
$$U_n := \bigcup_{F \in \partial(v_n)} F.$$

and for $j = n-1, n-2, \ldots, 1$ :

$$\mathcal{V}_j := \{v_1, \ldots, v_j\}$$
$$\mathcal{E}_j := (\mathcal{E}_{j+1} \setminus \partial(v_{j+1})) \cup \{U_{j+1} \setminus \{v_{j+1}\}\}$$
$$\partial(v_j) := \{F \in \mathcal{E}_j \mid v_j \in F\}$$
$$U_j := \bigcup_{F \in \partial(v_j)} F.$$

The *induced width* on $\mathcal{H}$ by $\sigma$ is defined as $\max_{i \in [n]} |U_i| - 1$. We further consider the setting in which $\mathcal{H}$ has some distinguished vertices. As we will see shortly, these distinguished vertices correspond to the free index variables of tensor language expressions. Without loss of generality, we assume that the distinguished vertices are $v_1, v_2, \ldots, v_f$. When such distinguished vertices are present, an elimination sequence is just as before, except that the distinguished vertices come first in the sequence. If $v_1, \ldots, v_f$ are the distinguished vertices, then we define the induced width of the sequence as $f + \max_{f+1 \le i \le n} |U_i \setminus \{v_1, \ldots, v_f\}| - 1$. In other words, we count the number of distinguished vertices, and then augment it with the induced width of the sequence, starting from $v_{f+1}$ to to $v_n$, hereby ignoring the distinguished variables in the $U_i$'s. One could, more generally, also try to reduce the number of free index variables but we assume that this number is fixed, similarly as how GNNs operate.

### G.2 CONJUNCTIVE TL EXPRESSIONS AND TREEWIDTH

We start by considering a special form of TL expressions, which we refer to as *conjunctive* TL expressions, in analogy to conjunctive queries in database research and logic. A conjunctive TL expression is of the form

$$\varphi(\boldsymbol{x}) = \sum_{\boldsymbol{y}} \psi(\boldsymbol{x}, \boldsymbol{y}).$$

where $\boldsymbol{x}$ denote the free index variables, $\boldsymbol{y}$ contains all index variables under the scope of a summation, and finally, $\psi(\boldsymbol{x}, \boldsymbol{y})$ is a product of base predicates in TL. That is, $\psi(\boldsymbol{x}, \boldsymbol{y})$ is a product of $E(z_i, z_j)$ and $P_\ell(z_i)$ with $z_i, z_j$ variables in $\boldsymbol{x}$ or $\boldsymbol{y}$. With such a conjunctive TL expression, one can associate a multi-hypergraph in a canonical way (Abo Khamis et al., 2016). More precisely, given a conjunctive TL expression $\varphi(\boldsymbol{x})$ we define $\mathcal{H}_\varphi$ as:

- $\mathcal{V}_\varphi$ consist of all index variables in $\boldsymbol{x}$ and $\boldsymbol{y}$;
- $\mathcal{E}_\varphi$: for each atomic base predicate $\tau$ in $\psi$ we have an edge $F_\tau$ containing the indices occurring in the predicate; and
- the vertices corresponding to the free index variables $\boldsymbol{x}$ form the distinguishing set of vertices.

We now define an *elimination sequence for* $\varphi$ as an elimination sequence for $\mathcal{H}_\varphi$ taking the distinguished vertices into account. The following observation ties elimination sequences of $\varphi$ to the number of variables needed to express $\varphi$.

**Proposition G.1.** *Let $\varphi(\boldsymbol{x})$ be a conjunctive* TL *expression for which an elimination sequence of induced with $k - 1$ exists. Then $\varphi(\boldsymbol{x})$ is equivalent to an expression $\tilde{\varphi}(\boldsymbol{x})$ in* $\mathsf{TL}_k$.

*Proof.* We show this by induction on the number of vertices in $\mathcal{H}_\varphi$ which are not distinguished. For the base case, all vertices are distinguished and hence $\varphi(\boldsymbol{x})$ does not contain any summation and is an expression in $\mathsf{TL}_k$ itself.

Suppose that in $\mathcal{H}_\varphi$ there are $p$ undistinguished vertices. That is,

$$\varphi(\boldsymbol{x}) = \sum_{y_1} \cdots \sum_{y_p} \psi(\boldsymbol{x}, \boldsymbol{y}).$$

By assumption, we have an elimination sequence of the undistinguished vertices. Assume that $y_p$ is first in this ordering. Let us write

$$\varphi(\boldsymbol{x}) = \sum_{y_1} \cdots \sum_{y_p} \psi(\boldsymbol{x}, \boldsymbol{y})$$
$$= \sum_{y_1} \cdots \sum_{y_{p-1}} \psi_1(\boldsymbol{x}, \boldsymbol{y} \setminus y_p) \cdot \sum_{y_p} \psi_2(\boldsymbol{x}, \boldsymbol{y})$$

where $\psi_1$ is the product of predicates corresponding to the edges $F \in \mathcal{E}_\varphi \setminus \partial(y_p)$, that is, those not containing $y_p$, and $\psi_2$ is the product of all predicates corresponding to the edges $F \in \partial(y_p)$, that is, those containing the predicate $y_p$. Note that, because of the induced width of $k-1$, $\sum_{y_p} \psi_2(\boldsymbol{x}, \boldsymbol{y})$ contains all indices in $U_p$ which is of size $\leq k$. We now replace the previous expression with another expression

$$\varphi'(\boldsymbol{x}) = \sum_{y_1} \cdots \sum_{y_{p-1}} \psi_1(\boldsymbol{x}, \boldsymbol{y} \setminus y_p) \cdot R_p(\boldsymbol{x}, \boldsymbol{y})$$

Where $R_p$ is regarded as an $|U_p| - 1$-ary predicate over the indices in $U_p \setminus y_p$. It is now easily verified that $\mathcal{H}_{\varphi'}$ is the hypergraph $\mathcal{H}_{p-1}$ corresponding to the variable ordering $\sigma$. We note that this is a hypergraph over $p-1$ undistinguished vertices. We can apply the induction hypothesis and replace $\varphi'(\boldsymbol{x})$ with its equivalent expression $\tilde{\varphi}'(\boldsymbol{x})$ in $\mathsf{TL}_k$. To obtain the expression $\tilde{\varphi}(\boldsymbol{x})$ of $\varphi(\boldsymbol{x})$, it now remains to replace the new predicate $R_p$ with its defining expression. We note again that $R_p$ contains at most $k-1$ indices, so it will occur in $\tilde{\varphi}'(\boldsymbol{x})$ in the form $R_p(\boldsymbol{x}, \boldsymbol{z})$ where $|\boldsymbol{z}| \leq k-1$. In other words, one of the variables in $\boldsymbol{z}$ is not used, say $z_s$, and we can simply replace $R_p(\boldsymbol{x}, \boldsymbol{z})$ by $\sum_{z_s} \psi_x(\boldsymbol{x}, \boldsymbol{z}, z_s)$. □

As a consequence, one way of showing that a conjunctive expression $\varphi(\boldsymbol{x})$ in $\mathsf{TL}$ is equivalently expressible in $\mathsf{TL}_k$, is to find an elimination sequence of induced width $k-1$. This in turn is equivalent to $\mathcal{H}_\varphi$ having a *treewidth* of $k-1$, as is shown, e.g., in Abo Khamis et al. (2016). As usual, we define the treewidth of a conjunctive expression $\varphi(\boldsymbol{x})$ in $\mathsf{TL}$ as the treewidth of its associated hypergraph $\mathcal{H}_\varphi$.

We recall the definition of treewidth (modified to our setting): A *tree decomposition* $T = (V_T, E_T, \xi_T)$ of $\mathcal{H}_\varphi$ with $\xi_T : V_T \to 2^\mathcal{V}$ is such that

- For any $F \in \mathcal{E}$, there is a $t \in V_T$ such that $F \subseteq \xi_T(t)$; and
- For any $v \in \mathcal{V}$ corresponding to a non-distinguished index variable, the set $\{t \mid t \in V_T, v \in \xi(t)\}$ is not empty and forms a connected sub-tree of $T$.

The *width of a tree decomposition* $T$ is given by $\max_{t \in V_T} |\xi_T(t)| - 1$. Now the treewidth of $\mathcal{H}_\varphi$, $\mathrm{tw}(\mathcal{H})$ is the minimum width of any of its tree decompositions. We denote by $\mathrm{tw}(\varphi)$ the treewidth of $\mathcal{H}_\varphi$. Again, similar modifications are used when distinguished vertices are in place. Referring again to Abo Khamis et al. (2016), $\mathrm{tw}(\varphi) = k-1$ is equivalent to having a variable elimination sequence for $\varphi$ of an induced width of $k-1$. Hence, combining this observation with Proposition G.1 results in:

**Corollary G.2.** *Let $\varphi(\boldsymbol{x})$ be a conjunctive $\mathsf{TL}$ expression of treewidth $k-1$. Then $\varphi(\boldsymbol{x})$ is equivalent to an expression $\tilde{\varphi}(\boldsymbol{x})$ in $\mathsf{TL}_k$.*

That is, we have established Proposition 4.5 for conjunctive $\mathsf{TL}$ expressions. We next lift this to arbitrary $\mathsf{TL}(\Omega)$ expressions.

### G.3 ARBITRARY $\mathsf{TL}(\Omega)$ EXPRESSIONS

First, we observe that any expression in $\mathsf{TL}$ can be written as a linear combination of conjunctive expressions. This readily follows from the linearity of the operations in $\mathsf{TL}$ and that equality and

inequality predicates can be eliminated. More specifically, we may assume that $\varphi(\boldsymbol{x})$ in TL is of the form

$$\sum_{\alpha \in A} a_\alpha \psi_\alpha(\boldsymbol{x}, \boldsymbol{y}),$$

with $A$ finite set of indices and $a_\alpha \in \mathbb{R}$, and $\psi_\alpha(\boldsymbol{x}, \boldsymbol{y})$ conjunctive TL expressions. We now define

$$\mathsf{tw}(\varphi) := \max\{\mathsf{tw}(\psi_\alpha) \mid \alpha \in A\}$$

for expressions in TL. To deal with expressions in $\mathsf{TL}(\Omega)$ that may contain function application, we define $\mathsf{tw}(\varphi)$ as the maximum treewidth of the expressions: (i) $\varphi_{\mathsf{nofun}}(\boldsymbol{x}) \in \mathsf{TL}$ obtained by replacing each top-level function application $f(\varphi_1, \ldots, \varphi_p)$ by a new predicate $R_f$ with free indices $\mathsf{free}(\varphi_1) \cup \cdots \cup \mathsf{free}(\varphi_p)$; and (ii) all expressions $\varphi_1, \ldots, \varphi_p$ occurring in a top-level function application $f(\varphi_1, \ldots, \varphi_p)$ in $\varphi$. We note that these expression either have no function applications (as in (i)) or have function applications of lower nesting depth (in $\varphi$, as in $(ii)$). In other words, applying this definition recursively, we end up with expressions with no function applications, for which treewidth was already defined. With this notion of treewidth at hand, Proposition 4.5 now readily follows.

## H  Higher-order MPNNs

We conclude the supplementary material by elaborating on $k$-MPNNs and by relating them to classical MPNNs (Gilmer et al., 2017). As underlying tensor language we use $\mathsf{TL}_{k+1}(\Omega, \Theta)$ which includes arbitrary functions ($\Omega$) and aggregation functions ($\Theta$), as defined in Section C.5.

We recall from Section 3 that $k$-MPNNs refer to the class of embeddings $f : \mathcal{G}_s \to \mathbb{R}^\ell$ for some $\ell \in \mathbb{N}$ that can be represented in $\mathsf{TL}_{k+1}(\Omega, \Theta)$. When considering an embedding $f : \mathcal{G}_s \to \mathbb{R}^\ell$, the notion of being represented is defined in terms of the existence of $\ell$ expressions in $\mathsf{TL}_{k+1}(\Omega, \Theta)$, which together provide each of the $\ell$ components of the embedding in $\mathbb{R}^\ell$. We remark, however, that we can alternatively include concatenation in tensor language. As such, we can concatenate $\ell$ separate expressions into a single expression. As a positive side effect, for $f : \mathcal{G}_s \to \mathbb{R}^\ell$ to be represented in tensor language, we can then simply define it by requiring the existence of a single expression, rather than $\ell$ separate ones. This results in a slightly more succinct way of reasoning about $k$-MPNNs.

In order to reason about $k$-MPNNs as a class of embeddings, we can obtain an equivalent definition for the class of $k$-MPNNs by inductively stating how new embeddings are computed out of old embeddings. Let $X = \{x_1, \ldots, x_{k+1}\}$ be a set of $k + 1$ distinct variables. In the following, $\boldsymbol{v}$ denotes a tuple of vertices that have at least as many components as the highest index of variables used in expressions. Intuitively, variable $x_j$ refers to the $j$th component in $\boldsymbol{v}$. We also denote the image of a graph $G$ and tuple $\boldsymbol{v}$ by an expression $\varphi$, i.e., the semantics of $\varphi$ given $G$ and $\boldsymbol{v}$, as $\varphi(G, \boldsymbol{v})$ rather than by $[\![\varphi, \boldsymbol{v}]\!]_G$. We further simply refer to embeddings rather than expressions.

We first define "atomic" $k$-MPNN embeddings which extract basic information from the graph $G$ and the given tuple $\boldsymbol{v}$ of vertices.

- **Label embeddings** of the form $\varphi(x_i) := \mathsf{P}_s(x_i)$, with $x_i \in X$, and defined by $\varphi(G, \boldsymbol{v}) := (\mathsf{col}_G(v_i))_s$, are $k$-MPNNs;

- **Edge embeddings** of the form $\varphi(x_i, x_j) := \mathsf{E}(x_i, x_j)$, with $x_i, x_j \in X$, and defined by

$$\varphi(G, \boldsymbol{v}) := \begin{cases} 1 & \text{if } v_i v_j \in E_G \\ 0 & \text{otherwise,} \end{cases}$$

  are $k$-MPNNs; and

- **(Dis-)equality embeddings** of the form $\varphi(x_i, x_j) := \mathbf{1}_{x_i \,\mathsf{op}\, x_j}$, with $x_i, x_j \in X$, and defined by

$$\varphi(G, \boldsymbol{v}) := \begin{cases} 1 & \text{if } v_i \,\mathsf{op}\, v_j \\ 0 & \text{otherwise,} \end{cases}$$

  are $k$-MPNNs.

We next inductively define new $k$-MPNNs from "old" $k$-MPNNs. That is, given $k$-MPNNs $\varphi_1(\boldsymbol{x}_1), \dots, \varphi_\ell(\boldsymbol{x}_\ell)$, the following are also $k$-MPNNs:

- **Function applications** of the form $\varphi(\boldsymbol{x}) := \mathbf{f}(\varphi_1(\boldsymbol{x}_1), \dots, \varphi_\ell(\boldsymbol{x}_\ell))$ are $k$-MPNNs, where $\boldsymbol{x} = \boldsymbol{x}_1 \cup \dots \cup \boldsymbol{x}_\ell$, and defined by

$$\varphi(G, \boldsymbol{v}) := \mathbf{f}\left(\varphi_1(G, \boldsymbol{v}|_{\boldsymbol{x}_1}), \dots, \varphi_\ell(G, \boldsymbol{v}|_{\boldsymbol{x}_\ell})\right).$$

Here, if $\varphi_i(G, \boldsymbol{v}|_{\boldsymbol{x}_i}) \in \mathbb{R}^{d_i}$, then $\mathbf{f} : \mathbb{R}^{d_1} \times \dots \times \mathbb{R}^{d_\ell} \to \mathbb{R}^d$ for some $d \in \mathbb{N}$. That is, $\varphi$ generates an embedding in $\mathbb{R}^d$. We remark that our function applications include concatenation.

- **Unconditional aggregations** of the form $\varphi(\boldsymbol{x}) := \mathsf{agg}_{x_j}^{\mathbf{F}}(\varphi_1(\boldsymbol{x}, x_j))$ are $k$-MPNNs, where $x_j \in X$ and $x_j \notin \boldsymbol{x}$, and defined by

$$\varphi(G, \boldsymbol{v}) := \mathbf{F}\left(\{\!\{\varphi_1(G, v_1, \dots, v_{j-1}, w, v_{j+1}, \dots, v_k) \mid w \in V_G\}\!\}\right).$$

Here, if $\varphi_1$ generates an embedding in $\mathbb{R}^{d_1}$, then $\mathbf{F}$ is an aggregation function assigning to multisets of vectors in $\mathbb{R}^{d_1}$ a vector in $\mathbb{R}^d$, for some $d \in \mathbb{N}$. So, $\varphi$ generates an embedding in $\mathbb{R}^d$.

- **Conditional aggregations** of the form $\varphi(x_i) := \mathsf{agg}_{x_j}^{\mathbf{F}}(\varphi_1(x_i, x_j) | E(x_i, x_j))$ are $k$-MPNNs, with $x_i, x_j \in X$, and defined by

$$\varphi(G, \boldsymbol{v}) := \mathbf{F}\left(\{\!\{\varphi_1(G, v_i, w) \mid w \in N_G(v_i)\}\!\}\right).$$

As before, if $\varphi_1$ generates an embedding in $\mathbb{R}^{d_1}$, then $\mathbf{F}$ is an aggregation function assigning to multisets of vectors in $\mathbb{R}^{d_1}$ a vector in $\mathbb{R}^d$, for some $d \in \mathbb{N}$. So again, $\varphi$ generates an embedding in $\mathbb{R}^d$.

As defined in the main paper, we also consider the subclass $k$-MPNNs$^{(t)}$ by only considering $k$-MPNNs defined in terms of expressions of aggregation depth at most $t$. Our main results, phrased in terms of $k$-MPNNs are:

$$\rho_1(\mathsf{vwl}_k^{(t)}) = \rho_1(k\text{-MPNNs}^{(t)}) \text{ and } \rho_0(\mathsf{gwl}_k) = \rho_0(k\text{-MPNNs}).$$

Hence, if the embeddings computed by GNNs are $k$-MPNNs, one obtains an upper bound on the separation power in terms of $k$-WL.

The classical MPNNs (Gilmer et al., 2017) are subclass of 1-MPNNs in which no unconditional aggregation can be used and furthermore, function applications require input embeddings with the same single variable ($x_1$ or $x_2$), and only $\mathbf{1}_{x_i = x_i}$ and $\mathbf{1}_{x_i \neq x_i}$ are allowed. In other words, they correspond to guarded tensor language expressions (Section 4.2). We denote this class of 1-MPNNs by MPNNs and by MPNNs$^{(t)}$ when restrictions on aggregation depth are in place. And indeed, the classical way of describing MPNNs as

$$\varphi^{(0)}(x_1) = (P_1(x_1), \dots, P_\ell(x_1))$$
$$\varphi^{(t)}(x_1) = \mathbf{f}^{(t)}\left(\varphi^{(t-1)}(x_1), \mathsf{aggr}_{x_2}^{\mathbf{F}^{(t)}}\left(\varphi^{(t-1)}(x_1), \varphi^{(t-1)}(x_2) | E(x_i, x_j)\right)\right)$$

correspond to 1-MPNNs that satisfy the above mentioned restrictions. Without readouts, MPNNs compute vertex embeddings and hence, our results imply

$$\rho_1(\mathsf{cr}^{(t)}) = \rho_1(\text{MPNNs}^{(t)}).$$

Furthermore, MPNNs with a readout function fall into the category of 1-MPNNs:

$$\varphi := \mathsf{aggr}_{x_1}^{\mathsf{readout}}(\varphi^{(t)}(x_1))$$

where unconditional aggregation is used. Hence,

$$\rho_0(\mathsf{gcr}^{(t)}) = \rho_0(\mathsf{gwl}_1^{(t)}) = \rho_0(1\text{-MPNNs}^{(t+1)}).$$

We thus see that $k$-MPNNs gracefully extend MPNNs and can be used for obtaining upper bounds on the seperation power of classes of GNNs.

