# OpenReview forum: "Expressiveness and Approximation Properties of Graph Neural Networks"
_ICLR.cc/2022/Conference — ICLR 2022 Oral_

### Official Review · Reviewer_KRNm · 2021-10-20

**Correctness:** 4
**Technical Novelty And Significance:** 3
**Empirical Novelty And Significance:** Not applicable
**Recommendation:** 8
**Confidence:** 4

**Main Review:**

## Strong Points

1. Evaluation of the expressive power of GNN is indeed not straightforward and needs a lot of architecture specific proofs. Indeed, we need model agnostic way to evaluate expressive power of any GNN.  Thus the main motivation of the paper is appropriate and might have a significant effect on the literature.

2. Paper was written well, quite clear and understandable. It includes enough theoretical proofs and examples on how to find the expressive power of some well-known GNNs.

## Clarifying the used WL test order

Clarifying the used WL test order would be great. In GNN papers, generally we call CR as 1-WL, then original 1-WL as 2-WL and so on and so forth. It seems that this paper follows (Cai et al 1992) notations. It means the GNN in (Maron et al.,2019b) , known to have 3-WL power in the original paper, has 2-WL power according to this paper. It would be better if this point is clearly mentioned.

## Weak Points

1. The way of evaluating expressive power by tensor language is not new at all. According to my knowledge, this kind of evaluation was recently proposed in [1]. They introduced Geerts' matrix query language (MATLANG) for GNN world and showed how to use it in order to evaluate some well known GNN's expressive power by write down given GNN's forward calculations in matrix form. Then, they figured the expressive power out by determining which language can explain GNN's forward calculations. Thus, In Separation Power sections "We do not have any general technique allowing us to expand these results for arbitrary GNNs" and in Related Work section "general matrix query languages are known,albeit not in the context of GNNs" are basically not true. Also [1] falsified the claim that "In summary, our paper draws new and interesting connections between tensor languages, GNN architectures and classic graph algorithms". I strongly think the connection between the tensor language and WL-test order of GNN models is interesting and valuable, but not novel. Definitely the differences between similar recent works should be discussed.

2. Showing how many iterations of the WL-test equivalent of the given GNN is trivial. It is well-known that an additional layer in GNN is equivalent to WL-test's additional iteration (coloring rounds). So in expressive power analysis, we can assume that we always have enough layers as we assume the WL test continues till the stabilization of colors. Introducing that depth parameter as a contribution seems a little exaggerated. In my mind, just WL test order equivalence is enough.

3. Even though the main claim is that by this paper, anyone can determine the GNN's expressive power easily, is not that straight forward. One needs to write down the GNN layer in tensor form. It seems possible while using summation and/or weighted summation for neighborhood aggregation. How about other aggregations? Can PNA ( GNN that uses different aggregation such as max, min, std) be evaluated by this framework? I have seen that in Appendix, there is an example for GraphSage's version which uses sum aggregation. But the main version of GraphSage uses max.  There is no analysis for max aggregator of GraphSage. Is it possible to do it? Also some GNN such as Chebnet can be more powerful than 1-WL in some certain cases. But in general it is not less powerful than 1-WL. Can the proposed method determine this special case? Can Chebnet expressive power analysis be at least on the appendix?

4. By Geerts' matrix query language MATLANG, we can determine the expressive power up to 3-WL (or 2-WL in Cai et al 1992 notation). This paper extends this matrix language in order to determine the expressive power of GNN which goes beyond the 3-WL. However, due to the memory and cpu complexity it is not that practical and I have never seen any GNN in practice whose expressive power is more than 3-WL. So the main contribution of the paper beside what MATLANG provides, seems not necessary in practice at least for now. I am suspicious if we really need more powerful than 3-WL GNN. Thus maybe these theoretical analyses will never be used.

5. One of the main practical advantages of this theoretical work would be to give us some insight on how to increase the expressive power of GNN. Even though it is mentioned in the abstract, I am not sure these insights are clear enough. Can we create a new GNN architecture to use these insights? I see that experimental work such that proposing a new GNN is not the main idea of the authors. But at least some clear examples on how to increase expressive power would be very valuable.


### Reference
[1] Muhammet Balcilar, Pierre Heroux, Benoit Gauzere, Pascal Vasseur, Sebastien Adam, and Paul Honeine. Breaking the limits of message passing graph neural networks. In Proceedings of the 38th International Conference on Machine Learning (ICML), 2021


**Summary Of The Paper:**

The paper proposed a novel way of evaluating the expressive power of any arbitrary GNNs in terms of WL test order using tensor language. The main motivation of this paper is that there is no straight forward expressive power evaluation tool so far. All existing methods need architecture specific proofs. Main challenge is the translating given GNN into tensor language. Later, they easily parametrize given GNN by two numbers which are the number of indices of tensor language ($k$ ) and number of summation depth ($t$). Then, they claimed that given GNN's expressive power is equal to $k-1$-WL test when it applies $t$ coloring rounds.

**Summary Of The Review:**

I think theoretical sounds of the paper is quite strong and the main motivation is valid.
I am slightly to lean to recommend acceptance for this paper. But  above 5 points should be addressed.

After rebuttal,the authors pointed out all my concerns very well. Therefore I would recommend clear acceptance for this work.

---

> ### Author Response · Authors · 2021-11-22
> **Response to weak point 5 (insights for GNNs)**
>
> **Insights for $\mathsf{GNN}$s.** The reviewer would like to see more examples of how the gained insights can be used to construct more powerful $\mathsf{GNN}$. These insights are, in some sense, implied by our main results:
>
> - To go beyond $\mathsf{CR}$ or $1$-$\mathsf{WL}$, one has to use $\mathsf{GNN}$s that, in terms of $\mathsf{TL}$, use more than two variables, or whose $\mathsf{TL}$ expressions have a treewidth larger than one. That is, their layer definitions need to use linear algebraic computations only expressible in $\mathsf{TL}_k$ for $k>2$. *In the revision, we further highlight this after Theorem 4.1 and in Section 5 when discussing treewidth (item 4).*
>  - For vertex embeddings, if one wants to build $\mathsf{GNN}$s that match $1$-$\mathsf{WL}$ (rather than $\mathsf{CR}$) in separation power, non-guarded $\mathsf{TL}_2$ layers are needed. For example, this can be achieved by allowing global aggregation in layers as in $\mathsf{eGIN}$s in Section 5. *We now further highlight this in Section 5 in the revision, when discussing $\mathsf{eGIN}$s.*

---

> ### Author Response · Authors · 2021-11-22
> **Response to weak point 4 (relevance of k-WL)**
>
> **Higher-order $\mathsf{GNN}$s.** The reviewer raises a concern that, especially in the context of higher-order $\mathsf{GNN}$s, our results may be primarily of theoretical interest. Furthermore, the reviewer indicates that $\mathsf{MATLANG}$ suffices for most practical $\mathsf{GNN}\text{s}$.
>
> We would like to point out that $\mathsf{TL}$ and in particular the fragments $\mathsf{TL}_k$ not only allow for the analysis of $k$-order $\mathsf{GNN}$s, but also of  "augmented'' $\mathsf{GNN}$s in which complex graph structural information is used (e.g., the $\mathsf{GSN}$s by Bouritsas et al. (2020), $\mathcal F$-$\mathsf{MPNN}$s by Barceló et al. (2021),  and Simplicial $\mathsf{MPNN}$s by Bodnar et al. (2021)). The latter kind of $\mathsf{GNN}$s are more practical than higher-order $\mathsf{GNN}$s, yet cannot be (provably) analyzed by $\mathsf{MATLANG}$ because they can exceed $2$-$\mathsf{WL}$ in separation power.  Furthermore, $\mathsf{TL}$ allows to relate summation depth to the  number of $k$-$\mathsf{WL}$ iterations, which is less transparent (and again impossible for $k>3$) in $\mathsf{MATLANG}$.
>
> In summary, we respectfully disagree that $\mathsf{MATLANG}$ suffices to analyse practical $\mathsf{GNN}$s formalisms. *In the revision, we provide more details on these augmented $\mathsf{GNN}$s in Section D.2. in the supplementary material and added a paragraph related to these architectures in Section 5 in the main paper.*
>
> - Giorgos Bouritsas, Fabrizio Frasca, Stefanos Zafeiriou, and Michael M. Bronstein. Improving graph neural network expressivity via subgraph isomorphism counting. In Graph Representation Learning and Beyond (GRL+) Workshop at the 37 th International Conference on Machine Learning, 2020.
> - Pablo Barceló, Floris Geerts, Juan L. Reutter, and Maksimilian Ryschkov. Graph neural networks  with local graph parameters.Neurips, 2021.
> - Cristian Bodnar, Fabrizio Frasca, Yuguang Wang, Nina Otter, Guido F. Montufar, Pietro Lio and Michael M. Bronstein. Weisfeiler and Lehman go topological: Message passing simplicial networks. ICML 2021.

---

> ### Author Response · Authors · 2021-11-22
> **Response to weak point 3 (ChebNet, spectral methods)**
>
> **Spectral GNNs.**  Another interesting question raised by the reviewer is whether spectral-based $\mathsf{GNN}$s can be analysed as well. And indeed, special kinds of spectral $\mathsf{GNN}$s can be analyzed. For example, ChebNet was analyzed in Balcilar et al. (2021) using connections to $\mathsf{MATLANG}$. We can carry out a similar (and slightly refined analysis of ChebNet resulting in bounds by colour refinement) when the use of the maximal eigenvalue of the graph Laplacian does not play a role (see also Theorem 2 in Balcilar et al. (2021). Using the guarded fragment we get a bound in terms of colour refinement rather than $1$-$\mathsf{WL}$, which makes a difference for vertex embeddings. As another example, we also show that CayleyNets can be analysed using tensor languages. This analysis seems to be new and nicely complements the spectral analysis of CayleyNets in Balcilar et al. (2021b).  We show that CayleyNets are bounded by $2$-$\mathsf{WL}$.
>
>  *In the revision, we now mention the Chebnet result in the paper and provide details in Section D.3, as requested by the reviewer. The analysis of CayleyNets can be found in Section D.3. as well.*
>
> In general, It is an interesting avenue for further research to obtain a better understanding of connections between $\mathsf{TL}_k$ and spectral properties. To our knowledge, the relation between $k$-$\mathsf{WL}$ and spectral properties is rather unexplored, except that $2$-$\mathsf{WL}$ equivalent graphs have the same spectrum (Dawar et al. 2019). Rattan et al. (2021) explore further connections between $k$-$\mathsf{WL}$ and spectral properties, but how all this connects to $\mathsf{TL}_k$ or $\mathsf{GNN}$s needs further investigation.
>
> - Muhammet Balcilar, Pierre Héroux, Benoit Gaüzère, Pascal Vasseur, Sébastien Adam, Paul Honeine: Breaking the limits of message passing graph neural networks. In Proceedings of the 38th International Conference on Machine Learning (ICML), 2021.
> - Muhammet Balcilar, Guillaume Renton, Pierre Héroux, Benoit Gaüzère, Sébastien Adam, Paul Honeine: Analyzing the Expressive Power of Graph Neural Networks in a Spectral Perspective, ICLR 2021b.
> - Anuj Dawar, Simone Severini, Octavio Zapata: Descriptive complexity of graph spectra. Ann. Pure Appl. Log. 170(9): 993-1007 (2019).
> - Gaurav Rattan, Tim Seppelt: Weisfeiler-Leman, Graph Spectra, and Random Walks. CoRR abs/2103.02972 (2021).

---

> ### Author Response · Authors · 2021-11-22
> **Response to weak point 3 (Other aggregation functions)**
>
> **Other aggregation functions.** The reviewer asks whether $\mathsf{GNN}$s that use aggregation functions, other than summation, can be analysed as well. We thank the reviewer for raising this question. We can indeed deal with any other aggregation function but decided to only introduce summation in order not to overload the syntax of $\mathsf{TL}$. Our results can be generalised when $\mathsf{TL}$ is extended with a collection (say $\Theta$) of aggregation functions and such that in the syntax of $\mathsf{TL}$ one can write “$\mathsf{aggr}(F,x,e)$”, with $F$ an aggregation function (such as max, stdv, min,...) in $\Theta$. The notion of summation depth is then replaced by a notion of aggregation depth. It requires some additional notation and minor modification of the proof of Proposition C.3. Of course, the choice of aggregation functions matter when $\mathsf{CR}$ or $k$-$\mathsf{WL}$ lower bounds are required, as is well known from the seminal work by Xu et al. (2019).
>
> *In the revision, we now state in Section 3  that $\mathsf{TL}$ can be generalised to include arbitrary aggregation functions that work on multisets of values and hint in a footnote how this can be done. In the supplementary material (Section C.5) we describe what changes are needed. In addition, we mention GraphSage using different aggregation functions and PNAs in Section 5, and give their analysis in Section D.1. as requested by the reviewer.*

---

> ### Author Response · Authors · 2021-11-22
> **Response to weak point 3 (ease of using TL)**
>
> **Ease of writing $\mathsf{GNN}$s in $\mathsf{TL}$.** The reviewer finds that casting $\mathsf{GNN}$s in $\mathsf{TL}$ is not that straightforward as one needs to write down $\mathsf{GNN}$s in tensor form. Indeed, the complexity of writing down $\mathsf{GNN}$s in $\mathsf{TL}$ is related to the complexity of the $\mathsf{GNN}$s under consideration. For complex $\mathsf{GNN}$s, defining $\mathsf{TL}$ expressions will inherit the $\mathsf{GNN}$s model complexity. In most cases that we encountered, however, writing down the $\mathsf{TL}$ expression simply required writing down, almost verbatim, the definitions of $\mathsf{GNN}$s as found in the literature. For $\mathsf{GNN}$s that use tensors, for example $k$-$\mathsf{FGNN}$s and $k$-$\mathsf{IGN}s$, the definitions given in e.g., Maron et a.l (2019) are already using explicit tensor manipulations, which translate easily into $\mathsf{TL}$.
>
> We also want to reiterate that once the $\mathsf{TL}$ expressions are obtained, no further analysis in terms of $\mathsf{CR}$ or $k$-$\mathsf{WL}$ is needed, as this follows from our results. We believe that writing down $\mathsf{TL}$ expressions should be considerably easier than proving bounds in terms of $k$-$\mathsf{WL}$ from scratch for each $\mathsf{GNN}$ model. This is really where the advantage of $\mathsf{TL}$ comes into play. Finally, when implementing $\mathsf{GNN}$s in practice one often has to think in terms of explicit tensor manipulation.
>
> - Haggai Maron, Heli Ben-Hamu, Hadar Serviansky, Yaron Lipman: Provably Powerful Graph Networks, Neurips 2019.

---

> ### Author Response · Authors · 2021-11-22
> **Response to weak point 2 (layer/iteration connection)**
>
> **Fine-grained analysis in terms of number of layers/iterations.** The reviewer finds the fine-grained analysis of the separation power in terms of the number of iterations of $k$-$\mathsf{WL}$ not so relevant and trivial. We here respectfully disagree with the reviewer. While this may be immediate for $\mathsf{CR}$ or maybe $1$-$\mathsf{WL}$, for more complex architectures such as $\mathsf{GCN}$s, $k$-$\mathsf{IGN}$s, $\mathsf{GSN}$s, etc., the connection between layers and iterations of $k$-$\mathsf{WL}$ is quite intricate.
>
> We solve this problem by connecting the number of rounds of $k$-$\mathsf{WL}$ with the summation depth of $\mathsf{TL}$ expressions used to model  $\mathsf{GCN}$s, $k$-$\mathsf{IGN}$s, $\mathsf{GSN}$s, etc. A nice example of this are the $k$-$\mathsf{IGN}$s, where the summation depth needed  for each layer precisely tells how many steps the $k$-$\mathsf{WL}$ test needs to do, to simulate a single $k$-$\mathsf{IGN}$ layer. This gives additional insights of what the equivariant layers can do.
>
> As such, our contribution goes beyond simply connecting layers to iterations. Furthermore, the number of layers is an important parameter of $\mathsf{GNN}$s in practice. No $\mathsf{GNN}$ architectures can have as many layers as needed to compute the stable $k$-$\mathsf{WL}$ colouring as this depends on the graph (size). Understanding the limitations of finite layered architectures is therefore desired. Finally, it is known that using more iterations of $k$-$\mathsf{WL}$ adds separation power, similarly as using more variables adds to the separation power. Hence understanding the iterations needed allows us to compare different architectures, or even the same architecture with a different number of layers.

---

> ### Author Response · Authors · 2021-11-22
> **Response to weak point 1 (novelty)**
>
> **Novelty.** The reviewer correctly points out that Baliclar et al. (2021) already connect $\mathsf{GNN}$s to an, albeit different, matrix query language $\mathsf{MATLANG}$ (Brijder et al. 2019), and use known connections between $\mathsf{MATLANG}$ and $1$-$\mathsf{WL}$ and $2$-$\mathsf{WL}$, as established in Geerts (2021), to gain upper bounds on the separation power of ($1$st-order) $\mathsf{GNN}$s. In this way, Baliclar et al. (2021) provide additional motivation for our work and we thank the reviewer for pointing out this reference.
>
> *In the revision, we now both include Baliclar et al (2021) as additional motivation for our work and also describe it in the related work section (both in the main paper and in Section A in the supplementary material.*
>
> In our opinion, Baliclar et al. (2021)  does not reduce the novelty of our approach. In contrast to Baliclar et al. (2021), we use a new matrix query language ($\mathsf{TL}$) which allows us to make new and important observations: the separation power of $\mathsf{GNN}$s is related to the number of index variables (or more generally treewidth) and the summation depth of $\mathsf{TL}$ expressions needed to define the layers of $\mathsf{GNN}$s. In addition, whether guarded or unguarded expressions are needed, relates to whether $\mathsf{CR}$ or $1$-$\mathsf{WL}$ are required to assess the separation power of some $\mathsf{GNN}$s. These connections are, to our knowledge, new and are not reported (and neither easily follow from) Baliclar et al. (2021) or previous works on matrix query languages. The simple quantitative measures (number variables, treewidth, summation depth) are the distinguishing and new aspects in our work compared to previous works.
>
> We also want to clarify some statements in our paper with which the reviewer disagrees. With "We do not have any general technique allowing us to expand these results for arbitrary $\mathsf{GNN}$s", we mean that no previous approach, whether based on matrix query languages or other any other method, allows for the analysis of the separation power of general $\mathsf{GNN}$s. The $\mathsf{GNN}$s considered in Baliclar et al. (2021) are restricted to those representable in $\mathsf{MATLANG}$, which is a much less powerful and less general matrix query language than $\mathsf{TL}$. Indeed, $\mathsf{MATLANG}$ can be seen to be subsumed by $\mathsf{TL}_3$. but  even when restricted to $\mathsf{GNN}$s that are bounded by $2$-$\mathsf{WL}$, it is not so immediate to cast, say $2$-$\mathsf{FGNN}$s in $\mathsf{MATLANG}$. As another example,  standard $\mathsf{GNN}$s and $\mathsf{GCN}$s can be represented in $\mathsf{MATLANG}$, but when writing them in $\mathsf{TL}$, one sees that a different summation depth is needed, and hence they are bounded by a different number of iterations of $\mathsf{CR}$.  *In the revision we rephrased the sentence and now state "We provide a tensor language-based technique to analyse the separation power of GNNs".*
>
> Similarly, when we say "general matrix query languages are known, albeit not in the context of GNNs", we mean again that, to our knowledge, no specification language geared towards $\mathsf{GNN}$s has been proposed. Bacilcar et al. (2021) use an existing language ($\mathsf{MATLANG}$) and Geerts et al. (2021b) propose more general matrix query languages in which computations can be done beyond what is needed for $\mathsf{GNN}$s. Our language $\mathsf{TL}$ is designed for $\mathsf{GNN}$s and we can easily model classical $\mathsf{GNN}$s, higher-order $\mathsf{GNN}$s, and a variety of newer models in which classical $\mathsf{GNN}$s are extended with additional graph structural information. *In the revision we reworked the related work entirely.*
>
> Finally, the statement "In summary, our paper draws new and interesting connections between tensor languages, $\mathsf{GNN}$ architectures and classic graph algorithms" refers to the introduction of $\mathsf{TL}$, a specification language for general $\mathsf{GNN}$s, and the connection between the number of variables (treewidth) and summation depth and $k$-$\mathsf{WL}$. All of this is, to our knowledge, new, and not covered by any previous works. Furthermore, $\mathsf{TL}$ allows for generalising previously known approximation results. *We therefore kept this sentence in the revision.*
>
> - Muhammet Balcilar, Pierre Héroux, Benoit Gaüzère, Pascal Vasseur, Sébastien Adam, Paul Honeine: Breaking the limits of message passing graph neural networks. In Proceedings of the 38th International Conference on Machine Learning (ICML), 2021.
> - Robert Brijder, Floris Geerts, Jan Van den Bussche, Timmy Weerwag: On the Expressive Power of Query Languages for Matrices. ACM Trans. Database Syst. 44(4): 15:1-15:31 (2019).
> - Floris Geerts: On the Expressive Power of Linear Algebra on Graphs. Theory Comput. Syst. 65(1): 179-239 (2021).
> - Floris Geerts, Thomas Muñoz, Cristian Riveros, Domagoj Vrgoc: Expressive Power of Linear Algebra Query Languages. PODS 2021b

---

> ### Author Response · Authors · 2021-11-22
> **Answer to reviewer 4**
>
> We thank the reviewer for the detailed reviews and insightful comments which we will next address in turn.
>
> **Clarification of the order of $\mathsf{WL}$-tests.** It is indeed confusing that some works on the separation power of $\mathsf{GNN}$s use the "folklore" version of $k$-$\mathsf{WL}$ while other works use the "oblivious" version $k$-$\mathsf{OWL}$. We use the folklore version of $k$-$\mathsf{WL}$ which, roughly speaking, corresponds to $(k+1)$-$\mathsf{OWL}$. Colour refinement ($\mathsf{CR}$) is not precisely the same as our $1$-$\mathsf{WL}$, however, when it comes to vertex embeddings. We refer to Grohe (2021) for a detailed comparison between $k$-$\mathsf{WL}$, $k$-$\mathsf{OWL}$ and $\mathsf{CR}$.
>
> *In the revision, we now clearly state in Section 2, that we use the "folklore" version of Cai et al. (1992).* Thanks for pointing this out.
>
> - Martin Grohe: The Logic of Graph Neural Networks. LICS, 2021.
> - Jin-yi Cai, Martin Fürer, Neil Immerman: An optimal lower bound on the number of variables for graph identifications. Comb. 12(4): 389-410 (1992).

---

> ### Comment · Reviewer_KRNm · 2021-11-22
> **General Response to Authors**
>
> Indeed the answers pointed out my concerns very well. Now the connection to the similar work is solid. Especially extending the analysis on higher power GNN such as CWN, GSN and MPNN which uses different aggregation are highly appreciated. I therefore increase the mark to 8.

---

> > ### Author Response · Authors · 2021-11-27
> > **Response to comment on revision**
> >
> > The reviewer's suggestions concerning related work, the use of other aggregation function and the inclusion of spectral GNNs were all very valuable for making the revision. We are pleased that the reviewer is now fully satisfied with the revision.

---

### Official Review · Reviewer_QRwU · 2021-10-28

**Correctness:** 4
**Technical Novelty And Significance:** 3
**Empirical Novelty And Significance:** Not applicable
**Recommendation:** 8
**Confidence:** 4

**Main Review:**

The unified framework provided by TL is interesting, and allows a more uniform study of GNN models, and a simpler means to derive expressiveness bounds for new GNN models. The study of TL is also very well-grounded in the literature, as it includes most of the key works on GNN expressive power, and additionally confirms their findings. Furthermore, TL establishes a new set of results, thereby addressing some open questions in the field. In particular, it produces interesting insights about the effect of more layers, connecting this explicitly to treewidth and thus adding a nice nuance to this discussion. The results also appear intuitive and sound, though I did not check these thoroughly. Hence, the paper seems to be a valuable addition to the literature on GNNs.

Nonetheless, I find that the novelty of the approach is limited, particular relative to other frameworks characterizing GNNs. In particular, I cannot see the novelty proposed by TL relative to the matrix query languages mentioned in the paper. Therefore, I strongly suggest a more through comparison with related work. Furthermore, it is not clear how much simpler writing a TL expression for GNNs is, particularly with respect to non-standard GNNs involving, e.g, sub-structure counting. Hence, a more detailed presentation of specific model TL expressions, including some from the supplementary material, should be added into the paper to more clearly present the TL translation process. On a more minor note, the current writing style and notation are hard to follow. I refer specifically to Page 3, and the explanation of cr, gwl,vwl, etc. This section required multiple reads to be properly understood, and is quite dense. This is also true of other parts in the paper, e.g., Page 7. Therefore, a more simplified presentation, supported by examples, would be beneficial. Finally, the authors can also consider studying their framework with respect to classes of functions (or universality), as in Barcelo et al. [1], rather than just function approximation relative to separating power, to provide a more holistic overview of the GNN landscape. Alternatively, illustrations of functions with separation power limits can also benefit the presentation in this part of the paper.

[1] Barcelo et al. The logical expressiveness of graph neural networks. ICLR 2020.

**Summary Of The Paper:**

The paper proposes Tensor Language (TL), a language with which popular GNN models can be uniformly studied, so as to yield insights about their expressiveness and separation power. More specifically, the paper shows how TL corresponds to the WL hierarchy, and shows that summation depth in TL, as well as the number of index variables, correspond directly to iterations and tuple size for k-WL, respectively. Moreover, the paper establishes a direct correspondence between GNN update equations and expressions in TL, thus offering a simpler means to understand the expressive and separation power of GNNs, even considering their number of layers, by mapping their architectures to expressions in TL. TL , as well as its guarded fragment GTL, are then used to re-establish a set of known GNN results from the literature.

Beyond expressive power, TL is also used to quantify the function approximation power of GNNs, and also shows that GNNs, characterized by a TL fragment, can learn functions with separation power upper-bounded by a refinement algorithm corresponding to this fragment. Finally, the TL framework is used to establish new results. In particular, it shows that k-IGN cannot achieve expressiveness beyond (k-1)-WL, as this model corresponds to tk iterations of (k-1)-WL, and also offers insights as to the expressiveness of k-IGN with a polynomial number of layers: Indeed, model power does not increase with more standard layers, e.g., those relying on the adjacency matrix, but rather, any increase in layers can only improve expressiveness by deriving GNN functions with e.g., increased treewidth.

**Summary Of The Review:**

The paper's main proposal, TL, offers a novel and simple means to characterize the power of GNNs irrespective of specific design choices, and helps derive some new results for GNNs. All in all, the paper makes good contributions, but some of its claims, namely the simplicity of TL translation, as well as its novelty, should be better explained.

---

> ### Author Response · Authors · 2021-11-22
> **Answer to Novelty**
>
> We thank the reviewer for the positive review and suggestions of how to improve the paper.
>
> **Novelty.**  We wish to emphasise that it is the study of the separation power of a general tensor language (and its use for assessing the separation power of general $\mathsf{GNN}$s) in terms of the number of index variables, treewidth and summation depth, that constitutes the novelty of our work. Such an analysis was not carried out for previous matrix query languages. To clarify the connections with previous matrix query languages, *in the revision, we have expanded the related work section in the main paper and provide a more detailed comparison with other matrix query languages in the supplementary material (Section A).*
>
> More specifically, as is now mentioned in the related work section, $\mathsf{TL}$ is inspired by the “$\mathsf{sum}$-$\mathsf{MATLANG}$” matrix query language in Geerts et al. (2021b) but specialised to $\mathsf{GNN}$s and graphs. Furthermore,  $\mathsf{TL}$ can be used to define arbitrary tensors, whereas $\mathsf{sum}$-$\mathsf{MATLANG}$ works on general matrices but is defined such that only scalars, vectors or matrices can be returned. Furthermore, $\mathsf{TL}$ uses free vector variables, which combined with the notion of summation depth allows for a much more precise connection with logics, $\mathsf{CR}$ and $k$-$\mathsf{WL}$. Free variables are needed to represent vertex embeddings. Moreover, neither separation power nor connections to $\mathsf{CR}$ and $k$-$\mathsf{WL}$ were considered for $\mathsf{sum}$-$\mathsf{MATLANG}$. Indeed, for $\mathsf{sum}$-$\mathsf{MATLANG}$, only an equivalence to the positive fragment of the relational algebra (on real-valued relations) is known (Geerts et al. (2021b)).
>
> An earlier matrix query language, $\mathsf{MATLANG}$ (Brijder et al. (2019)) was connected to $1$-$\mathsf{WL}$ and $2$-$\mathsf{WL}$ in Geerts (2021), yet $\mathsf{MATLANG}$ is designed in a different way by only allowing a specific set of matrix operations. $\mathsf{MATLANG}$ can be shown to be included in $\mathsf{TL}_3$, and again, $\mathsf{MATLANG}$ can only output scalars, vectors or matrices. No notion of free variables or summation depth was considered in $\mathsf{MATLANG}$. $\mathsf{MATLANG}$ has, however, been used before to analyse $\mathsf{GNN}$s (Balcilar et al. (2021))
>
> In summary, $\mathsf{TL}$ is different from previously proposed matrix query languages and furthermore, the connection to finite variable fragments of logics is novel as are the results on the separation power of $\mathsf{TL}$ (Theorem 4.2, 4.4) . We also want to emphasise that the guarded fragment and its relation to colour refinement was not considered before in the matrix query language literature (Theorem 4.3), and neither was the bound on the number of indices with respect to the treewidth of expressions (Theorem 4.5).
>
> * Robert Brijder, Floris Geerts, Jan Van den Bussche, Timmy Weerwag: On the Expressive Power of Query Languages for Matrices. ACM Trans. Database Syst. 44(4): 15:1-15:31 (2019).
> * Floris Geerts: On the Expressive Power of Linear Algebra on Graphs. Theory Comput. Syst. 65(1): 179-239 (2021).
> * Floris Geerts, Thomas Muñoz, Cristian Riveros, Domagoj Vrgoc: Expressive Power of Linear Algebra Query Languages. PODS 2021.
> * Muhammet Balcilar, Pierre Héroux, Benoit Gaüzère, Pascal Vasseur, Sébastien Adam, Paul Honeine: Breaking the limits of message passing graph neural networks. ICML 2021.

---

> ### Author Response · Authors · 2021-11-22
> **Answer to other frameworks**
>
> **Other frameworks.** The reviewer mentions “other frameworks characterising $\mathsf{GNN}$s”, but we are not aware of any other framework as general as ours. Of course, there is the message-passing neural network formalism used in e.g. Barcelo et al. 2020, but these vanilla $\mathsf{MPNN}$s only allow analysing $\mathsf{GNN}$s whose separation power is bounded by $\mathsf{CR}$ (on vertex level) and $1$-$\mathsf{WL}$ on the graph level (or, at most, $1$-$\mathsf{WL}$ at the vertex level by using global readouts).
> *If the reviewer could point out more general or other frameworks, then we would be happy to include them in our related work.*
>
> * Pablo Barceló, Egor V. Kostylev, Mikaël Monet, Jorge Pérez, Juan L. Reutter, Juan Pablo Silva: The Logical Expressiveness of Graph Neural Networks. ICLR 2020.

---

> > ### Comment · Reviewer_QRwU · 2021-11-27
> > **Reviewer Response**
> >
> > I was mostly referring to a better discussion of matrix query languages and a presentation of the contributions made by TL, and this has been addressed. However, in terms of other frameworks and expressiveness studies, you should also include approaches based on indivualization/randomization to study/improve expressiveness in Section 2.
> >
> > [1] Sato et al. Random Features Strengthen Neural Networks. ICDM 2021.
> > [2] Abboud et al. The Surprising Power of Graph Neural Networks with Random Node Initialization. IJCAI 2021.
> > [3] Dasoulas et al. Coloring Graph Neural Networks for Node Disambiguation. IJCAI 2020.
> >
> > These papers offer a means to improve on the expressiveness power of standard GNNs, with [2,3] providing universality results.

---

> > > ### Author Response · Authors · 2021-11-27
> > > **Comment on suggested related work**
> > >
> > > We thank the reviewer for the clarification and will include the suggested works in our related work, both when talking about expressive power and universality. We are aware of these works, which use a completely different approach to boost expressive power than what we consider. It would, in fact, be interesting to understand how the separation power of our TL language changes (and what kind of analysis can be made) when random vectors (for random vertex initialisation) are supported.

---

> ### Author Response · Authors · 2021-11-22
> **Answer to ease of using tensor languages**
>
> **Ease of writing $\mathsf{GNN}$s in $\mathsf{TL}$.**  The reviewer mentions that it is not clear that writing $\mathsf{GNN}$s in $\mathsf{TL}$ is easier. We do propose $\mathsf{TL}$  to be able to analyze $\mathsf{GNN}$s and not as a new (or easier) specification language for $\mathsf{GNN}$s. Where the value of our proposal comes from is that, once $\mathsf{GNN}$s are represented in $\mathsf{TL}$ , bounds on their separation power follow from our general results on $\mathsf{TL}$ . We believe that writing $\mathsf{GNN}$s in $\mathsf{TL}$  is easier than proving bounds on the separation power directly for $\mathsf{GNN}$s. So “easier” refers to showing bounds and not to writing $\mathsf{GNN}$s.
>
> The modelling of $\mathsf{GNN}$s in terms of $\mathsf{TL}$  is, however, unavoidably related to the complexity of the $\mathsf{GNN}$s under consideration. For more complex $\mathsf{GNN}$s, their description in the literature will be more involved, as will the $\mathsf{TL}$  expressions needed. Due to space limitations, we only showed $\mathsf{TL}$  expressions for simple $\mathsf{GNN}$s, deferring more complex $\mathsf{GNN}$s translations to the supplementary material. As can be seen there, the $\mathsf{TL}$  translations are in most cases verbatim translations of the $\mathsf{GNN}$ layer definitions. Exceptions are indeed models like $\mathsf{GSN}$s or simplicial $\mathsf{GNN}$s, where one basically has to express all additional computations in $\mathsf{TL}$  in order to understand their separation power.
> In response to the reviewer’s request for more examples, *in the revision, we could only find space for one more example, $2$-$\mathsf{FGNN}$s, and the addition of a paragraph (Section 5) providing some general intuition on how $\mathsf{GSN}$s, $\mathcal{F}$-$\mathsf{MPNN}$s, and Simplicial $\mathsf{GNN}$s can be analysed. We also provide additional details in the supplementary material (Section D.2, “augmented GNNs”).*

---

> ### Author Response · Authors · 2021-11-22
> **Answer to presentation**
>
> **Presentation.**  We are aware that the presentation is sometimes dense. We tried our best to include concise yet correct definitions in the paper and the page limit necessarily limits to what extent things can be explained. *In the revised version, we did a pass over the paper and attempted to make the things more clear, with the focus on page 3 and page 7, as suggested by the reviewer.* For page 7, the reviewer would like to see more examples. We again are limited by the space limitation and as mentioned in the other comments *only found a place for one more example.*

---

> ### Author Response · Authors · 2021-11-22
> **Answer to expressibility of functions**
>
> **Expressing functions.** The reviewer suggests also to include some more holistic overview of the $\mathsf{GNN}$ landscape in terms of expressible functions (as in Barceló et al (2020)). *In the revision, we now describe how our results can help in designing $\mathsf{GNN}$s for certain functions  based on the limit of the separation power. (Section 4)*. To characterise what functions can be computed using $\mathsf{GNN}s$ would require to lift the results in Barceló et. al. (2020) to $\mathsf{TL}_k$. This in turn would require new interpolation theorems for bounded variable logics, which we are not aware of.
>
> We would  like to note that our results can, to some extent, be understood as an extension of Barceló et al. (2020) notion of universality from logical classifiers to the general setting of arbitrary classifiers. Whereas they show that guarded modal logic, $\mathsf{MPNN}$s and $\mathsf{CR}$ are equivalent in expressive power when restricted to logical classifiers, we show that $\mathsf{GTL}$, $\mathsf{MPNN}$s and $\mathsf{CR}$ are somewhat equivalent also in the general setting, in the sense that both $\mathsf{MPNN}$s and $\mathsf{GTL}$ formulas can approximate any graph classifier bounded by $\mathsf{CR}$.
>
> * Pablo Barceló, Egor V. Kostylev, Mikaël Monet, Jorge Pérez, Juan L. Reutter, Juan Pablo Silva: The Logical Expressiveness of Graph Neural Networks. ICLR 2020.

---

> ### Comment · Reviewer_QRwU · 2021-11-27
> **Response to Authors**
>
> I am satisfied with the changes made to the paper. It is now much clearer what the contributions of the work are. I have replied to your request in the "Answer to other frameworks" comment, and have listed some relevant references on universality/approximation results for GNNs, based on the invidualization/randomization paradigm, to include in the paper.
>
> All in all, my concerns have been well-addressed. I thus increase my score.

---

> > ### Author Response · Authors · 2021-11-27
> > **Response to comment on revision**
> >
> > We are pleased that the reviewer is satisfied with the revision. The additional suggested references will be included in the related work (see also answer to "Answer to other frameworks" comment).

---

### Official Review · Reviewer_tPHz · 2021-11-02

**Correctness:** 4
**Technical Novelty And Significance:** 3
**Empirical Novelty And Significance:** Not applicable
**Recommendation:** 8
**Confidence:** 2

**Details Of Ethics Concerns:**

N.A.

**Main Review:**

# Post-rebuttal Comments (11/29/2021)

I appreciate the authors for their response. I am satisfied with their answers. So, I want to keep my score and vote for acceptance.

# Initial Comments

【Strength】
- [1] This paper gives a general procedure for deriving the upper bound of expressive power for any GNN as long as we can translate it into the tensor language.
- [2] Theorem 6.1 and Corollary 6.2 allow us to give lower bounds on the expressive power of GNNs as well.

【Weakness】
- [1] Compared to the upper bound, it is more difficult to show the lower bound of expressive power because the former needs individual proof for concrete GNNs.
- [2] Those familiar with GNNs but not with the first-order logic may be hard to follow up the discussion of the paper.

【Correctness】
- [1] As far as I checked, I could not find any inappropriate points in the proofs.
- [2] What I am wondering is that in Theorem 4.1, the equivalence relation $\rho_1(\mathsf{TL}_{k+1}(\Omega))$ is independent of the choice of $\Omega$. Considering the case of $\Omega=\emptyset$, does this result imply that the separation power of the tensor language does not increase even if we add expressive functions such as non-linear activation functions or MLPs?

【Technical Novelty And Significance】
- [1] As mentioned in this paper, several studies such as [Barcelo et al., 2020 ] have studied the expressive power of GNNs via first-order logic. However, while [Barcelo et al., 2020] related GNNs to the existing first-order logic (namely, $\mathrm{FOC}_2$), this paper defined a new grammar, the tensor language $\mathsf{TL}$, and related GNNs to the first-order logic via this language. In this sense, the approach of this paper is novel.
- [2] This approach allows us to analyze expressive power independently of particular GNNs. In addition, it is relatively easy to derive the expressive power of a GNN because it is sufficient to translate the GNN into the tensor language. Therefore, I think this approach is significant.
- [3] There is no easy and general way to obtain the guarantees for the lower bound of the expressive power compared with the upper bound. We have to check the sufficient condition of Corollary 6.2 for each GNN, which needs proof tailored to the GNN. If we can find a general approach to obtain the guarantees, it would increase the paper's significance. (However, it is also notable that this paper obtained lower bounds systematically to some extent, as mentioned in Proposition E.3.)
- [4] The paper gives positive answers to the unresolved issues raised by existing studies. In this sense, this paper is significant.

【Empirical Novelty And Significance】
- [1] This paper does not have numerical experiments.

【Detailed Comments】
- [1] P.3: I think it is better to write the definition of irreflexivity as it is not well-known.
- [2] P.3: Initially, for a graph $G$ and $\mathbfit{v}\in V^k_G$, ... → I am afraid this sentence is hard to understand, especially the part "where, atp_k(G, v) is the atomic type...". Could you reconsider the sentence?
- [3] P.4: I think $S$ is undefined.
- [4] P.6: The definition of $\mathsf{C}_{k+1}$ does not appear in the main text (only available in Appendix).
- [5] P.13: $\pi_{\sigma\star G}\textlbrackdbl \varphi_1, \sigma \star \nu \textrbrackdbl \cdot \pi_{\star G}\textlbrackdbl \varphi_2, \sigma \star \nu \textrbrackdbl$ → $\star G$ should be $\sigma\star G$.
- [6] P.15: e..g., → e.g.,
- [7] P.16: Here the unravelling is the (infinite tree ... → remove the parenthesis
- [8] P.17: $\pi_{H}\textlbrackdbl \varphi_1, \mu[x_i \to v] \textrbrackdbl$ → $\pi_{G}\textlbrackdbl \varphi_1, \mu[x_i \to v] \textrbrackdbl$

**Summary Of The Paper:**

This paper defined formal languages for describing tensor expressions. It showed that their expressive power is identical to (suitably parameterized) color refinement algorithms or (vertex/graph) WL algorithms. Then, by translating existing GNNs into the tensor languages, this paper provided upper bounds for the expressive power of GNNs, which recovered several existing results. In addition, this paper characterized the closure of a function class by functions whose expressive power is equal to or less than the functions (Theorem 6.1). As an application, this paper characterized the expressive power of several GNNs via the tensor language (Corollary 6.2, Proposition E.3).

**Summary Of The Review:**

As far as I checked, I could not find any incorrect points in the proofs. I think this paper is technically novel and significant as it gave a model-agnostic approach for analyzing the expressive power of GNNs and solved several conjectures.

---

> ### Author Response · Authors · 2021-11-22
> **Answer to Reviewer 2**
>
> We thank the reviewer for the positive review and appreciate that the reviewer finds our approach interesting and our contributions significant. The list of typo’s provided by the reviewer are corrected in the revision, and we also clarified some sentences as suggested. We next address the comments related to function applications and lower bounds.
>
> **The role of functions in $\mathsf{TL}$.**  The reviewer asks whether function applications do add to the separation power. This is a very good question. The upper bounds hold indeed for any set $\Omega$ of functions, including  $\Omega=\emptyset$, so adding function applications *does not increase* the separation power. The lower bounds also hold for $\Omega=\emptyset$, as shown in the proof of Proposition C.5. This seems rather counter-intuitive, but the reason is that in $\mathsf{TL}$ we do support addition, scalar multiplication and multiplication of expressions. These can be seen as specific function applications by themselves. Only addition and multiplication are needed to construct polynomials that can be used to interpolate between specific values, required to encode logical formulae (Proposition C.5). *In the revision (at the end of Section 4) we now include a comment on the impact of functions on the separation power.*
>
> **General lower bounds.** The reviewer raises the interesting question whether a general approach is possible for also obtaining lower bounds in terms of $\mathsf{CR}$ or $k$-$\mathsf{WL}$ tests. This seems challenging. One has to ensure that classes of $\mathsf{GNN}$s can simulate $\mathsf{CR}$ or $k$-$\mathsf{WL}$ by means of injective functions on multisets (as e.g., in Xu et al. (2019), Morris et al. (2019)), or that the classes of $\mathsf{GNN}$s are rich enough to compute homomorphism counts of treewidth $k$ patterns (Dell et al. (2018), Nguyen et al. (2020)). In some sense, these are just restatements of the requirement that $\mathsf{CR}$ or $k$-$\mathsf{WL}$ tests can be simulated, and still require a case by case analysis. We agree that having a more principled approach would be very valuable, but such an approach seems out of reach at the moment.
>
> * Keyulu Xu, Weihua Hu, Jure Leskovec, Stefanie Jegelka: How Powerful are Graph Neural Networks? ICLR 2019.
> * Christopher Morris, Martin Ritzert, Matthias Fey, William L. Hamilton, Jan Eric Lenssen, Gaurav Rattan, Martin Grohe:Weisfeiler and Leman Go Neural: Higher-Order Graph Neural Networks. AAAI 2019.
> * Holger Dell, Martin Grohe, Gaurav Rattan: Lovász Meets Weisfeiler and Leman. ICALP 2018.
> * Hoang Nguyen, Takanori Maehara: Graph Homomorphism Convolution. ICML 2020.

---

### Official Review · Reviewer_2fbv · 2021-11-05

**Correctness:** 4
**Technical Novelty And Significance:** 4
**Empirical Novelty And Significance:** Not applicable
**Recommendation:** 10
**Confidence:** 4

**Main Review:**

Strengths: the paper is very clearly written and makes a clear new connection between programming language and deep learning. Formalizing GNN as a TL is a new toolbox that allows the authors to get an unified theoretical analysis of the separating power of GNNs. In particular, they are able to get new results characterizing the expressive power of GNNs as a function of the number of layers. This alone is a very nice contribution.

Weakness: I think Section 6 on function approximation is less relevant. The main reason is that the authors consider the discrete topology on the set of graphs so that any function is continuous. But GNN are more restricted functions and are continuous as mapping acting on tensors. As a consequence, I think:
1- the results presented in section 6 are correct but could probably be proved with simpler arguments similar to the ones used in Appendix A of 'On the equivalence between graph isomorphism testing and function approximation with GNNs' by Zhengdao Chen, Soledad Villar, Lei Chen, Joan Bruna.
2- the consequences of Theorem 6.1 and Corollary 6.2 are not clear for GNNs as GNNs cannot model any function taking a graph as argument. Indeed the importance of continuity is highlighted in section 3 of 'On the Limitations of Representing Functions on Sets' by Edward Wagstaff, Fabian Fuchs, Martin Engelcke, Ingmar Posner, Michael Osborne.
I think the authors should address this issue in Section 6.2 by explaining the possible limitation of their approach. Note that Azizian and Lelarge probably consider continuous functions of tensor representation of the graph for this reason.

**Summary Of The Paper:**

This paper introduces a new approach to study the separation power and approximation properties of graph neural networks (GNN). The authors introduce the Tensor Languages (TL) and show the representation of GNN as TL expressions. In Section 4, they show the separation power of TL and relate it to color refinement algorithms like k-WL. In particular, they are able to characterize the separation power of such algorithms after a given number of iterations. These results allows them to compute in Section 5 the separation power of GNNs. They are able to recover all results in the literature in an unified way and prove some new results. In Section 6, they give consequences of their results in term of approximation of GNNs, recovering known results and proving new ones.
This is a theoretical paper without any experimental results.

**Summary Of The Review:**

Very nice contribution making a new connection between programming language and GNN to study their expressive power.

---

> ### Author Response · Authors · 2021-11-22
> **Answer to Reviewer 1**
>
> We thank the reviewer for the positive review and shared enthusiasm for our contributions.
>
> The reviewer only raises one concern related to the relevance of our approximation results. More specifically, the reviewer argues for considering a *continuous setting*, motivated by the results in Wagstaff et al. (2019), rather than the discrete vertex label setting considered in our paper. In accordance, *we have revised the paper such that the vertices in graphs can hold real-valued labels, thereby lifting the discrete to the continuous setting.* Indeed, our proof techniques generalise easily to graphs with continuous vertex labels (labels in some $\mathbb{R}^\ell$). As can be seen in the main paper, no changes are required to the definition of our tensor language $\mathsf{TL}$. Furthermore, we show how our proofs on the separation power of $\mathsf{TL}$ (Section 4) can be generalised to the continuous setting in the supplementary material (Section C.6). For the approximation results (Section 6), we now assume that $\mathcal{G}_s$ is a *compact set of graphs*, that is, we consider all graphs of size $n$ with vertex labels coming from a compact set $K$ in $\mathbb{R}^\ell$. As a consequence, we now consider the same setting as in Azizian et al. (2021), to which the reviewer also refers.
>
> As such, we believe that the limitation mentioned by the reviewer is lifted entirely in the revision. The reviewer suggests an alternative proof technique for our approximation results (for the discrete case) based on $\sigma$-algebras (Chen et al. (2019). Such an approach may indeed be feasible for the discrete case, but it is less clear how these techniques carry over the continuous setting which we now consider. By contrast, as in Azizian et al. (2021), our approach uniformly treats the discrete and continuous setting. One remark is that when looking at approximations by GNNs, for some of the architectures, lower bounds are only known for the discrete setting (e.g., GINs),  as also observed in Azizian et al. (2021).
>
> * Edward Wagstaff, Fabian Fuchs, Martin Engelcke, Ingmar Posner, Michael A. Osborne: On the Limitations of Representing Functions on Sets. ICML 2019.
> * Waiss Azizian, Marc Lelarge: Expressive Power of Invariant and Equivariant Graph Neural Networks. ICLR 2021.
> * Zhengdao Chen, Soledad Villar, Lei Chen, Joan Bruna: On the equivalence between graph isomorphism testing and function approximation with GNNs. NeurIPS 2019.

---

> > ### Comment · Reviewer_2fbv · 2021-11-30
> > **thank you for your answer**
> >
> > I raised my score to 10.

---

> > > ### Author Response · Authors · 2021-12-01
> > > **Response to comment on revision**
> > >
> > > We thank the reviewer for the very positive assessment of the original submission and the revision.

---

### Author Response · Authors · 2021-11-22
**General response**

We thank the reviewers for their positive and detailed assessment of our paper. In the revision, we clarify the novelty of our approach in relation to previous work. We extend our approach in two major ways to accommodate for comments by reviewers 1 and 4. First, we now consider graphs with vertex real-valued vertex labels, such that our separation and approximation results apply to a continuous setting. Second, we now allow for arbitrary aggregation functions, thereby enabling the analysis of GNN that use aggregation functions different from summation. We also revised the main paper in accordance with other comments raised. For reasons of space, not all comments could be addressed in the main paper and had to be deferred to the supplementary material. Changes in the main paper are marked in blue. New sections and additional details in the supplementary material are marked with a vertical line in the margin.

---

### Decision · Program_Chairs · 2022-01-20

**Decision:**

Accept (Oral)

**Comment:**

This paper gives a new theoretical framework to characterize the expressive power of graph neural networks that describes GNN by tensor language (TL) and then makes it possible to analyze its expressive power through the lens of TL. The authors connect the expressive ability of TL to the color refinement algorithms and (vertex/graph) k-WL algorithms. By doing so, the several existing results can be recovered in a unifying manner. In addition to that, the function approximation ability is also investigated.

The paper gives a novel theoretical framework that gives a clear perspective to the problem of expressive power of GNN, which would be quite beneficial to the community and open up a new research direction. The reviewers have raised several questions on the paper, but the authors addressed all the concerns properly. Therefore, I recommend acceptance to ICLR2022.